

# Photochemical modeling of molecular and atomic oxygen based on multiple *in-situ* emissions measured during the Energy Transfer in the Oxygen Nightglow rocket campaign

Olexandr Lednyts'kyy[1] and Christian von Savigny[1]

[1]University of Greifswald, Greifswald, Germany

**Correspondence:** O. Lednyts'kyy (olexandr.lednytskyy@uni-greifswald.de), ORCID: 0000-0002-8343-6357

**Abstract.** Electronically excited states of molecular and atomic oxygen (six of $O_2$ and two of O) were implemented in the proposed Multiple Airglow Chemistry (MAC) model as minor species coupled with each other as well as with the ground states of $O_2$ and O to represent the photochemistry in the upper Mesosphere and Lower Thermosphere (MLT) region. The MAC model is proposed combining chemical processes of the well-known photochemical models related to identified $O_2$ and O species and some additional processes. Concentrations of excited $O_2$ and O species were retrieved using the MAC model on the basis of the multiple *in-situ* nightglow emissions measured during the Energy Transfer in the Oxygen Nightglow (ETON) rocket campaign. The proposed retrieval procedure to obtain concentrations of these MLT minor species is implemented avoiding *a priori* data sets. Unknown and poorly constrained reaction rates were tuned and reaction rates of the well-known models were updated with the MAC model comparing *in-situ* and evaluated emission profiles as well as *in-situ* and retrieved O concentration profiles. As a result, precursors of $O_2$ and O species responsible for transitions considered in the MAC model are identified and validated by calculations with the MAC model.

## 1 Introduction

Airglow is a permanent global atmospheric phenomenon that can be hardly seen without appropriate instruments. Ångström (1869) used such instruments and observed the green line emission at 557.7 nm in the nightglow (airglow at night) from the Earth's surface in 1868 for the first time. The origin of airglow was considered to be the same as the origin of aurora, a sporadic arc-like atmospheric phenomenon, which fascinated numerous spectators for many thousands of years.

Table 1 provides an overview of relatively strong airglow emissions detected in the upper Mesosphere and Lower Thermosphere (MLT) *in-situ* and remotely. The Energy Transfer in the Oxygen Nightglow (ETON) rocket campaign conducted in March 1982 and discussed in Section 2 was conceptualized to obtain *in-situ* profiles of airglow Volume Emission Rates (VER) and other atmospheric parameters like atomic oxygen (O) in the ground state ($O(^3P)$) to verify and validate photochemical models describing airglow.

$O(^3P)$ is a chemically active MLT trace gas which is a critical component of the energy budget of the MLT region. $O(^3P)$ is also required to retrieve carbon dioxide ($CO_2$) concentrations, profiles of kinetic temperature and pressure (Remsberg et al.,



2008; García-Comas et al., 2008; Rezac et al., 2015). $O(^3P)$ is also a major component of the neutral bath gas in the upper thermosphere significantly contributing to the nighttime ionosphere (Shematovich et al., 2011; Wei et al., 2014).

The transition $O(^1S - {}^1D)$ from the second excited O state ($O(^1S)$) to the first one ($O(^1D)$) is detected as the 557.7 nm green line emission. The Chapman excitation scheme and the Barth excitation transfer scheme were proposed in 1931 and

1962, respectively, to explain the origin of the green line emission in the MLT. The Chapman excitation scheme considers a collision of two $O(^3P)$ atoms and a third body represented by $O(^3P)$ to produce $O(^1S)$ (Chapman, 1931, 1937). The Barth excitation transfer scheme considers (1) a collision of two $O(^3P)$ atoms and a third body represented by an abundant molecule, e.g. molecular nitrogen ($N_2$) and oxygen ($O_2$), to produce $O_2$ in a not identified excited state $O_2^*$, and (2) an energy transfer from $O_2^*$ to $O(^3P)$ so that $O(^1S)$ is produced (Bates, 1979). Comparing both excitation schemes Bates (1979) interpreted the

Chapman excitation process to consist of four steps as follows: (1) two $O(^3P)$ atoms collide (2) creating a common surface of potential energy of, presumably, an electronically excited $O_2$ molecule in the upper Herzberg state (Greer et al., 1987), (3) after its collision with a third $O(^3P)$ atom (4) one vibrationally excited $O_2$ molecule and one $O(^1S)$ atom are created. One of the differences between the Chapman and Barth excitation schemes is the kind of a third body being an $O(^3P)$ atom or an abundant molecule in the MLT, respectively. The energy transfer considered in the Barth scheme includes $O_2^*$ acting as the

$O(^1S)$ precursor, but the Chapman scheme does not include it. Photochemical models proposed to implement the Chapman and Barth schemes are hereafter referred to be of the first (one-step) and the second (two-step) type, respectively.

Airglow emissions are very complex atmospheric phenomena so that photochemical models are often proposed to derive unknown or poorly constrained reaction rates, which can be backed up by reaction rates determined in the ground-based laboratory with the use of the Stern-Volmer method. The Stern-Volmer method is applied to analyze concentration dependent

kinetics in a homogeneous system, to which a quencher was added (Lakowicz, 2006). According to the Stern-Volmer method, excited and quenching chemical species are considered in a system of a few photochemical reactions so that steady-state methods can be applied to describe emissions. Then measurements of lifetimes or concentrations of emitting species enable determining the true pseudo-first order decay required to calculate the rate coefficient of the considered quenching reaction. However, the same values of the pseudo-first order decay rate are possible for both the dynamic quenching and the static

quenching at the given temperature (Lakowicz, 2006). Dynamic quenching reduces the apparent fluorescent lifetime, and static quenching rather reduces the apparent concentration of fluorescent species during inelastic collisions (Lakowicz, 2006). Unfortunately, reactive collisions responsible for the static quenching are not so well understood compared to the products of the dynamic quenching that can introduce difficulties calculating the rate coefficient of the considered quenching reaction.

If no more than one emission, e.g. VER$\{O(^1S - {}^1D)\}$ in McDade et al. (1986), is considered in the model of the second

type then the resulting mass conservation (continuity) equation is of the third degree with respect to $[O(^3P)]$, and the respective solutions can be easily interpreted. As for the $O_2(b - X)$ transition, McDade et al. (1986) developed photochemical models of the first and second types to describe transitions from $O_2(b)$ (the second electronically excited state of molecular oxygen, $O_2$) to $O_2(X)$ (the electronic ground state of $O_2$). This transition in the Atmospheric band was measured *in-situ* in the Earth's atmosphere during the ETON campaign to retrieve VER$\{O_2(b - X)\}$. The model of the second type with the $O_2(b)$ precursor

and $O_2(b)$ was proposed to explain non-linearities detected in quenching processes simulated by using the model of the first





type with $O_2(b)$ only. McDade et al. (1986) used known reaction rates and tuned poorly constrained reaction rates of these quenching processes in the atmosphere so that simulated profiles match the *in-situ* observations. The processes considered in the models of the first and second types and provided in Table 2 were developed by McDade et al. (1986) to describe atmospheric airglow emissions and to verify the obtained results in the ground-based laboratory using the Stern-Volmer relationship.

The total number of reactions considered in the models of López-González et al. (1992b) and McDade et al. (1986) with the $O(^1S)$ precursor $(O_2^*)$ and $O(^1S)$ was limited to ten, and these reactions are separated in two groups according to the Barth excitation transfer scheme. A full overview of these reactions including $O_2$ in a not identified excited state $O_2^*$ is not provided in this short overview excepting two reactions. Specifically, López-González et al. (1992b) considered the reaction $O(^1S) + O(^3P) \rightarrow products$, which McDade et al. (1986) did not consider. But McDade et al. (1986) considered the reaction

$O_2^* + N_2 \rightarrow products$, which López-González et al. (1992b) did not consider. Possible reasons to limit the list of all possible reactions in these models are as follows: (1) the Barth excitation transfer scheme can be represented by the most important (e.g., ten) reactions, (2) the system of a few reactions can be easily represented by a low degree polynomial equation regarding $[O(^3P)]$, (3) additional reactions would introduce difficulties to derive their rates, which can also be considered in ratio values tuned as empirical coefficients, and (4) the choice of approaches applied to derive empirical coefficients is limited depending

on the considered reactions, e.g., compare approaches applied by McDade et al. (1986) and López-González et al. (1992b).

These reasons limit the applicability of the mentioned methods used to analyze laboratory results and atmospheric measurements, which are usually studied without propagation in time. The computational simulation of a chemical kinetics system enables studying the time evolution of chemical species by using the ordinary differential equations (ODE) system matrix and initial conditions, see, e.g., Sandu and Sander (2006) for an overview of zero-dimensional box models developed to integrate

ODEs numerically in time. Unfortunately, computer modeling depends on *a priori* data sets used to initialize a box model. *In-situ* atmospheric measurements may be influenced by gravity waves and atmospheric tides at the particular moments of time that hinders the use of box models on the basis of such measurements. This article here studies the MLT photochemistry on the basis of the *in-situ* ETON measurements with the use of steady state continuity equations, i.e. without propagation in time, and without *a priori* data sets used for it.

The ETON multiple airglow emissions are described in Section 2, they can be applied simultaneously in the model proposed in this study to decrease uncertainties when tuning unknown and poorly constrained reaction rates with the use of the verification and validation procedures.

The first application of multiple emissions measured remotely with spectrometers in a model considering several $O_2$ states seems to be done by Torr et al. (1985) analyzing data sets from the shuttle Spacelab 1. In fact, these data sets were extremely

scattered in the time and place that might have stopped Torr et al. (1985) combining identified $O_2$ states in one model. Instead, they considered a number of photochemical models with some excited $O_2$ states in each model so that all discussed excited $O_2$ states appeared to be uncoupled with each other. Note that Torr et al. (1985) also considered $O_2(c)$ as the $O(^1S)$ precursor according to Greer et al. (1981), and applied the $O(^1S)$ quenching with $O_2(a)$ according to Bates (1981) and Kenner and Ogryzlo (1982).



In summary, this investigation was conducted to study the following topics regarding the new photochemical model proposed here: (1) processes of the $O(^1S)$ formation and quenching, see Section 3.1, (2) processes including identified $O_2$ states, see Sections 3.2.1, 3.2.2 and 3.3, and (3) the $O(^1S)$ precursor represented by one $O_2$ state or a group of them, see Section 4.4.

The $O(^3P)$ retrieval scheme was proposed to be solved in subsequent steps as described in Appendix A on the basis of
multiple airglow emission profiles that was discussed by Lednyts′kyy and von Savigny (2016) and Lednyts′kyy et al. (2018). Note that *a priori* data are not required to initiate calculations with in the MAC model. Concentrations of $O_2$ in higher excited states are calculated earlier at these retrieval steps, and are used to calculate concentrations of $O_2$ in lower excited states at the other retrieval steps. It should be noted that a limited number of multiple airglow emissions available from the ETON measurements or other sources can be also applied to retrieve $[O(^3P)]$ values at some of the mentioned retrieval steps, see
Sections 2 and 5 for details.

## 2  Data sets applied in the Multiple Airglow Chemistry (MAC) model

*In-situ* measurements obtained during the Energy Transfer in the Oxygen Nightglow (ETON) campaign and simulations using the most recent version of the MSIS (Mass Spectrometer Incoherent Scatter) semi-empirical model are in the focus of this section.

Volume Emission Rates (VER) of the nightglow emissions measured *in-situ* during the ETON campaign and the corresponding statistical errors provided by Greer et al. (1986) were used in this study. The ETON campaign is comprised of measurements obtained during coordinated launches of seven sounding rockets at South Uist ($\sim$57° 16′ N, $\sim$7° 19′ W) in Scotland, Great Britain, in westerly direction on 23rd March 1982 from $\sim$21:27 UT to $\sim$23:55 UT (Greer et al., 1986, 1987).

The maximal number of VER profiles related to various $O_2$ and O transitions were obtained by two ETON rockets, which
are discussed here. As for the Infrared Atmospheric band emissions at 1.27 $\mu$m, they were measured with a photometer aboard only one ETON rocket: the P227H rocket launched at $\sim$22:11 UT. The Herzberg I and Atmospheric band emissions at 320 and 761.9 nm, respectively, were also measured by the P227H rocket. The P229H rocket was launched at $\sim$22:58 UT right after the P227H rocket and provided measurements of the Herzberg I, Chamberlain and Atmospheric band emissions at 330, 370 and 761.9 nm, respectively, as well as the oxygen green line emission at 557.7 nm. It should be noted that the Chamberlain
band emissions were measured by the P229H rocket only. The absolute accuracy of $\pm$20% in VER peak values for the Infrared Atmospheric band emissions and better than $\pm$10% in other wavelength ranges (Greer et al., 1986) introduces uncertainties in the $[O(^3P)]$ retrievals.

*In-situ* measurements of atomic oxygen (O) concentrations ($[O]$) in the ground state ($[O(^3P)]$) were carried out directly by the rockets P232H and P234H launched at $\sim$21:49 UT and $\sim$23:55 UT, respectively. $[O(^3P)]$ values were determined directly
using the resonance fluorescence and absorption technique at $\sim$130 nm (Greer et al., 1986) and were interpolated for the launch time of the other ETON rockets. The statistical (and the systematic) error was less than about $\pm$10% (and about $\pm$30%) at about 100 km (where $[O(^3P)]$ peak values are measured) and increased up to $\pm$50% (and about $\pm$20%) at other altitudes (where $[O(^3P)]$ low values are measured) (Greer et al., 1986).





The most recent version of the MSIS model, NRLMSISE-00 (Naval Research Laboratory MSIS Extended, 2000), was used to obtain the following input parameters required to run the MAC model: temperature (T), molecular nitrogen concentrations ($[N_2]$) and $[O_2]$. Because the highest number of $O_2$ and O transitions were sounded by the P229H rocket compared to those using the P227H rocket, the time of *in-situ* measurements obtained by the P229H rocket at $\sim 97\,km$ over South Uist in Scotland were specified for the NRLMSISE-00 model. It should be mentioned that McDade et al. (1986) developed the well-known cubic equation deriving empirical coefficients by using the MSIS-83 model, which is not available anymore.

The input parameters required to run the established models and the proposed MAC model are profiles of T, $[N_2]$, $[O_2]$ and VER values. The following abbreviations of *in-situ* VER profiles are used in this study: $VER\{O_2(A-X)\}$ (Herzberg I band, HzI), $VER\{O_2(A'-a)\}$ (Chamberlain band, Cha), $VER\{O_2(b-X)\}$ (Atmospheric band, Atm), $VER\{O(^1S - {}^1D)\}$ (green line, GrL) and $VER\{O_2(a-X)\}$ (Infrared Atmospheric band, IRAtm). Some of $O_2$ transitions listed in Table 1 correspond to these VER profiles. Note that the other listed $O_2$ transitions were also considered in the proposed MAC model, see Section 3.3 for details. It is worth mentioning that all of these $O_2$ transitions were measured remotely using the instrument SCIAMACHY (SCanning Imaging Absorption spectroMeter for Atmospheric CHartographY) aboard the satellite ENVISAT (ENVIronmental SATellite) launched by the European Space Agency (Burrows et al., 1995; Bovensmann et al., 1999). Gravity waves and atmospheric tides influence *in-situ* measurements, but averaging of the long time measurement series in latitude intervals enabled suppressing effects caused by gravity waves, although tidal effects were still present, see, e.g., Lednyts′kyy et al. (2017) for analysis of the SCIAMACHY datasets.

It should be mentioned that Lednyts′kyy and von Savigny (2016) tuned unknown or poorly constrained reaction rates considered in the MAC model on the basis of data sets obtained during the ETON campaign. The corresponding reaction rates are shown in Tables 9 … 12. Then, the MAC model was applied on the basis of data sets obtained during three campaigns as follows: the WADIS-2 (WAve propagation and DISsipation in the middle atmosphere), WAVE2000 (WAVes in airglow structures Experiment, 2000) and WAVE2004 campaigns, see the next publication. The WADIS-2 rocket provided all data sets required to retrieve $[O(^3P)]$ values. Data sets measured *in-situ* with rockets launched during the WAVE2000 and WAVE2004 campaigns were combined with the collocated data sets measured remotely. Convincing retrieval results enabled validating tuned reaction rates and calculations carried out with the MAC model.

# 3 Development of the MAC model

## 3.1 The $O(^1S)$ nightglow model with $O_2^*$ as the $O(^1S)$ precursor

The established photochemical models of McDade et al. (1986), Gobbi et al. (1992) and Semenov (1997) related to the oxygen green line emission are described in this section in short, see Lednyts′kyy et al. (2015) for details.

McDade et al. (1986) considered processes provided in Table 2 that resulted in two photochemical models according to the two-step Barth excitation transfer scheme implemented in each model and involving precursors of $O_2(b)$ and $O(^1S)$, respectively. McDade et al. (1986) also implemented one model according to the one-step excitation scheme and related to $O_2(b)$, but excluding the $O_2(b)$ precursor. Both models related to $O_2(b)$ were used to retrieve $[O(^3P)]$ on the basis of Volume



**Table 1.** Relevant optical transitions of terrestrial airglow in the Earth's atmosphere. Emissions (see column "Emission") observed in the wavelength range shown in column "$\lambda$" are denoted by abbreviations (see column "Ident.") and transition types (see column "Type"). Typical intensity values of an integrated (limb) emission rate profile are given for nightglow (see column "Int." before the comma) and, if available, dayglow (see column "Int." after the comma). Altitudes of the corresponding emission rate peaks are shown in column "Alt.". Atomic oxygen emissions are denoted by abbreviations as follows: **GrL** is for the green line emission at 557.7 nm, **ReL** – the red line emissions at 630.0 and 636.4 nm, **UVL** and **UVL**$^*$ – the ultraviolet line emissions at 297.2 and 295.8 nm, respectively. Molecular oxygen emissions are denoted by abbreviations as follows: **IRAtm** is for the InfraRed Atmospheric band emission at 1270 nm, **Atm** – the Atmospheric band emission at 761.9 nm, **Nox** – the Atmospheric band emission at 1908 nm, **HzI** – the Herzberg I band emissions, **BG** – the Broida-Gaydon band emissions, **Cha** – the Chamberlain band emissions, **HzIII** – the Herzberg III band emissions, **HzII** – the Herzberg II band emissions, **cbK** – the New system band emissions measured by using the Keck I/II instrument (Slanger et al., 2004a), **RJ** – the Richards-Johnson band emissions. References are marked with upper indices as follows: sc is for Slanger and Copeland (2003), mc – McConkey et al. (1966), na – Nagy et al. (2008), md – McDade (1998), na – Nagy et al. (2008), kh – Khomich et al. (2008).

| Emission | Ident. | $\lambda$ (nm) | Int., night, day | Alt.(km) |
|---|---|---|---|---|
| $O_2(A^3\Sigma_u^+ - b^1\Sigma_g^+)$ | BG | 300…1100$^{kh}$ | | |
| $O_2(A^3\Sigma_u^+ - X^3\Sigma_g^-)$ | HzI | 240…520$^{kh}$ | 600 R, 600 R | 98.8$^{kh}$ |
| $O_2(A'^3\Delta_u - a^1\Delta_g)$ | Cha | 300…870$^{kh}$ | 150 R, 150 R | 98.3$^{kh}$ |
| $O_2(A'^3\Delta_u - X^3\Sigma_g^-)$ | HzIII | 260…600$^{kh}$ | 70$^{kh}$ R | 97.7$^{kh}$ |
| $O_2(c^1\Sigma_u^- - b^1\Sigma_g^+)$ | cbK | 384…550$^{sc}$ | 30 R | |
| $O_2(c^1\Sigma_u^- - a^1\Delta_g)$ | RJ | 280…1000$^{kh}$ | | |
| $O_2(c^1\Sigma_u^- - X^3\Sigma_g^-)$ | HzII | 250…530$^{kh}$ | 50 R, 50 R | 98.1$^{kh}$ |
| $O(^1S - ^1D)$ | GrL | 557.7 | 300 R, 4…13 kR | 97$^{md}$ |
| $O(^1S - ^3P_1)$ | UVL | 297.2 | 30 R, 0.4…1.3 kR | |
| $O(^1S - ^3P_2)$ | UVL$^*$ | 295.8 | 0.1$^{mc}$ R | |
| $O(^1D - ^3P_2, ^3P_1)$ | ReL | 630.0, 636.4 | 0…50 R, 50 kR | 250$^{na}$ |
| $O_2(b^1\Sigma_g^+ - a^1\Delta_g)$ | Nox | 1908 | | |
| $O_2(b^1\Sigma_g^+ - X^3\Sigma_g^-)\{0-0\}$ | Atm | 761.9$^{na}$ | 5 kR, 100 kR | 94$^{md}$ |
| $O_2(a^1\Delta_g - X^3\Sigma_g^-)\{0-0\}$ | IRAtm | 1270$^{na}$ | 50 kR | 90$^{md}$ |

Emission Rates (VER) of the Atmospheric band emissions. All processes of the $O_2(b)$-model involving the $O_2(b)$ precursor are provided in the upper part of Table 2. In fact, reactions related to $O_2^{**}$ ($R_{u1.1-2}^P$, $R_{u3.1-3}^P$ and $R_{u4.0}^P$) are absent in the model implemented without the $O_2(b)$ precursor. The model implemented without the $O_2(b)$ precursor exhibits non-linearities in quenching processes, but the model implemented with the $O_2(b)$ precursor ($O_2^{**}$) does not result in such non-linearities (McDade et al., 1986). Note that McDade et al. (1986) described the green line emission considering the $O(^1S)$ precursor according to the Barth excitation transfer scheme. In fact, the well-known quadratic equation resulting from the model with the $O_2(b)$ precursor and the well-known cubic equation resulting from the model with the $O(^1S)$ precursor were concluded by McDade et al. (1986) to be favorable comparing to models based on the one-step (Chapman) excitation scheme. It is worth





mentioning that Grygalashvyly et al. (2018) proposed a model combining the Chapman and Barth excitation schemes, which were implemented in both $O_2(b)$-models of McDade et al. (1986) separately. Applying self-consistent data sets (see Section 2) and fitting retrieved data sets, Grygalashvyly et al. (2018) applied methods of McDade et al. (1986) to derive new values of empirical coefficients, which were initially derived by McDade et al. (1986) for the well-known quadratic equation. The newly

derived coefficients were preferred by Grygalashvyly et al. (2018) to be applied in their model.

The well-known cubic equation of McDade et al. (1986) provided here in the full form and in the short form was used here to retrieve $[O(^3P)]$ on the basis of VER of the green line emission ($\text{VER}_{558}$ also referred to as $\text{VER}\{O(^1S - ^1D)\}$).

The well-known cubic equation provided by McDade et al. (1986) in the full form is as follows:

$$\frac{A_{558}\kappa_1[O(^3P)]^3([N_2]+[O_2])}{\text{VER}_{558}(A_{1S} + ^3\kappa_5[O_2])} = \frac{1}{\beta\delta}\frac{A^*}{\beta^*_O} + \frac{1}{\beta\delta}[O(^3P)] + \frac{1}{\beta\delta}\frac{\beta^*_{O_2}}{\beta^*_O}[O_2] + \frac{1}{\beta\delta}\frac{\beta^*_{O_2}}{\beta^*_O}R\frac{\beta^*_{N_2}}{\beta^*_{O_2}}[O_2], \tag{1}$$

where $R \approx 4$ represents the mean $[N_2]/[O_2]$ ratio valid in the altitude range $80\ldots120$km according to McDade et al. (1986). All reaction rates shown in Eq. (1) correspond to the ones provided in the lower part of Table 2. Ratios of some of these reaction rate values were derived by McDade et al. (1986), see empirical coefficients in Eq. (2) of Murtagh et al. (1990), on the basis of the ETON *in-situ* measurements as well as simulated temperature, $[N_2]$ and $[O_2]$ profiles.

The well-known cubic equation and the derived empirical coefficients in particular were verified by Murtagh et al. (1990),

who provided the well-known cubic equation in the short form as follows:

$$\text{VER}_{558} = \kappa_1[O(^3P)]^2([N_2]+[O_2])\cdot$$

$$\frac{[O]}{C(0)+C(1)[O(^3P)]+C(2)[O_2]}\cdot$$
$$\frac{A_{558}}{A_{1S} + ^3\kappa_5[O_2]}, \tag{2}$$

where the rate coefficient of the $R_{g1.2}$ reaction provided in Table 2 is $^3\kappa_5 = 4 \cdot 10^{-12}\exp(-865/T)\,\text{molec}^{-1}\,\text{cm}^3\,\text{s}^{-1}$, the Einstein coefficients of the reactions $R_{g3.0}$ and $R_{g(3-4).0}$ are $A_{558} = 1.18\,\text{s}^{-1}$ and $A_{1S} = 1.35\,\text{s}^{-1}$, the rate $\beta\kappa_1$ of the three-

body recombination reaction $R^P_{v1.1-2}$ is the product of $\kappa_1 = 4.7 \cdot 10^{-33}(300/T)^2\,\text{molec}^{-2}\,\text{cm}^6\,\text{s}^{-1}$ and an empirical $\beta$ value. The $R^P_{v1.1-2}$ reaction refers to the first step of the Barth excitation transfer scheme describing the production of $O_2^*$, the $O(^1S)$ precursor. The rates $\beta^*_O, \beta^*_{N_2}, \beta^*_{O_2}$ of the $R^P_{v3.1-3}$ reactions describe the $O_2^*$ quenching. The $R^P_{v2.1}$ reaction with the rate value $\delta\beta^*_O$, where $\delta$ is an empirical value, refers to the second step of the Barth excitation transfer scheme resulting in $O(^1S) + O_2$.

The values of the empirical coefficients C(0), C(1) and C(2) are equal to 0, 211 and 15, respectively, and these values are used

in this study for retrievals using the well-known cubic equation according to Murtagh et al. (1990). Note that these empirical coefficients were derived by McDade et al. (1986) using the semi-empirical models including MSIS-83, which is not available nowadays. The NRLMSISE-00 model mentioned in Section 2 is used in this study to simulate temperature, $[N_2]$ and $[O_2]$ profiles. McDade et al. (1986) used various available models that resulted in other values of temperature, $[N_2]$ and $[O_2]$ profiles and different values of the empirical coefficients. The minimal values of C(0), C(1) and C(2) from all obtained ones, which are

related to the $O(^1S)$ precursor, were found by McDade et al. (1986) to be equal to $13\pm4$, $183\pm10$ and $9\pm3$, respectively, and their maximal values were found to be equal to $23\pm9$, $224\pm20$ and $17\pm3$, respectively.



**Table 2.** Processes of the $O(^1S)$ nightglow model with $O_2^*$ as the $O(^1S)$ precursor were proposed by McDade et al. (1986) and modified by Lednyts′kyy et al. (2015) according to Gobbi et al. (1992) and Semenov (1997). Odd oxygen processes related to $O_2(b)$ were described with the well-known quadratic equation of McDade et al. (1986). Odd oxygen processes related to $O(^1S)$ were described by using two models. The first model excluded two processes, $R_{g1.2}$ and $R_{g2.1}$, and resulted in the well-known cubic Eq. (2) of McDade et al. (1986). The second model included two processes, $R_{g1.2}$ and $R_{g2.1}$, and resulted in the extended cubic Eq. (3) of Lednyts′kyy et al. (2015). The processes marked with a character $P$ were not considered proposing the MAC model, but were used in the 1$^{\text{st}}$ step (prior) retrieval of $[O(^3P)]$. Odd oxygen processes related to $O(^1S)$ represent the two-step Barth transfer scheme (see reactions $R_{v1.1-2}$, $R_{v2.1}$ and the resulting reaction $R_{g3.0}$) accompanied by quenching. The symbolic representation of the reaction rates shown above the arrows in the second column of this table was adopted from Khomich et al. (2008) and used in Section 3.1. The symbolic representation shown in the third column of this table was used by Lednyts′kyy et al. (2015). For instance, the reaction rate $\beta_O$ (Khomich et al., 2008) corresponds to $\gamma_{3P}^{\text{SP}}$ (Lednyts′kyy and von Savigny, 2016), $\beta_{O_2}$ (Khomich et al., 2008) corresponds to $\gamma_{O2}^{\text{SP}}$ (Lednyts′kyy and von Savigny, 2016), $A_{558}$ (Khomich et al., 2008) corresponds to $\gamma_{557n7}^{\text{A}}$ (Lednyts′kyy and von Savigny, 2016) and $A_{558} + (A_{297} + A_{296})$ (Khomich et al., 2008) corresponds to $\gamma_{1S3Pe}^{\text{A}}$ (Lednyts′kyy and von Savigny, 2016). Processes marked with a character P were used at the prior retrieval steps applied to calculate $[O(^1S)]$ (see Section A1.1) and $[O(^3P)]$ (see Sections 3.1 and 3.4).

| R$_\#$ | Odd oxygen processes related to $O_2(b)$ | Symbol |
|---|---|---|
| $R_{u1.1-2}^{P}$ | $O(^3P) + O(^3P) + \{N_2, O_2\} \xrightarrow{\alpha\kappa_1, \alpha\kappa_1} O_2^{**} + \{N_2, O_2\}$ | $\alpha\kappa_1$ |
| $R_{u2.1}^{P}$ | $O_2^{**} + O_2 \xrightarrow{\gamma, {}^3\kappa_3} O_2(b) + O_2$ | $C^{O2}$ |
| $R_{u3.1-3}^{P}$ | $O_2^{**} + \{O(^3P), N_2, O_2\} \xrightarrow{{}^1\kappa_3, {}^2\kappa_3, {}^3\kappa_3} \text{All products}$ | $C^{O}$ |
| $R_{u4.0}^{P}$ | $O_2^{**} \xrightarrow{\beta_u^{\text{A}}} O_2 + h\nu$ | $A_u$ |
| $R_{u5.1-3}^{P}$ | $O_2(b) + \{O(^3P), N_2, O_2\} \xrightarrow{{}^1\kappa_2, {}^2\kappa_2, {}^3\kappa_2} \text{Quenched products}$ | ${}^i\kappa_2$ |
| $R_{b5.0}$ | $O_2(b) \xrightarrow{\beta_{762}^{\text{A}}} O_2 + h\nu(\lambda = 762\,\text{nm})$ | $A_{762}$ |
| $R_{b6.0}$ | $O_2(b) \xrightarrow{\beta_{\text{Atm}}^{\text{A}}} O_2 + h\nu(\text{Atmospheric band})$ | $A_{Atm}$ |

| R$_\#$ | Odd oxygen processes related to $O(^1S)$ | Symbol |
|---|---|---|
| $R_{v1.1-2}^{P}$ | $O(^3P) + O(^3P) + \{N_2, O_2\} \xrightarrow{\alpha_{O_2}, \alpha_{O_2}} O_2^{*} + \{N_2, O_2\}$ | $\beta\kappa_1$ |
| $R_{v2.1}^{P}$ | $O_2^{*} + O(^3P) \xrightarrow{\delta, \beta_O^{*}=\alpha_O} O(^1S) + O_2$ | $C(1)$ |
| $R_{v3.1-3}^{P}$ | $O_2^{*} + \{O(^3P), N_2, O_2\} \xrightarrow{\beta_O^{*}, \beta_{N_2}^{*}, \beta_{O_2}^{*}} O_2 + \{O(^3P), N_2, O_2\}$ | $C(2)$ |
| $R_{v4.0}^{P}$ | $O_2^{*} \xrightarrow{A^{*}} O_2 + h\nu$ | $C(0)$ |
| $R_{g1.1}$ | $O(^1S) + O(^3P) \xrightarrow{\beta_O} O(^1D) + O(^1D)$ | ${}^1\kappa_5$ |
| $R_{g2.1}$ | $O(^1S) + N_2 \xrightarrow{\beta_{N_2}} O(^3P) + N_2$ | ${}^2\kappa_5$ |
| $R_{g1.2}$ | $O(^1S) + O_2 \xrightarrow{\beta_{O_2}} O(^3P) + O_2$ | ${}^3\kappa_5$ |
| $R_{g3.0}$ | $O(^1S) \xrightarrow{A_{558}} O(^1D) + h\nu(\lambda = 557.7\,\text{nm})$ | $A_{558}$ |
| $R_{g(3-4).0}$ | $O(^1S) \xrightarrow{A_{558}+(A_{297}+A_{296})} \{O(^1D), O(^3P)\} + h\nu$ | $A_{1S}$ |




Gobbi et al. (1992) suggested that processes of the enhanced $O(^1S)$ quenching with $O(^3P)$ and $N_2$ should also be considered in the well-known Eq. (2). The extended cubic equation provided by Gobbi et al. (1992) is as follows:

$$\text{VER}_{558} = \kappa_1[O(^3P)]^2([N_2] + [O_2]) \cdot$$

$$\frac{[O(^3P)]}{C(0) + C(1)[O(^3P)] + C(2)[O_2]} \cdot$$

$$\frac{A_{558}}{A_{^1S} + {}^1\kappa_5[O(^3P)] + {}^2\kappa_5[N_2] + {}^3\kappa_5[O_2]}, \tag{3}$$

where the rate coefficients corresponding to the reactions $R_{g1.1}$, $R_{g2.1}$ and $R_{g1.2}$ are ${}^1\kappa_5 = 2 \cdot 10^{-14}\,\text{molec}^{-1}\,\text{cm}^3\,\text{s}^{-1}$, ${}^2\kappa_5 = 5 \cdot 10^{-17}\,\text{molec}^{-1}\,\text{cm}^3\,\text{s}^{-1}$ and ${}^3\kappa_5 = 4.9 \cdot 10^{-12}\exp(-885/T)\,\text{molec}^{-1}\,\text{cm}^3\,\text{s}^{-1}$, respectively. The other coefficients are shown and described for Eq. (2). The photochemical model resulting in the extended cubic equation is hereafter referred to as the G-model in short according to the surname of the first author in Gobbi et al. (1992), who proposed this model.

The $O(^1S)$ quenching with $N_2$ is not effective according to Atkinson and Welge (1972) because the ${}^2\kappa_5$ value is five orders

lower than the ${}^3\kappa_5$ value of the $O(^1S)$ quenching with $O_2$. Therefore, the $O(^1S)$ quenching with $N_2$ was neglected (${}^2\kappa_5 = 0$) by Semenov (1997), who considered a relatively high ${}^1\kappa_5$ value of $5 \cdot 10^{-11}\exp(-305/T)\,\text{molec}^{-1}\,\text{cm}^3\,\text{s}^{-1}$ compared to the ${}^1\kappa_5$ value of $2 \cdot 10^{-14}\,\text{molec}^{-1}\,\text{cm}^3\,\text{s}^{-1}$ used by Gobbi et al. (1992). The low ${}^1\kappa_5$ value was obtained by Krauss and Neumann (1975) theoretically and approved experimentally by Kenner and Ogryzlo (1982), but Johnston and Broadfoot (1993) and a number of other scientists including Khomich et al. (2008) used the high ${}^1\kappa_5$ value.

Semenov (1997) developed the photochemical model that resulted in the cubic equation as follows:

$$\text{VER}_{558} = \alpha_{O_2}[O(^3P)]^2([N_2] + [O_2]) \cdot$$

$$\frac{\alpha_O[O(^3P)]}{A^* + \beta_{N_2}^*[N_2] + \beta_O^*[O(^3P)] + \beta_{O_2}^*[O_2]} \cdot$$

$$\frac{A_{558}}{A_{^1S} + {}^1\kappa_5[O(^3P)] + {}^3\kappa_5[O_2]}, \tag{4}$$

where $\alpha_{O_2} = \beta\kappa_1$ and $\alpha_O = \delta\beta_O^*$, see the notation of process rates provided in Table 2. The notation of other process rates shows that Eq. (4) can be transformed into Eq. (3) by using $A^* + \beta_{N_2}^*[N_2] := C(0)$, $\beta_O^* := C(1)$, $\beta_{O_2}^* := C(2)$ and ${}^2\kappa_5 =$

$0\,\text{molec}^{-1}\,\text{cm}^3\,\text{s}^{-1}$.

The $O(^1S)$ quenching with $O_2(a)$ is very effective according to Bates (1981) and Kenner and Ogryzlo (1982), but the direct inclusion of $O_2(a)$ in Eq. (3) would increase its order so that the number of the obtained solutions would be very complicated to interpret. Therefore, the high ${}^1\kappa_5$ value of the $O(^1S)$ quenching with $O(^3P)$ was adopted by Lednyts′kyy et al. (2015) in order to implicitly include the $O(^1S)$ quenching with $O_2(a)$ and to keep the order of the polynomial in Eq. (3). In this

context it is worth mentioning that – according to Garcia and Solomon (1985) – $O(^1S)$ quenching reactions are not completely established. The direct correspondence of Eq. (4) (with defined empirical coefficients C(0), C(1) and C(2)) and Eq. (3) to each other enabled specifying how the specific relationship between values of $[O(^3P)]$ and $\text{VER}_{558}$ can be used to solve both Eqs. (4) and (3) applying the analytical method of Semenov (1997).



The well-known cubic Eq. (2) represents the reduced form of the extended Eq. (3). Indeed, the reaction rates $^1\kappa_5$ and $^2\kappa_5$ are not equal to zero in the extended Eq. (3) to represent the $O(^1S)$ quenching with $O_2$, $O(^3P)$ and $N_2$. On the contrary, if they are equal to zero then the extended Eq. (3) becomes identical to the well-known Eq. (2). The other values of reaction rates and empirical coefficients were proposed by Lednyts′kyy et al. (2015) to be the same in both Eqs. (2) and (3) and calculated by

Lednyts′kyy et al. (2015) according to the discussion provided in the next paragraph. The $[O(^3P)]$ retrievals obtained by using Eqs. (2) and (3) were verified by Lednyts′kyy et al. (2015), analyzed by von Savigny and Lednyts′kyy (2013), von Savigny et al. (2015) and Lednyts′kyy et al. (2017) and validated here, see Section 3.4 for details.

Gobbi et al. (1992) used *in-situ* measurements obtained during the solar minimum phase at the transition from solar cycle 21 to cycle 22, but the ETON *in-situ* measurements were obtained during the solar maximum phase of the 21st solar cycle. It

is worth being mentioned that Gobbi et al. (1992) used Eq. (3) instead of Eq. (2) with the same empirical coefficients derived by McDade et al. (1986). Lednyts′kyy et al. (2015) adjusted values of these empirical coefficients applied here according to publications related to various phases of solar activity. This was done to reflect differences in ultraviolet irradiance and optical depth values during phases of the solar maximum and minimum. Indeed, Dudok de Wit et al. (2009) and Meier (1991) reported that the irradiance in the extreme ultraviolet wavelength range 30...121 nm affects thermospheric $O(^3P)$, $O_2$, $N_2$, N

and $N_2O$ ionization. Colegrove et al. (1965) emphasized that $O(^3P)$ is generated in the lower thermosphere and transported downwards to the mesosphere. Equation (2) of Murtagh et al. (1990) was extended by Lednyts′kyy et al. (2015) with the empirical coefficient $C(0) \neq 0$ because the first term on the right hand side of Eq. (1) is not equal to zero so that $C(0)$ should be introduced. However, the influence of $C(0)$ on solutions of Eq. (2) is negligible compared to $C(1)[O(^3P)]$ or $C(2)[O_2]$ so that the exact $C(0)$ value is not important. The NRLMSISE-00 model was applied adjusting the empirical coefficients $C(0)$, $C(1)$

and $C(2)$ instead of the MSIS-83 model applied by McDade et al. (1986).

In summary, polynomial equations of the second and the third orders with respect to $[O(^3P)]$ (McDade et al., 1986) are used to retrieve $[O(^3P)]$, see left panels in Figs. 4 and 5 in Section 3.4. The extended cubic Eq. (3) was solved for this study using the analytical method of Semenov (1997) also described by Khomich et al. (2008). As for the well-known cubic Eq. (2), it solved for this study using the program available at https://idlastro.gsfc.nasa.gov/ftp/contrib/freudenreich/cuberoot.pro within

the Astronomy User's Library distributed by the National Aeronautics and Space Administration. Note that values of reaction rates and empirical coefficients provided by Lednyts′kyy et al. (2015) were used according to the extended cubic Eq. (3) for $O(^3P)$ retrievals in this study. As for the well-known cubic Eq. (2) used for $O(^3P)$ retrievals in this study, values of reaction rates and empirical coefficients provided by Murtagh et al. (1990).

Photochemical models based on identified $O_2$ states and their coupling with each other are described in the following Section

30   3.2.

### 3.2   Models with identified excited $O_2$ states

A short review regarding approaches developing photochemical models is provided in Section 1. The established photochemical models described in the following sections include $O_2(b, a, X)$ in the first model, see Section 3.2.1, and $O_2(c, b, X)$ in the second model, see Section 3.2.1, developed using available data sets.





### 3.2.1 The modified kinetic model of $O_2$ and $O_3$ photolysis products

A photochemical model taking $O_2(b, a, X)$ states and $O(^1D, {}^3P)$ states into account was developed by Mlynczak et al. (1993) with the use of the basic daytime $O_2(a)$ kinetic model employed by Thomas (1990). The model of Mlynczak et al. (1993) was extended by Sharp et al. (2014) by including the three-body recombination reaction producing $O_2(a)$ during night time, see the $R_{a1.1-2}$ reactions provided in Table 3. The model of Sharp et al. (2014) also included processes related to the laser excitation, but these processes are not relevant for the present study and are excluded.

All other processes of the model proposed by Sharp et al. (2014) are shown in Table 3. The modified kinetic model with these processes is hereafter referred to as the M-model in short according to the surname of the first author in Mlynczak et al. (1993). Processes marked with a character E and shown in Table 3 were excluded from the resulting M-model because they were not found in the latest version of the 2015 database of the Jet Propulsion Laboratory (Burkholder et al., 2015).

The M-model was verified on the basis of a few emission lines (with high signal to noise ratios) from possible band emissions measured remotely. Some of these strongest $O_2$ nightglow emissions are provided in Table 1. One of them in the Infrared Atmospheric band is represented by the vibrational transition $0-0$ of the forbidden electronic transition $a^1\Delta_g, \nu' = 0 - X^3\Sigma_g^-, \nu'' = 0$ ($O_2(a-X)\{0-0\}$). Note that processes of the M-model were used to develop the MAC model on the basis of VER profiles from the ETON campaign including VER values of $O_2(a-X)\{0-0\}$ (VER$\{O_2(a-X)\}$). Note that Yankovsky et al. (2016) developed the new YM2011 model considering $O(^1D)$ and various electronic-vibrationally excited levels in the new model of electronic-vibrational kinetics: 3 of $O_2(b, \nu \leq 2)$, 6 of $O_2(a, \nu \leq 5)$ and 35 of $O_2(X, \nu \leq 35)$. In contrast to the YM2011 model, reaction rates in the modified kinetic model of Mlynczak et al. (1993) refer to a specific portion of vibrational states from their statistical equilibrium in each $O_2$ electronic state. Yankovsky et al. (2016) and Yankovsky et al. (2007) reported on differences among $O_3$ altitude profiles obtained by using the modified kinetic model of Mlynczak et al. (1993) and the YM2011 model.

### 3.2.2 The extended $O(^1S)$ nightglow model with $O_2(c)$ as the $O(^1S)$ precursor

A photochemical model taking $O_2(c, b, X)$ states and $O(^1S, {}^1D, {}^3P)$ states into account was developed by Huang and George (2014) on the basis of the photochemical $O(^1S)$ nightglow model proposed by Hickey et al. (1997). The first implementation of $O_2(c)$ as the $O(^1S)$ precursor seems to be carried out by Torr et al. (1985) on the basis of multiple emissions simultaneously measured from the Spacelab 1 shuttle. The $O(^1S)$ precursor was also assumed to be $O_2(c)$ by Greer et al. (1981) describing *in-situ* measurements of the ETON campaign and Hickey et al. (1997).

Huang and George (2014) tuned some rates of quenching reactions on the basis of measurements of the green line emissions at 557.7 nm and the Atmospheric band emissions at 864.5 nm. The vibrational transition $0-1$ of the electronic transition $b^1\Sigma_g^+, \nu' = 1 - X^3\Sigma_g^-, \nu'' = 0$ at 864.5 nm can be observed from the Earth's surface and it is denoted as $O_2(b-X)\{0-1\}$. Volume Emission Rates (VER) of the $O_2(b-X)\{0-1\}$ transition are about 30 times less intense than VER$\{O_2(b-X)\}$ of the $O_2(b-X)\{0-0\}$ transition at 762.2 nm in the Atmospheric band (Meinel, 1950). Prof. Huang provided rate coefficients of the model of Huang and George (2014) for VER$\{O_2(b-X)\}$ of the $O_2(b-X)\{0-0\}$ transition, but the same reactions were





**Table 3.** Processes of the model of Mlynczak et al. (1993) modified by Sharp et al. (2014) are hereafter referred to as the M-model, see Section 3.2.1. Processes of $O_2$ and $O_3$ photolysis occur at sunlight conditions. The processes marked with a character $M$ are not considered in the MAC model shown in Tables 6 and 7 because they were not listed in the online version of the JPL 2015-year database (Burkholder et al., 2015) and were replaced by other relevant up-to-date processes.

| $R_\#$ | Odd oxygen processes related to $O_2(b)$, $O_2(a)$ and $O(^1D)$ |
|---|---|
| $R_{s1.2-3}$ | $O_2 + h\nu \xrightarrow{\sigma_{PD}^{LA},\sigma_{PD}^{Sc}} O(^3P) + \{O(^1D), O(^1D)\}$ |
| $R_{s2.3}$ | $O_3 + h\nu \xrightarrow{\sigma_{aD}^{Ha}} O(^1D) + O_2(a)$ |
| $R_{s3.1}$ | $O_2 + h\nu(\lambda = 762\,\mathrm{nm}) \xrightarrow{\sigma_{b1}^{O2}} O_2(b)$ |
| $R_{b2.1}^{E}$ | $O_2(b) + O_3 \xrightarrow{\beta_{O3}^{ba}} O_2(a) + O_2 + O(^3P)$ |
| $R_{b2.2-5}$ | $O_2(b) + \{O, N_2, O_2, CO_2\} \xrightarrow{\beta_{3P}^{ba},\beta_{N2}^{ba},\beta_{O2}^{ba},\beta_{C2}^{ba}} O_2(a) + \{O, N_2, O_2, CO_2\}$ |
| $R_{b5.0}$ | $O_2(b) \xrightarrow{\beta_{762}^{A}} O_2 + h\nu(\lambda = 762\,\mathrm{nm})$ |
| $R_{b6.0}$ | $O_2(b) \xrightarrow{\beta_{Atm}^{A}} O_2 + h\nu(\text{Atmospheric band})$ |
| $R_{a1.1-2}$ | $O(^3P) + O(^3P) + \{N_2, O_2\} \xrightarrow{\alpha_{N2}^{Pa},\alpha_{O2}^{Pa}} O_2(a) + \{N_2, O_2\}$ |
| $R_{a2.2-4}$ | $O_2(a) + \{O, N_2, O_2\} \xrightarrow{\alpha_{3P}^{ax},\alpha_{N2}^{ax},\alpha_{O2}^{ax}} O_2 + \{O, N_2, O_2\}$ |
| $R_{a3.0}$ | $O_2(a) \xrightarrow{\alpha_{1u27}^{A}} O_2 + h\nu(\lambda = 1.27\,\mu\mathrm{m})$ |
| $R_{a4.0}$ | $O_2(a) \xrightarrow{\alpha_{IRA}^{A}} O_2 + h\nu(\text{IR Atmospheric band})$ |
| $R_{r2.1,3}$ | $O(^1D) + \{N_2, O_2\} \xrightarrow{\rho_{N2}^{DP},\rho_{Ob}^{DP}} O(^3P) + \{N_2, O_2(b)\}$ |
| $R_{r2.2}^{E}$ | $O(^1D) + O_2 \xrightarrow{\rho_{O2}^{DP}} O(^3P) + O_2$ |

used for the transitions $O_2(b-X)\{0-0\}$ and $O_2(b-X)\{0-1\}$. All processes of the model of Huang and George (2014) are hereafter referred to as processes of the H-model with the capital H for the surname of the first author in Huang and George (2014).

It should be noted that both transitions $O_2(b-X)\{0-0\}$ and $O_2(b-X)\{0-1\}$ can be observed remotely from space,
5  e.g. using the SCIAMACHY instrument mentioned in Section 2 because radiation was measured using the SCIAMACHY instrument simultaneously and contiguously in the wavelength range from 240 to 1750 nm (Bovensmann et al., 1999).

As for a photochemical model considering electronic-vibrational kinetics according to the Barth excitation transfer scheme, Makhlouf et al. (1998) proposed to consider $O_2(c, \nu' \geq 3)$ instead of $O_2^*$ as the $O(^1S)$ precursor based on conclusions of Krasnopolsky (1981). The results of Makhlouf et al. (1998) were obtained for $O_2(c, \nu' = 0\ldots16)$ regarding the oxygen green
10  line emission simulating gravity wave-driven fluctuations like Huang and George (2014) did. Nevertheless, rate values of the $O_2(c, \nu' = 0, 1)$ quenching used by Makhlouf et al. (1998) differ from those ones used by Huang and George (2014) implying that these rate values derived by tuning the H-model depend on the used data sets. It should be mentioned that tuning results for the M-model also depended on the used data sets, see Section 3.2.1.




**Table 4.** Processes of the extended O($^1S$) nightglow model (Hickey et al., 1997; Huang and George, 2014) hereafter referred to as the H-model, see Section 3.2.2. The MAC modell includes all processes listed here and also the processes shown in Tables 6 and 7.

| R$_\#$ | Odd oxygen processes related O$_2$(c), O$_2$(b) and O($^1S$) |
|---|---|
| $R_{c1.1-2}$ | O($^3P$) + O($^3P$) + {N$_2$, O$_2$} $\xrightarrow{\varsigma_{N2}^{Pc}, \varsigma_{O2}^{Pc}}$ O$_2$(c) + {N$_2$, O$_2$} |
| $R_{c2.1}$ | O$_2$(c) + O($^3P$) $\xrightarrow{\varsigma_{1S}^{cx}}$ O$_2$ + O($^1S$) |
| $R_{c3.1-2}$ | O$_2$(c) + {O($^3P$), O$_2$} $\xrightarrow{\varsigma_{3P}^{cb}, \varsigma_{O2}^{cb}}$ O$_2$(b) + {O($^3P$), O$_2$} |
| $R_{c7.1}$ | O$_2$(c) + O($^3P$) $\xrightarrow{\varsigma_{3P}^{cx}}$ O$_2$ + O($^3P$) |
| $R_{c8.0}$ | O$_2$(c) $\xrightarrow{\varsigma_{HII}^{A}}$ O$_2$ + $h\nu$(Herzberg II band) |
| $R_{b1.1-2}$ | O($^3P$) + O($^3P$) + {N$_2$, O$_2$} $\xrightarrow{\beta_{N2}^{Pb}, \beta_{O2}^{Pb}}$ O$_2$(b) + {N$_2$, O$_2$} |
| $R_{b4.2-4}$ | O$_2$(b) + {O($^3P$), N$_2$, O$_2$} $\xrightarrow{\beta_{3P}^{bx}, \beta_{N2}^{bx}, \beta_{O2}^{bx}}$ O$_2$ + {O($^3P$), N$_2$, O$_2$} |
| $R_{b5.0}$ | O$_2$(b) $\xrightarrow{\beta_{762}^{A}}$ O$_2$ + $h\nu(\lambda = 762\,\text{nm})$ |
| $R_{b6.0}$ | O$_2$(b) $\xrightarrow{\beta_{Atm}^{A}}$ O$_2$ + $h\nu$(Atmospheric band) |
| $R_{g1.2}$ | O($^1S$) + O$_2$ $\xrightarrow{\gamma_{O2}^{SP}}$ O($^3P$) + O$_2$ |
| $R_{g3.0}$ | O($^1S$) $\xrightarrow{\gamma_{557n7}^{A}}$ O($^1D$) + $h\nu(\lambda = 557.7\,\text{nm})$ |
| $R_{g4.0}$ | O($^1S$) $\xrightarrow{\gamma_{1S3Pe}^{A}}$ O($^3P$) + $h\nu$ |
| $R_{x1.1-2}$ | O($^3P$) + O($^3P$) + {N$_2$, O$_2$} $\xrightarrow{\chi_{N2}^{Px}, \chi_{O2}^{Px}}$ O$_2$ + {N$_2$, O$_2$} |

### 3.3 Processes of the MAC model

Processes of the proposed MAC model extend a combination of the processes adopted from the G-model, see Section 3.1, the M-model, see Section 3.2.1 and the H-model, see Section 3.2.2. Rate values of the processes considered in these models were updated using the JPL 2015 database (Burkholder et al., 2015) and the database of the National Institute of Standards and Technology (NIST) available at https://www.nist.gov/pml/productsservices/physical-reference-data as well as other high ranking sources listed in Huestis (2002) and Jones et al. (2006).

The following processes were adopted in the MAC model from the M-model of Mlynczak et al. (1993) and Sharp et al. (2014), see Table 3 in Section 3.2.1. These processes are related to:

1. the photolysis of O$_2$ and O$_3$ ($R_{s1.2-3}$, $R_{s3.1}$, $R_{s2.3}$),

2. the Atmospheric band emission ($R_{b2.1}^{E}$, $R_{b2.2-5}$, $R_{b5.0}$, $R_{b6.0}$),

3. the Infrared Atmospheric band emission ($R_{a1.1-2}$, $R_{a2.2-4}$, $R_{a3.0}$, $R_{a4.0}$),

4. the red line emission ($R_{r2.1,3}$, $R_{r2.2}^{E}$).





It should be noted that the processes $R^E_{b2.1}$ and $R^E_{r2.2}$ were replaced by processes with other products according to Burkholder et al. (2015). These replaced processes and the other processes of the M-model were adopted in the proposed MAC model, they are also referred to as M-processes.

The following processes were adopted in the MAC model from the H-model of Huang and George (2014) and Hickey et al.
(1997), see Table 4 in Section 3.2.2. These processes are related to:

1. the singlet Herzberg state ($R_{c1.1-2}$, $R_{c2.1}$, $R_{c3.1-2}$, $R_{c7.1}$, $R_{c8.0}$),

2. the Atmospheric band emission ($R_{b1.1-2}$, $R_{b4.2-4}$, $R_{b5.0}$, $R_{b6.0}$),

3. the green line emission ($R_{g1.2}$, $R_{g3.0}$, $R_{g4.0}$),

4. the three-body recombination ($R_{x1.1-2}$).

These processes were all adopted in the proposed MAC model, they are also referred to as H-processes.

The following processes were adopted in the MAC model from the G-model of Gobbi et al. (1992), see Table 2 in Section
3.1. These processes are related to:

1. the green line emission ($R_{g1.1-2}$, $R_{g2.1}$, $R_{g3.0}$, $R_{g(3-4).0}$) and

2. the O($^1S$) precursor responsible for the green line emission ($R^P_{v1.1-2}$, $R^P_{v2.1}$, $R^P_{v3.1-3}$, $R^P_{v4.0}$).

It should be noted that the G-model processes $R^P_{v1.1-2}$, $R^P_{v2.1}$, $R^P_{v3.1-3}$ and $R^P_{v4.0}$ were replaced by corresponding processes of the H-model which were adopted in the proposed MAC model. All processes of the G-model are also referred to as G-processes.

In addition to the G-, M- and H-processes, complementary processes (C-processes) were proposed to couple O$_2$($^5\Pi$, $A$, $A'$, $c$, $b$, $a$, $X$) with each other and O($^1S$, $^1D$, $^3P$) taking the hypotheses of Huestis (2002) and Slanger et al. (2004b) into account and discussed by Lednyts'kyy and von Savigny (2016) and Lednyts'kyy et al. (2018).

Huestis (2002) suggested that the de-excitation of O$_2$ states with higher energy to O$_2$ states with lower energy only occurs in a cascade that was described by Slanger et al. (2004b) as the integrity of the O$_2$ electronic states' identity. This enables assuming that the O($^1S$) precursor can be represented by one O$_2$ state or a group of O$_2$ states according to the hypothesis of the integrity of the O$_2$ electronic states' identity. In fact, the Barth excitation transfer scheme was formulated with O$_2^*$ considering it as one not identified O$_2$ state or a one group of many not identified O$_2$ states coupled in a cascade of de-excitation reactions is also possible.

The hypothesis of Huestis (2002) was refuted by Slanger et al. (2004b) on the basis of laboratory measurements discussed by Huestis (2002), Slanger et al. (2004b) and Pejaković et al. (2007) stating that energetically nearly resonant intermolecular processes are responsible for conversions of higher to lower excited O$_2$ electronic states according to Slanger and Copeland (2003). Specifically, Slanger et al. (2004b) suggested that the de-excitation of O$_2$ states occurs not in a cascade-like process. They emphasized the presence of a cycle of de-excitation and excitation of O$_2$($^5\Pi$) and the Herzberg O$_2$ states in high vibrational levels. These O$_2$ states transform back and forth into each other through collisions. Finally, the O$_2$($^5\Pi$)–O$_2$($A$, $A'$, $c$)–group is removed converting to very high vibrational levels of O$_2$($b$, $a$, $X$) states. It should be noted that $^5\Pi$ is the electronically excited



$O_2$ state with the higher energy than $O_2$ in the Herzberg states. This, in contrast to the hypothesis of Huestis (2002), makes it more complicated to operate with the $O(^1S)$ precursor as a group of many not identified $O_2$ states.

The C-processes related to the Herzberg states $A^3\Sigma_u^+$ and $A'^3\Delta_u$ (hereafter referred to as $O_2(A, A')$) are not considered in the G-, M- and H-models. These C-processes are related to:

1. the production of $O_2(A)$ ($R_{t1.1-2}$),

     2. the de-excitation of $O_2(A)$ to $O_2(A', c, b)$ ($R_{t2.1-3}$, $R_{t3.1-3}$, $R_{t4.1-3}$),

     3. the Broida-Gaydon band emission ($R_{t5.0}$),

     4. the de-excitation of $O_2(A)$ to $O_2(a, X)$ ($R_{t6.1-3}$, $R_{t7.1-3}$),

     5. the Herzberg I band emission ($R_{t8.0}$, $R_{t9.0}$),

6. the $O(^1S)$ precursor responsible for the green line emission ($R_{t10.1}$, $R_{d9.1}$).

     7. the production of $O_2(A')$ ($R_{d1.1-2}$),

     8. the de-excitation of $O_2(A')$ to $O_2(c, b, a)$ ($R_{d2.1-2}$, $R_{d3.1-2}$, $R_{d4.1-2}$),

     9. the Chamberlain band emission ($R_{d5.0}$, $R_{d6.0}$),

     10. the de-excitation of $O_2(A')$ to $O_2(X)$ ($R_{d7.1-2}$),

11. the Herzberg III band emission ($R_{d8.0}$).

These C-processes are shown here in Table 5, they were considered and discussed by Lednyts′kyy et al. (2018). The corresponding reaction rates are shown in Table 9.

The C-processes related to the G-, M- and H-processes complete the coupling of $O_2(^5\Pi, c, b, a, X)$ with each other and $O(^1S, {}^1D, {}^3P)$, they are related to:

1. the photolysis of $O_2$ and $O_3$ ($R_{s1.(1,4-5)}$, $R_{s2.(1-2,4-6)}$),

     2. the singlet Herzberg state ($R_{c4.0}$, $R_{c5.1-2}$, $R_{c6.0}$, $R_{c7.2}$),

     3. the Atmospheric band emission ($R_{b2.1}$, $R_{b4.1,5-6}$),

     4. the Infrared Atmospheric band emission ($R_{a2.1}$),

     5. the red line emission ($R_{r2.2,4}$, $R_{r1.1-3}$, $R_{r3.0}$),

6. the green line emission ($R_{g1.3}$, $R_{g2.2}$),

     7. three-body recombination and ozone ($R_{x1.1-2}$, $R_{x2.1}$, $R_{x3.1-2}$).



These C-processes are shown here in Tables 6 and 7, they were considered and discussed by Lednyts′kyy and von Savigny (2016). The corresponding reaction rates are shown in Tables 10 … 12.

Unknown or poorly constrained reaction rates of these complementary processes might be compromised by boundary effects if they were measured in a ground-based laboratory. Therefore, an appropriate photochemical model including many chemical species obtained on the basis of multiple emissions measured *in-situ* in the Earth's atmosphere may be a valuable complement to laboratory experiments conducted on the Earth's surface. In fact, unknown or poorly constrained reaction rates were tuned according to the verification and validation procedures discussed in Section 3.4 and applied on the basis of the ETON *in-situ* measurements. The advantage of the ETON campaign compared to another rocket campaigns is that multiple emissions and $[O(^3P)]$ were measured almost simultaneously. This enables comparing the *in-situ* and retrieved $[O(^3P)]$ using each particular emission profile described in Section 2.

Figure 1 shows processes coupling $O_2(^5\Pi, A, A', c, b, a, X)$ and $O(^1S, {}^1D, {}^3P)$ with each other, and Fig. 2 shows processes coupling $O_2(^5\Pi, c, b, a, X)$ and $O(^1S, {}^1D, {}^3P)$ with each other.

Considering the energy required for a spin flip in transitions among the triplet $O_2(A, A', X)$ and singlet $O_2(c, b, a)$ states it can be concluded that transitions from the $O_2(A, A')$ states to the $O_2(X)$ state are more probable than spin forbidden transitions from the $O_2(A, A')$ states to the $O_2(c, b, a)$ states. Therefore, at least two versions of the MAC model can be implemented on the basis of the ETON measurements. The first one involves $O_2(A)$ and $O_2(A')$, and the second one excludes them from the MAC model.

### 3.4 Verification and validation of calculations carried out with the MAC model

The input parameters of the MAC model are described in Section 2, they are: VER profiles retrieved on the basis of *in-situ* measurements during the ETON rocket campaign (Greer et al., 1986) as well as profiles of temperature (T), $[N_2]$ and $[O_2]$ obtained using the semi-empirical model NRLMSISE-00. Among the mentioned VER profiles are: VER$\{O_2(A-X)\}$ (Herzberg I band, HzI), VER$\{O_2(A'-a)\}$ (Chamberlain band, Cha), VER$\{O_2(b-X)\}$ (Atmospheric band, Atm), VER$\{O_2(a-X)\}$ (Infrared Atmospheric band, IRAtm) and VER$\{O(^1S-{}^1D)\}$ (green line, GrL). These VER profiles were retrieved on the basis of the raw integrated data (Greer et al., 1986) that is marked with a character R, e.g. R-VER$\{O_2(A-X)\}$.

Concentrations of various chemical species were retrieved using sequentially applied continuity equations in the steady state, i.e. polynomial equations of the second or the third order. An overview of all retrieval steps of the MAC model is provided in Appendix A devoted to the description of all algorithmic steps used in calculations with the MAC model, see also Table 8 for a short overview. The input-VER profiles shown in Table 8 correspond to $O_2$ transitions shown in Table 1. In fact, all reactions relevant for the particular chemical species were used in the retrievals, and the retrieved concentration profiles are marked with a character R, e.g. R-$[O_2(A)]$. Additionally, concentrations of the same chemical species were evaluated dividing the R-VER profiles, which correspond to the particular chemical species, by the respective transition probability. The evaluated concentration profiles are marked with a character E, e.g. E-$[O_2(A)]$. As for the evaluated VER profiles, which are marked with a character E as E-VER profiles (e.g. E-VER$\{O_2(A-X)\}$), they are obtained by dividing the retrieved concentrations of the respective same chemical species by the respective transition probability.



**Table 5.** Processes of the MAC model, continued by processes shown in Tables 6 and 7.

| $R_\#$ | Odd oxygen processes related to $O_2(A)$ and $O_2(A')$ |
|---|---|
| $R_{t1.1-2}$ | $O(^3P) + O(^3P) + \{N_2, O_2\} \xrightarrow{\theta^{Pt}_{N2}, \theta^{Pt}_{O2}} O_2(A) + \{N_2, O_2\}$ |
| $R_{t2.1-3}$ | $O_2(A) + \{O(^3P), N_2, O_2\} \xrightarrow{\theta^{td}_{3P}, \theta^{td}_{N2}, \theta^{td}_{O2}} O_2(A') + \{O(^3P), N_2, O_2\}$ |
| $R_{t3.1-3}$ | $O_2(A) + \{O(^3P), N_2, O_2\} \xrightarrow{\theta^{tc}_{3P}, \theta^{tc}_{N2}, \theta^{tc}_{O2}} O_2(c) + \{O(^3P), N_2, O_2\}$ |
| $R_{t4.1-3}$ | $O_2(A) + \{O(^3P), N_2, O_2\} \xrightarrow{\theta^{tb}_{3P}, \theta^{tb}_{N2}, \theta^{tb}_{O2}} O_2(b) + \{O(^3P), N_2, O_2\}$ |
| $R_{t5.0}$ | $O_2(A) \xrightarrow{\theta^A_{BG}} O_2(b) + h\nu \text{(Broida-Gaydon system)}$ |
| $R_{t6.1-3}$ | $O_2(A) + \{O(^3P), N_2, O_2\} \xrightarrow{\theta^{ta}_{3P}, \theta^{ta}_{N2}, \theta^{ta}_{O2}} O_2(a) + \{O(^3P), N_2, O_2\}$ |
| $R_{t7.1-3}$ | $O_2(A) + \{O(^3P), N_2, O_2\} \xrightarrow{\theta^{tx}_{3P}, \theta^{tx}_{N2}, \theta^{tx}_{O2}} O_2 + \{O(^3P), N_2, O_2\}$ |
| $R_{t8.0}$ | $O_2(A) \xrightarrow{\theta^A_{320n}} O_2 + h\nu(\lambda = 320\,\text{nm})$ |
| $R_{t9.0}$ | $O_2(A) \xrightarrow{\theta^A_{HI}} O_2 + h\nu \text{(Herzberg I band)}$ |
| $R_{t10.1}$ | $O_2(A) + O(^3P) \xrightarrow{\theta^{tx}_{IS}} O_2 + O(^1S)$ |
| $R_{d1.1-2}$ | $O(^3P) + O(^3P) + \{N_2, O_2\} \xrightarrow{\delta^{Pd}_{N2}, \delta^{Pd}_{O2}} O_2(A') + \{N_2, O_2\}$ |
| $R_{d2.1-2}$ | $O_2(A') + \{O(^3P), O_2\} \xrightarrow{\delta^{dc}_{3P}, \delta^{dc}_{O2}} O_2(c) + \{O(^3P), O_2\}$ |
| $R_{d3.1-2}$ | $O_2(A') + \{O(^3P), O_2\} \xrightarrow{\delta^{db}_{3P}, \delta^{db}_{O2}} O_2(b) + \{O(^3P), O_2\}$ |
| $R_{d4.1-2}$ | $O_2(A') + \{O(^3P), O_2\} \xrightarrow{\delta^{da}_{3P}, \delta^{da}_{O2}} O_2(a) + \{O(^3P), O_2\}$ |
| $R_{d5.0}$ | $O_2(A') \xrightarrow{\delta^A_{370n}} O_2(a) + h\nu(\lambda = 370\,\text{nm})$ |
| $R_{d6.0}$ | $O_2(A') \xrightarrow{\delta^A_{Cha}} O_2(a) + h\nu \text{(Chamberlain band)}$ |
| $R_{d7.1-2}$ | $O_2(A') + \{O(^3P), O_2\} \xrightarrow{\delta^{dx}_{3P}, \delta^{dx}_{O2}} O_2 + \{O(^3P), O_2\}$ |
| $R_{d8.0}$ | $O_2(A') \xrightarrow{\delta^A_{HIII}} O_2 + h\nu \text{(Herzberg III band)}$ |
| $R_{d9.1}$ | $O_2(A') + O(^3P) \xrightarrow{\delta^{dx}_{IS}} O_2 + O(^1S)$ |

The results of calculations carried out using the MAC model are verified by a visual comparison of retrieved and evaluated profiles, i.e. the respective emission and concentration values. Note that the prior step 1 shown in Table 8 and briefly described in Section A1 is omitted for the ETON campaign because such input parameters as $[O_3]$ and $[H]$ are not known *a-priori*. Instead, the short list of the input parameters required to run the MAC model is applied: T, $[N_2]$, $[O_2]$ from the NRLMSISE-00 model

5 and VER profiles from the ETON campaign. For instance, the quadratic continuity equation is solved to retrieve R-$[O_2(A)]$ on the basis of R-VER$\{O_2(A-X)\}$ by using all relevant processes of the MAC model. This retrieval step is shown as the step 2.1 in Table 8 and the step 2.1 described in Section A2.1 in Appendix A. Then, the verification of calculations at the step 2.1 is carried out comparing R-VER$\{O_2(A-X)\}$ with E-VER$\{O_2(A-X)\}$ and R-$[O_2(A)]$ with E-$[O_2(A)]$. The cubic equation is solved at the step 2.2 on the basis of T, $[N_2]$, $[O_2]$, R-VER$\{O_2(A'-a)\}$ and R-$[O_2(A)]$. Then, the verification of calculations

10 at the step 2.2 is carried out comparing R-VER$\{O_2(A'-a)\}$ with E-VER$\{O_2(A'-a)\}$ and R-$[O_2(A')]$ with E-$[O_2(A')]$.



**Table 6.** Processes shown here comprise the MAC model together with processes shown above in Table 5 and processes shown below in Table 7.

| R$_\#$ | Odd oxygen processes related to O$_2(c)$, O$_2(b)$ and O$_2(a)$ |
|---|---|
| $R_{c1.1-2}$ | $O(^3P) + O(^3P) + \{N_2, O_2\} \xrightarrow{\varsigma_{N2}^{Pc}, \varsigma_{O2}^{Pc}} O_2(c) + \{N_2, O_2\}$ |
| $R_{c2.1}$ | $O_2(c) + O(^3P) \xrightarrow{\varsigma_{1S}^{cx}} O_2 + O(^1S)$ |
| $R_{c3.1-2}$ | $O_2(c) + \{O(^3P), O_2\} \xrightarrow{\varsigma_{3P}^{cb}, \varsigma_{O2}^{cb}} O_2(b) + \{O(^3P), O_2\}$ |
| $R_{c4.0}$ | $O_2(c) \xrightarrow{\varsigma_{cbK}^{A}} O_2(b) + h\nu (\text{New system from Keck I/II})$ |
| $R_{c5.1-2}$ | $O_2(c) + \{O(^3P), O_2\} \xrightarrow{\varsigma_{3P}^{ca}, \varsigma_{O2}^{ca}} O_2(a) + \{O(^3P), O_2\}$ |
| $R_{c6.0}$ | $O_2(c) \xrightarrow{\varsigma_{RJ}^{A}} O_2(a) + h\nu (\text{Richards-Johnson system})$ |
| $R_{c7.1-2}$ | $O_2(c) + \{O(^3P), O_2\} \xrightarrow{\varsigma_{3P}^{cx}, \varsigma_{O2}^{cx}} O_2 + \{O(^3P), O_2\}$ |
| $R_{c8.0}$ | $O_2(c) \xrightarrow{\varsigma_{HII}^{A}} O_2 + h\nu (\text{Herzberg II band})$ |
| $R_{b1.1-2}$ | $O(^3P) + O(^3P) + \{N_2, O_2\} \xrightarrow{\beta_{N2}^{Pb}, \beta_{O2}^{Pb}} O_2(b) + \{N_2, O_2\}$ |
| $R_{b2.1-5}$ | $O_2(b) + \{O_3, O, N_2, O_2, CO_2\} \xrightarrow{\beta_{O3}^{ba}, \beta_{3P}^{ba}, \beta_{N2}^{ba}, \beta_{O2}^{ba}, \beta_{C2}^{ba}} O_2(a) + \{O_3, O, N_2, O_2, CO_2\}$ |
| $R_{b3.0}$ | $O_2(b) \xrightarrow{\beta_{Nox}^{A}} O_2(a) + h\nu (\text{Noxon transition})$ |
| $R_{b4.1-6}$ | $O_2(b) + \{O_3, O, N_2, O_2, CO_2, O_3\} \xrightarrow{\beta_{O3}^{bx}, \beta_{3P}^{bx}, \beta_{N2}^{bx}, \beta_{O2}^{bx}, \beta_{C2}^{bx}, \beta_{O3}^{x3}} O_2 + \{O + O_2, O, N_2, O_2, CO_2, O_3\}$ |
| $R_{b5.0}$ | $O_2(b) \xrightarrow{\beta_{762}^{A}} O_2 + h\nu (\lambda = 762\,\text{nm})$ |
| $R_{b6.0}$ | $O_2(b) \xrightarrow{\beta_{Atm}^{A}} O_2 + h\nu (\text{Atmospheric band})$ |
| $R_{a1.1-2}$ | $O(^3P) + O(^3P) + \{N_2, O_2\} \xrightarrow{\alpha_{N2}^{Pa}, \alpha_{O2}^{Pa}} O_2(a) + \{N_2, O_2\}$ |
| $R_{a2.1-4}$ | $O_2(a) + \{O_3, O, N_2, O_2\} \xrightarrow{\alpha_{O3}^{ax}, \alpha_{3P}^{ax}, \alpha_{N2}^{ax}, \alpha_{O2}^{ax}} O_2 + \{O + O_2, O, N_2, O_2\}$ |
| $R_{a3.0}$ | $O_2(a) \xrightarrow{\alpha_{1u27}^{A}} O_2 + h\nu (\lambda = 1.27\,\mu\text{m})$ |
| $R_{a4.0}$ | $O_2(a) \xrightarrow{\alpha_{IRA}^{A}} O_2 + h\nu (\text{IR Atmospheric band})$ |

Note that values of the *in-situ* R-VER$\{O(^1S - {}^1D)\}$ profile are less than zero randomly below 92 km due to the measurement noise. Therefore, the *in-situ* R-VER$\{O(^1S - {}^1D)\}$ profile is approximated by the asymmetrical Gaussian distribution described by Semenov (1997) and Khomich et al. (2008) to obtain the shown A-VER$\{O(^1S - {}^1D)\}$ profile and to retrieve the corresponding $[O(^1S)]$ profile.

5    The retrieved and evaluated VER profiles indicated by the dashed lines and the symbols, respectively, and shown on the left in Fig. 3 are compared with each other by sight to verify calculations carried out with the MAC model involving O$_2(A)$ and O$_2(A')$. The retrieved and evaluated VER profiles belonging to each pair regarding the considered excited O$_2$ state seem to be in perfect agreement with each other by sight. Next, the retrieved and evaluated concentration profiles shown on the right in Fig. 3 by the dashed lines and the symbols, respectively, are also compared with each other for each retrieval step; these

10   profiles also seem to be in perfect agreement with each other by sight. The comparison of the retrieved and evaluated products



**Table 7.** Processes shown here comprise the MAC model together with processes shown above in Tables 5 and 6.

| R$_\#$ | Odd oxygen processes related to O($^1S$) and O($^1D$) |
|---|---|
| $R_{g1.1-3}$ | O($^1S$) + {O($^3P$), O$_2$, O$_3$} $\xrightarrow{\gamma_{1D}^{SP}, \gamma_{O2}^{SP}, \gamma_{O3}^{SP}}$ {2 O($^1D$), O($^3P$) + O$_2$, 2 O$_2$} |
| $R_{g2.1-2}$ | O($^1S$) + {N$_2$, O$_2$($a$)} $\xrightarrow{\gamma_{N2}^{SP}, \gamma_{Oa}^{SP}}$ O($^3P$) + {N$_2$, O$_2$($a$)} |
| $R_{g3.0}$ | O($^1S$) $\xrightarrow{\gamma_{557n7}^{A}}$ O($^1D$) + $h\nu$ ($\lambda = 557.7$ nm) |
| $R_{g4.0}$ | O($^1S$) $\xrightarrow{\gamma_{1S3Pe}^{A}}$ O($^3P$) + $h\nu$ |
| $R_{r1.1-3}$ | O($^1D$) + {O($^3P$), O$_3$, O$_3$} $\xrightarrow{\rho_{3P}^{DP}, \rho_{2P}^{DP}, \rho_{O2}^{DP}}$ {2 O($^3P$), 2 O($^3P$) + O$_2$, 2 O$_2$} |
| $R_{r2.1-4}$ | O($^1D$) + {N$_2$, O$_2$, O$_2$, CO$_2$} $\xrightarrow{\rho_{N2}^{DP}, \rho_{Oa}^{DP}, \rho_{Ob}^{DP}, \rho_{C2}^{DP}}$ O($^3P$) + {N$_2$, O$_2$($a$), O$_2$($b$), CO$_2$} |
| $R_{r3.0}$ | O($^1D$) $\xrightarrow{\rho_{1D3Pe}^{A}}$ O($^3P$) + $h\nu$ |

| R$_\#$ | Odd oxygen processes related to the loss of atomic oxygen |
|---|---|
| $R_{x1.1-2}$ | O($^3P$) + O($^3P$) + {N$_2$, O$_2$} $\xrightarrow{\chi_{N2}^{Px}, \chi_{O2}^{Px}}$ O$_2$ + {N$_2$, O$_2$} |

| R$_\#$ | Odd oxygen processes related to catalytic ozone destruction and photolysis |
|---|---|
| $R_{x2.1}$ | O($^3P$) + O$_3$ $\xrightarrow{\chi_{O2}^{3P}}$ 2 O$_2$ |
| $R_{x3.1-2}$ | O$_2$ + O($^3P$) + {N$_2$, O$_2$} $\xrightarrow{\chi_{N2}^{P3}, \chi_{O2}^{P3}}$ O$_3$ + {N$_2$, O$_2$} |
| $R_{s1.1-5}$ | O$_2$ + $h\nu$ $\xrightarrow{\sigma_{PS}^{UV}, \sigma_{PD}^{LA}, \sigma_{PD}^{Sc}, \sigma_{PP}^{Sb}, \sigma_{PP}^{Hc}}$ O($^3P$) + {O($^1S$), O($^1D$), O($^1D$), O($^3P$), O($^3P$)} |
| $R_{s2.1-6}$ | O$_3$ + $h\nu$ $\xrightarrow{\sigma_{aS}^{UV}, \sigma_{PP}^{Ha}, \sigma_{aD}^{Hu}, \sigma_{xD}^{Hu}, \sigma_{aP}^{Ch}, \sigma_{xP}^{Ch}}$ {O($^1S$) + O$_2$($a$), 3 O, O($^1D$) + O$_2$($a$), O($^1D$) + O$_2$, O + O$_2$($a$), O + O$_2$} |
| $R_{s3.1}$ | O$_2$ + $h\nu$ ($\lambda = 762$ nm) $\xrightarrow{\sigma_{b1}^{O2}}$ O$_2$($b$) |

| R$_\#$ | Odd hydrogen processes |
|---|---|
| $R_{h1.1}$ | H + O$_3$ $\xrightarrow{\eta_{OH}^{H}}$ OH$^*$ + O$_2$ |
| $R_{h2.1}$ | OH$^*$ + O($^3P$) $\xrightarrow{\eta_{OH}^{3P}}$ H + O$_2$ |
| $R_{h3.1}$ | OH$^*$ + O$_3$ $\xrightarrow{\eta_{HO2}^{OH}}$ HO$_2$ + O$_2$ |
| $R_{h4.1}$ | HO$_2$ + O($^3P$) $\xrightarrow{\eta_{HO2}^{3P}}$ OH$^*$ + 2 O$_2$ |
| $R_{h5.1-2}$ | H + O$_2$ + {N$_2$, O$_2$} $\xrightarrow{\eta_{N2}^{H}, \eta_{O2}^{H}}$ HO$_2$ + {N$_2$, O$_2$} |
| $R_{h6.1-3}$ | H + HO$_2$ $\xrightarrow{\eta_{OH}^{HO2}, \eta_{H2}^{HO2}, \eta_{H2O}^{HO2}}$ {OH$^*$ + OH$^*$, H$_2$ + O$_2$, O($^3P$) + H$_2$O} |

(VER or concentration profiles) enables concluding that all calculations carried out using the MAC model are consistent with each other and coherent with measurements.

Before we discuss results of the [O($^3P$)] retrievals obtained with the proposed MAC model, a short overview of the previously used photochemical models is given to estimate our current situation and to argue whether the proposed MAC model


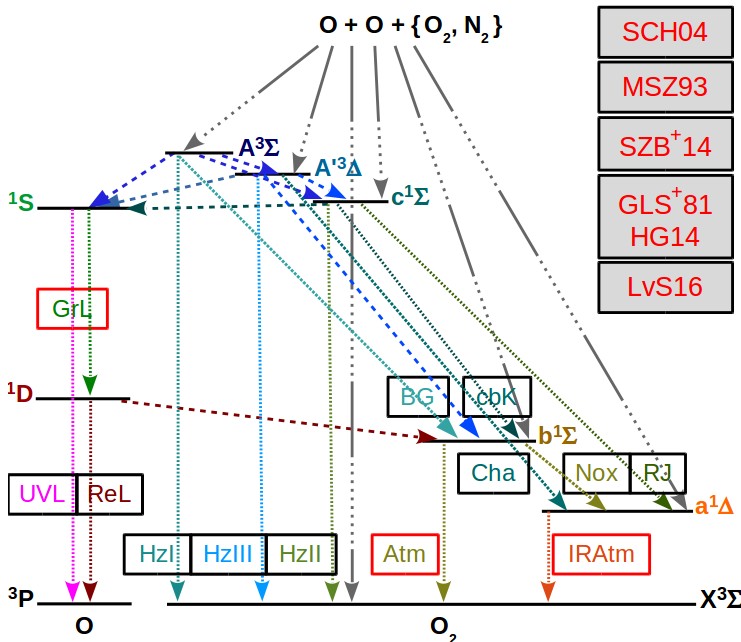

**Figure 1.** $O_2$ and O term diagrams showing processes of the MAC model comprised of processes considered in the G-, M- and H-models combined and extended with complementary C-processes completing the MAC model. C-processes are proposed to couple states of $O_2(^5\Pi, A, A', c, b, a, X)$ and $O(^1S, ^1D, ^3P)$ with each other according to the hypothesis of Slanger et al. (2004b) (SCH04) stating that the Herzberg states are in constant collisional communication with the higher excited $^5\Pi$ electronic state. All considered processes of the MAC model are provided in Tables 5, 6 and 7. Three-body recombination (association) reactions are indicated by the gray 10 dots–1 dash line and result in $O_2^*$ and $O_2^{**}$ due to reactions, rate values of which are $\beta\kappa_1$ and $\alpha\kappa_1$, respectively, (McDade et al., 1986). Radiative losses accompanied with quenching processes are indicated by an abbreviation near the fine dashed line, see Table 1 for abbreviations. Radiative losses only are also indicated by an abbreviation, but near the violet 2 dots–3 dashes line. Quenching processes only are indicated by the dashed lines. References shown in the legend as well as the representation of lines and abbreviations regarding various chemical processes are the same as those used in the mentioned above figures. The $O_2(A, A', c)$ Herzberg states are all implemented as possible $O(^1S)$ precursors because their energy in various vibrational levels exceeds the 4.19 eV excitation energy difference with respect to the triplet $O_2(X)$ ground state.

is needed. The published photochemical models based on processes provided in Table 2 resulted in the following continuity equations discussed here with respect to $[O(^3P)]$:

1. the well-known quadratic equation of McDade et al. (1986) (MMG$^+$86) was applied to the Atmospheric band emissions at 762.2 nm (see Sections 1 and 3.1),

5    2. the well-known cubic equation (2) was applied to the green line emissions at 557.7 nm (see Section 3.1) and



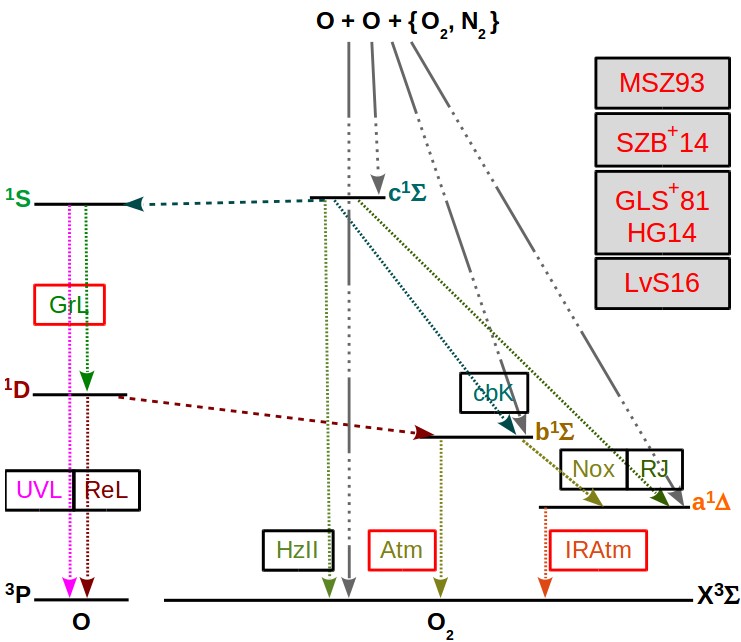

**Figure 2.** Similar to Fig. 1, but for processes excluding the triplet $O_2(A)$ and $O_2(A')$ Herzberg states from the MAC model. All considered processes of the MAC model indicated in the $O_2$ and O term diagrams are provided in Tables 6 and 7. The following conclusions drawn by Slanger et al. (2004b) and Krasnopolsky (2011) help to interpret processes indicated here in the $O_2$ and O term diagrams: (1) the $O_2(A, A')$ and $O_2(X)$ $(X^3\Sigma_g^-)$ are triplet states, which are strongly coupled with each other; (2) transitions among the singlet $O_2(c, b, a)$ states $(c^1\Sigma_u^-,$ $b^1\Sigma_g^+, a^1\Delta_g)$ and the triplet $O_2(A, A', X)$ states are less probable because they require a spin flip; (3) the $O_2(c)$ and $O_2(b, a)$ states seem to be rather weakly coupled with each other presumably because of Frank-Condon factors. This enabled neglecting the $O_2(A, A')$ states in the MAC model indicsted in Fig. 2.

3. the extended cubic equation proposed by Gobbi et al. (1992), see Eq. (3), was applied to the green line emission at 557.7 nm with empirical coefficients of Lednyts′kyy et al. (2015) (LSE$^+$15) and solved using the analytical method of Semenov (1997) modified by Lednyts′kyy et al. (2015).

These three continuity equations are applied, and the retrieved $[O(^3P)]$ profiles are shown on the left in Fig. 4. The peak
5  $[O(^3P)]$ profile values retrieved according to the well-known quadratic and cubic equations are lower, but those of the extended cubic equation are higher than the peak values of the *in-situ* ETON $[O(^3P)]$ profile. The $[O(^3P)]$ profile values retrieved according to the well-known and extended cubic equations are almost equidistant with respect to the *in-situ* $[O(^3P)]$ profile values, and can be considered as two profiles of extrema values. One could assume that the arithmetical averaging of the extrema $[O(^3P)]$ profile values might be appropriate to finalize the $[O(^3P)]$ retrievals, see the violet crosses on the left in both figures.
10  Indeed, the averaged peak $[O(^3P)]$ profile values are almost equal to those of the *in-situ* $[O(^3P)]$ profile. However, now we do not see any deeper significance in this finding. Empirical coefficients were derived for these previously used photochemical





**Table 8.** Overview of the calculation steps carried out using the MAC model. The first column shows the step number. Input-concentrations shown in the third column were retrieved at one of the previous steps and are required together with profiles of input-VER and the other MAC input parameters at the current retrieval step. The other MAC input parameters should be at least comprised of temperature (T), $O_2$ and $N_2$ that can be simulated using the NRLMSISE-00 model. If only these MAC input parameters are available, then the prior step 1 described in Section A1 is omitted. Nevertheless, if $[O_3]$ and $[H]$ were also available among the other MAC input parameters, then $[O(^1S)]$, $[O(^1D)]$, $[OH^*]$ and $[HO_2]$ would be calculated at the prior step 1 and also used as MAC input parameters at the following steps.

| Step # | Input-VER | Input-concentration | Output-concentration |
|---|---|---|---|
| 1 | – | – | – |
| 2.1 | VER$\{O_2(A-X)\}$ | – | $[O(^3P)]$, $[O_2(A)]$ |
| 2.2 | VER$\{O_2(A'-a)\}$ | $[O_2(A)]$ | $[O(^3P)]$, $[O_2(A')]$ |
| 2.3 | VER$\{O_2(b-X)\}$ | $[O_2(A)]$, $[O_2(A')]$ | $[O(^3P)]$, $[O_2(b)]$ |
| 3.1 | – | $[O(^3P)]$, $[O_2(A)]$, $[O_2(A')]$, $[O_2(b)]$ | $[O_2(c)]$ |
| 3.2 | VER$\{O_2(a-X)\}$ | $[O_2(A)]$, $[O_2(A')]$, $[O_2(c)]$, $[O_2(b)]$ | $[O(^3P)]$, $[O_2(a)]$ |
| 4.1 | VER$\{O(^1S-^1D)\}$ | $[O_2(A)]$, $[O_2(A')]$, $[O_2(c)]$, $[O_2(b)]$, $[O_2(a)]$ | $[O(^3P)]$, $[O(^1S)]$ |
| 5.1 | – | $[O_2(A)]$, $[O_2(A')]$, $[O_2(c)]$, $[O_2(b)]$, $[O_2(a)]$, | $[O(^3P)]$ |
| 5.1 | – | $[O(^1S)]$ | $[O(^3P)]$ |

models phenomenologically, i.e. in relation to rates of reactions in which a not identified $O_2^*$ is involved. Therefore, $[O(^3P)]$ retrievals on a new photochemical basis are required. Note processes of the previously used photochemical models were also used to propose the MAC model, which is applied as follows.

$[O(^3P)]$ profiles retrieved using the MAC model involving $O_2(A)$ and $O_2(A')$ are shown on the right in Fig. 4. The *in-situ*
5    $[O(^3P)]$ profile is compared with the $[O(^3P)]$ profiles obtained at the retrieval steps provided in Table 8. The retrieved profiles are indicated in the same color used to show them in the legend. The peak values of the $[O(^3P)]$ profiles retrieved directly on the basis of VER$\{O_2(A-X)\}$, VER$\{O_2(A'-a)\}$, VER$\{O_2(b-X)\}$ and VER$\{O_2(a-X)\}$ are lower than those of the *in-situ* ETON $[O(^3P)]$ profile, but the peak values of the $[O(^3P)]$ profile retrieved at the pre-last step 4.1 on the basis of VER$\{O(^1S-^1D)\}$ are higher. The peak magnitude and altitude values as well as the shape of the $[O(^3P)]$ profile retrieved
10    at the last step 5.1 on the basis of all chemical species are approximately the same comparing to those of the *in-situ* ETON $[O(^3P)]$ profile.

In the following, the retrieval results obtained with the MAC model excluding $O_2(A)$ and $O_2(A')$ are shown in Fig. 5 and discussed in the comparison to those obtained with the MAC model involving $O_2(A)$ and $O_2(A')$ and shown in Figs. 3 and 4.

Profiles of VER and $[O(^3P)]$ obtained at the retrieval steps 2.3, 3.1, 3.2, 4.1 and 5.1 are shown on the left and right of Fig. 5,
15    respectively. In fact, values of E-VER$\{O_2(A-X)\}$, E-VER$\{O_2(A'-a)\}$, R-$[O_2(A)]$ and R-$[O_2(A')]$ are equal zero, whereas E-$[O_2(A)]$ and E-$[O_2(A')]$ can not be shown because of the division by transition probabilities set to zero at the retrieval steps 2.1 and 2.2 for Fig. 5.





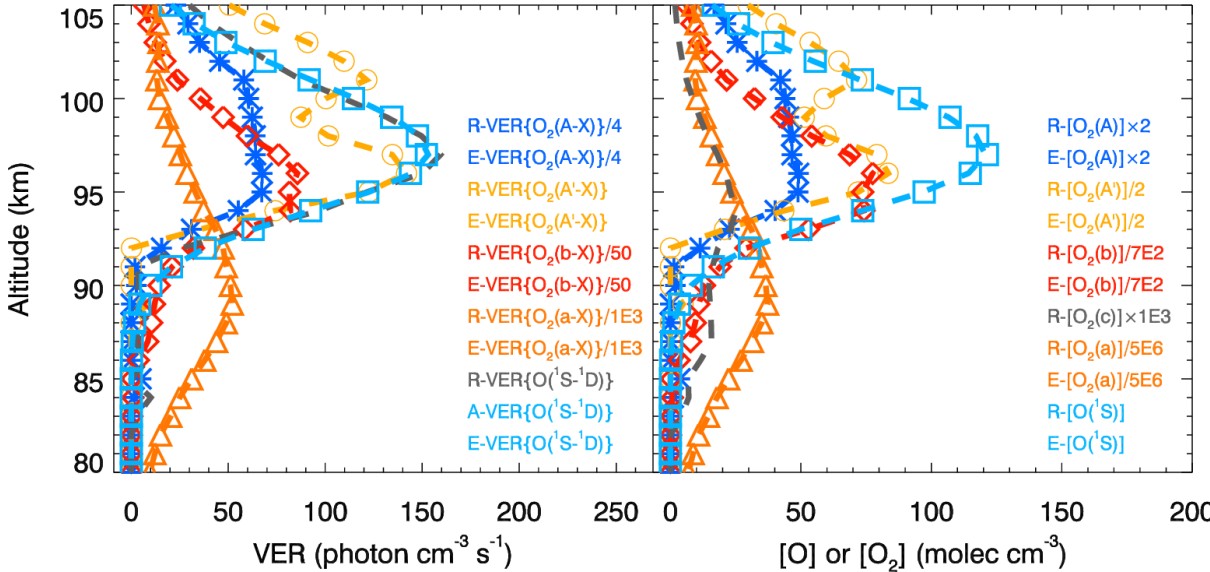

**Figure 3.** The retrieved VER (R-VER) profiles obtained during the ETON campaign (see Section 2) and the evaluated VER (E-VER) profiles obtained using the MAC model involving $O_2(A)$ and $O_2(A')$ are shown on the left by the dashed lines and symbols. Calculations carried out using the MAC model involving $O_2(A)$ and $O_2(A')$ are verified visually comparing the R-VER and E-VER profiles. Concentrations of various chemical species were retrieved on the basis of the corresponding R-VER profiles and all relevant processes of the MAC model; these concentrations are marked with a character R and shown shown on the right by the dashed lines. The respective transition probabilities are only used to evaluate concentrations marked with a character E as well as E-VER profiles. Again, the evaluated concentrations are shown with the use of symbols as it as done for E-VER profiles. Two corresponding profiles (R-VER and E-VER as well as of the retrieved and evaluated concentrations) seem to be in perfect agreement with each other by sight. This implies that all calculations carried out with the MAC model are consistent with each other and the results are coherent with measurements. The abbreviations indicating emissions are explained in Table 1, and the sequence of the retrieval steps is provided in Table 8. Values of temperature, $[N_2]$ and $[O_2]$ were obtained by using the NRLMSISE-00 model (see Section 2) for the time and place of the P229H rocket.

Comparing VER and $[O(^3P)]$ profiles shown on the left in Figs. 3 and 5 with each other, it can be concluded that all calculations carried out using the MAC model excluding or involving $O_2(A)$ and $O_2(A')$ are all consistent with each other and coherent with measurements. As far as R-VER$\{O(^1S - {}^1D)\}$ with E-VER$\{O(^1S - {}^1D)\}$ shown in these figures seem to be in perfect agreement with each other, we can argue about the $O(^1S)$ production implemented via different pathways indicated in Figs. 1 and 2. In fact, our suggestions about the origin of the $O(^1S - {}^1D)$ green line emission are also backed up by the comparison of various $[O(^3P)]$ shown on the right in Figs. 4 and 5. Specifically, $O_2(c)$ can be considered the major $O(^1S)$ precursor because the contribution of processes involving $O_2(A)$ and $O_2(A')$ to the $O(^1S)$ production is negligible.

The $[O(^3P)]$ profile values retrieved at the step 3.2 on the basis of VER$\{O_2(a-X)\}$ (Infrared Atmospheric band, IRAtm) are variable, and variabilities are higher than those of the *in-situ* ETON $[O(^3P)]$ profile at altitudes higher than $102\,\mathrm{km}$ and lower than $95\,\mathrm{km}$, see Figs. 4 and 5. $[O(^3P)]$ profile values retrieved at the other steps are in good agreement with those of the *in-situ*





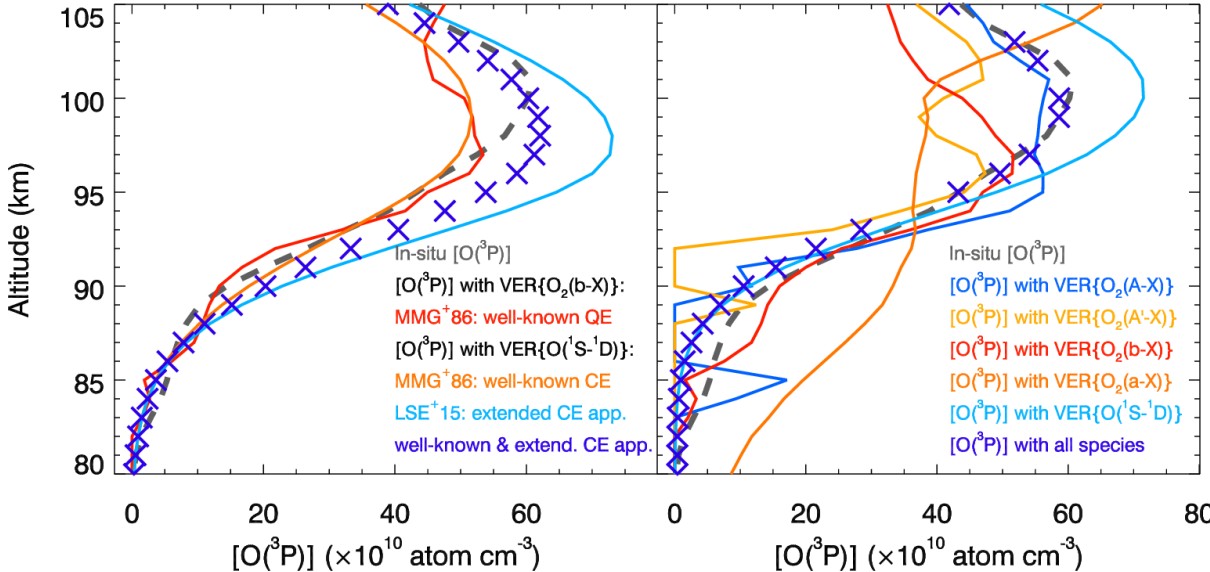

**Figure 4.** The *in-situ* and retrieved $[O(^3P)]$ profiles are shown and compared with each other. The *in-situ* $[O(^3P)]$ profile obtained during the ETON campaign (see Section 2) is shown with the dashed gray line to validate $[O(^3P)]$ retrievals. The well-known quadratic equation (QE) and the well-known cubic equation (CE) of McDade et al. (1986) (MMG$^+$86) as well as the extended CE of Lednyts′kyy et al. (2015) (LSE$^+$15) were applied to retrieve $[O(^3P)]$ profiles shown on the left. $[O(^3P)]$ profiles retrieved according to the cubic equations seem to represent two profiles of extreme values with respect to the *in-situ* $[O(^3P)]$ profile. Therefore, they were arithmetically averaged (see the violet crosses on the left in this figure), and seem to be in good agreement with the *in-situ* $[O(^3P)]$ profile values. This was done to estimate the efficiency of the known photochemical models, but we do not ascribe any deeper significance to this finding. Empirical coefficients were introduced in both cubic equations phenomenologically that stimulated to propose the MAC model. The MAC model involving $O_2(A)$ and $O_2(A')$, see Section 3.3, was applied at the retrieval steps provided in Table 8 and applied consequently to retrieve $[O(^3P)]$ profiles shown on the right by the solid colored lines similar to Fig. 3. Although the retrieval steps 2.1, 2.2, 2.3 and 3.2 applied on the basis of some ETON VER profiles result in lower $[O(^3P)]$ values compared to the *in-situ* ones, the retrieval step 4.1 applied on the basis of VER$\{O(^1S - ^1D)\}$ results in higher values. The last retrieval step 5.1 applied on the basis of concentrations of all chemical species retrieved at the previous steps results in $[O(^3P)]$ values being in good agreement with the *in-situ* values.

ETON $[O(^3P)]$ profile, but $[O(^3P)]$ profile values retrieved at the step 3.2 are in disagreement with all $[O(^3P)]$ profile values mentioned here. Llewellyn and Solheim (1978) analyzed emissions in the IRAtm and Meinel bands and proposed the rate of the reaction $OH(\nu' \geq 1) + O(^3P) \rightarrow H + O_2(a)$, which they suggested to implement in a photochemical model to retrieve $[O(^3P)]$. It should be mentioned that the vibrational population of $OH(\nu')$ has to be known in order to consider the reaction

5   $R_{h2.1}$ shown in Table 7 ($OH^* + O(^3P) \xrightarrow{\eta_{OH}^{3P}} H + O_2$, where $OH^*$ describes the hydroxyl radical in all possible levels $\nu'$) in the MAC model. Wayne (1994) presented an excellent overview of reactions involving $O_2(a)$, and assumed that the reaction emphasized by Llewellyn and Solheim (1978) only produces about one-half of the VER$\{O_2(a-X)\}$ intensity needed. Wayne (1994) suggested that the reaction $OH(\nu' \geq 3) + O_2 \rightarrow OH + O_2(a)$ can be neglected due to its negligible contribution that was




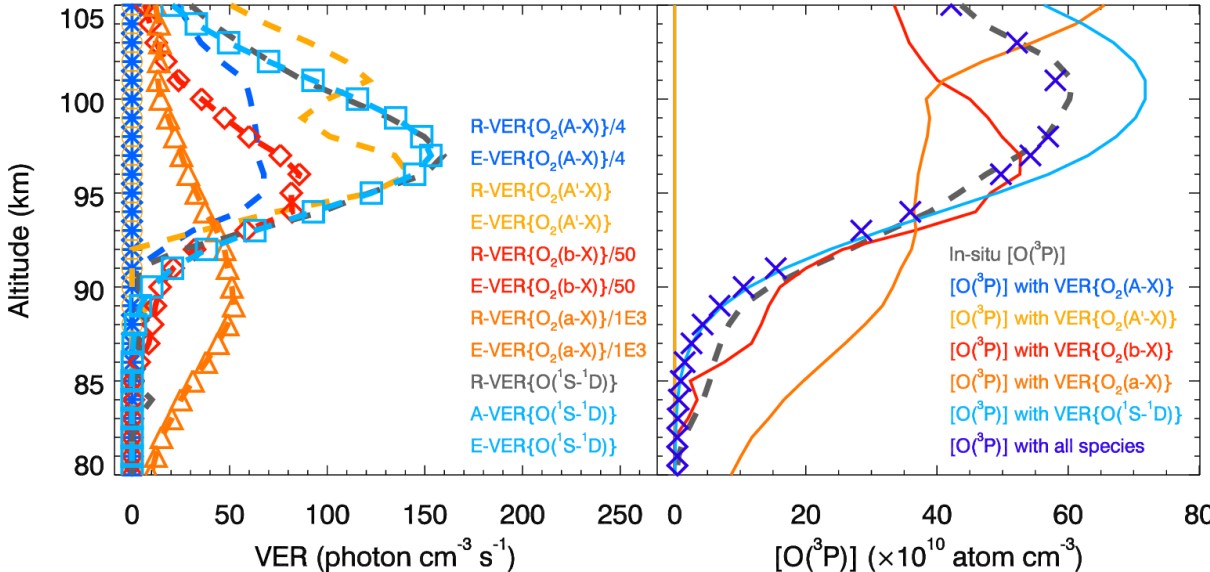

**Figure 5.** Similar to Figs. 3 and 4, but showing results obtained with the MAC model excluding $O_2(A)$ and $O_2(A')$. The first two retrievals steps 2.1 and 2.2 are not carried out, because now the $VER\{O_2(A-X)\}$ and $VER\{O_2(A'-a)\}$ profiles are not considered in the MAC calculations. The *in-situ* and retrieved VER and $[O(^3P)]$ profiles obtained at the retrieval steps 2.3, 3.2, 4.1 and 5.1 agree with each other by sight and with those shown in Fig. 4, and the MAC calculations are concluded to be verified and validated. The comparison of products related to $VER\{O(^1S-{}^1D)\}$ indicated by the cyan color and shown in this figure as well as in Fig. 3 enables concluding that the contribution of $O_2(A)$ and $O_2(A')$ to the $O(^1S)$ production is negligible. Therefore, $O_2(c)$ can be considered the major $O(^1S)$ precursor. It follows also that the triplet Herzberg states ($A^3\Sigma_u^+$, $A'^3\Delta_u$) are more strongly coupled with the triplet ground state ($X^3\Sigma_g^-$) than with the singlet states ($c^1\Sigma_u^-$, $b^1\Sigma_g^+$, $a^1\Delta_g$) because the $O_2(X)$ production is considered to be invariable.

experimentally confirmed. Hislop and Wayne (1977) emphasized two sources of the emission line at $\lambda_{1270}$=1270 nm. The first source is the $O_2(a-X)\{0-0\}$ transition at $\lambda_{1270}$ that enables determining $VER\{O_2(a-X)\}$ profiles. The second source is the $HO_2\{^2A'(001)-{}^2A''(000)\}$ electronic transition at $\lambda_{HO2}$=1265±10 nm, which is very close to $\lambda_{1270}$. $^2A''$ denotes the ground state of $HO_2$, $^2A'$ – its first excited state, and three numbers in parentheses – various levels of the vibrational excitation.

5   Additionally, Hislop and Wayne (1977) mentioned the reaction $HO_2\{^2A''(001)\} + O_2 \rightarrow HO_2 + O_2(a)$, which negligibly produces $O_2(a)$. It is possible to process $OH^*$ emissions in future versions of the MAC model applied to measurements obtained during the ETON campaign, but emissions related to the excited $HO_2$ ($HO_2^*$) were measured neither during the ETON campaign nor other rocket campaigns known to the authors of this article. Unfortunately, it would be not enough to extend future versions of the MAC model with processes considering vibrational levels of $OH^*$ because of the $HO_2^*$ contribution. Sharma

10   et al. (2015) proposed a new mechanism responsible for the deactivation of $OH^*$ as follows: $OH(\nu' \geq 5) + O(^3P) \rightarrow OH(0 \leq \nu'' \leq \nu' - 1) + O(^1D)$. Sharma et al. (2015) emphasized that this meachanism is represented by two reactions producing a transient $HO_2^*$ complex at first, which is de-excited resulting in products shown in the proposed mechanism on the right. Contributions of processes involving both $OH^*$ and $HO_2$ to the production of $O_2(a)$ need to be considered in order to retrieve



$[O(^3P)]$ using VER$\{O_2(a-X)\}$. This enables concluding that the disagreement of the reference $[O(^3P)]$ profiles with current $[O(^3P)]$ profiles retrieved at the step 3.2 using the MAC model will remain if the only currently known *in-situ* measurements are applied.

In summary, the MAC model was carefully applied to retrieve $[O(^3P)]$ on the basis of a limited number of VER profiles: (1) including or neglecting VER$\{O_2(A-X)\}$ and VER$\{O_2(A'-a)\}$ profiles and (2) using all VER profiles or a VER$\{O_2(b-X)\}$ profile only. This is possible because calculations carried out using the MAC model are separated by steps, and concentrations of various $O_2$ states are considered at each of the following retrieval steps provided in Table 8.

## 3.5 Influence of perturbations in model parameters on $[O(^3P)]$ retrieved using the MAC model

The results of the $[O(^3P)]$ retrievals carried out with the MAC model depend on values of the following MAC input parameters: temperature (T), $[N_2]$, $[O_2]$ and VER profiles. Therefore, the impact of perturbations in VER profiles by error values provided by Greer et al. (1986), see Section 2, and the impact of perturbations in profiles of T, $[N_2]$ and $[O_2]$ by 5% of their values on the retrieved $[O(^3P)]$ profiles is estimated and discussed in this section. Specifically, these retrieved (hereafter referred to as perturbed) $[O(^3P)]$ profiles are compared with the unperturbed (hereafter referred to as reference) $[O(^3P)]$ profiles estimating differences between them as follows:

$$\epsilon = [O^{\text{current}}] - [O^{\text{reference}}], \tag{5}$$

where the $[O^{\text{reference}}]$ profiles are shown in Fig. 4. To keep the results obtained according to Eq. (5) positive, perturbations in T were chosen to be introduced by 5% of T, but perturbations in $[N_2]$ and $[O_2]$ by -5% of the respective ($[N_2]+[O_2]$) values. Perturbations in VER profiles were introduced by positive values of the respective error values. Specifically, the absolute accuracy of VER$\{O_2(a-X)\}$ (Infrared Atmospheric band, IRAtm) values was assumed to be $\pm20\%$, and the absolute accuracy of the other VER values was assumed to be $\pm10\%$ according to Greer et al. (1986), see Section 2 for details.

Both the perturbed and reference $[O(^3P)]$ profiles were retrieved using the MAC model with one MAC input parameter perturbed at a time according to the description provided in the beginning of this section. For instance, values of one VER profile only were perturbed at the particular retrieval step, see Table 8 for an overview of all steps of the consequent retrieval procedure. Figure 6 shows $\epsilon$ values in units of atoms cm$^{-3}$ illustrating the influence of the perturbed input parameters on $[O(^3P)]$ profiles. Because the number of VER profiles used in the $[O(^3P)]$ retrieval increases with each step, the number of profiles of $[O(^3P)]$ differences also increases from the top left panel to the bottom middle panel of this figure. Note that a VER profile, which was considered to have a significant impact at one of the retrieval steps performed previously to calculate the corresponding concentration profile, was taken only implicitly into account at the current retrieval step, at which these concentrations are considered instead of the corresponding VER profile. A comparison of difference values shown in various panels indicates that perturbations in the VER and T profiles introduced simultaneously will cause the highest impact on $[O(^3P)]$ profiles.




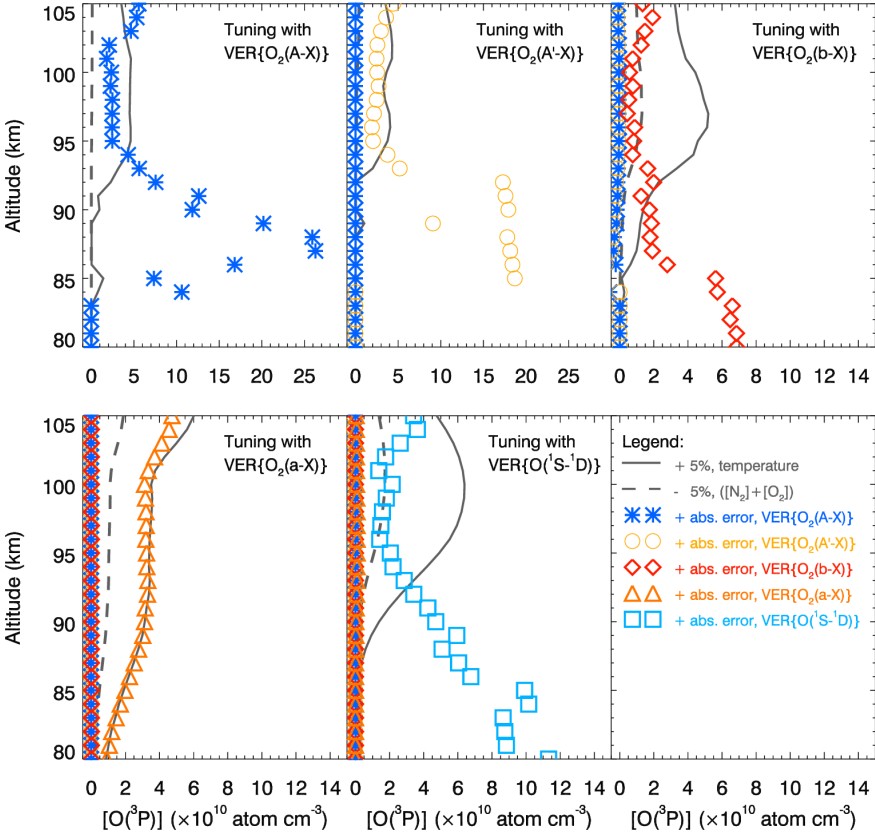

**Figure 6.** Effects of perturbations in the MAC input parameters on the retrieved $[O(^3P)]$ profiles. The retrievals were performed at the steps 2.1, 2.2, 2.3, 3.2 and 4.1 described in Table 8 on the basis of the following perturbed input parameters: Volume Emission Rates (VER), temperature (T), $[N_2]$ and $[O_2]$. Additionally, $[O(^3P)]$ profiles were retrieved on the basis of the not perturbed input parameters and denoted as reference $[O(^3P)]$ profiles shown in Fig. 4. Finally, differences between the reference and perturbed $[O(^3P)]$ profiles were estimated and shown in five panels using the colors of the perturbed input parameters shown in the legend, which is shown in the sixth panel (the last panel in the bottom row). The units of the differences shown in all panels of the top row are the same as those of the bottom row. VER values were perturbed by values of the absolute error: +20% for the VER$\{O_2(a-X)\}$ profile and +10% for the other VER profiles. Data sets of T, $[N_2]$ and $[O_2]$ were obtained using the NRLMSISE-00 model and perturbed by 5%: +5% for the T values and -5% for the sum of the $[N_2]$ and $[O_2]$ values. Profiles of $[O(^3P)]$ differences determined by perturbing VER profiles are shown by colored symbols, those determined by perturbing T profiles by solid gray lines, and those determined by perturbing ($[N_2] + [O_2]$) profiles by dashed gray lines. Each retrieval step is indicated by the name of the corresponding *in-situ* ETON VER profile shown in the upper right corner of each panel.





## 4  Discussion of the obtained results

*In-situ* measurements obtained during the ETON campaign enable estimating the efficiency of $[O(^3P)]$ retrievals carried out using the well-known photochemical models and the proposed MAC model, see Section 3.4. For instance, Lednyts′kyy et al. (2015) considered $O_2^*$ as the $O(^1S)$ precursor to retrieve the SCIAMACHY $[O(^3P)]$ time series, see Sections 3.1 and 3.4. Further work discussed here and by Lednyts′kyy and von Savigny (2016) and Lednyts′kyy et al. (2018) validated suggestions and retrievals of Lednyts′kyy et al. (2015) carried out on the basis of various rocket campaigns that enabled proposing the MAC model. For instance, $O_2(cba, X)$ were adopted in the MAC model from the M- and H-models (see Sections 3.2.1 and 3.2.2, respectively) instead of $O_2^*$ considered by Lednyts′kyy et al. (2015) in the G-model (see Section 3.1).

Additionally to the excited singlet states $O_2(c, b, a)$, Huestis (2002) and Slanger et al. (2004b) considered $O_2(^5\Pi)$ and the triplet Herzberg states ($O_2(A)$ and $O_2(A')$) coupled with $O_2(c, b, a, X)$ which was also adopted in the MAC model. Specifically, processes coupling $O_2(^5\Pi, A, A', c, b, a, X)$ and $O(^1S, ^1D, ^3P)$ with each other were proposed as complementary processes in the MAC model. In fact, the removal of the $O_2(^5\Pi)$–$O_2(A, A')$–group through collisions was suggested by Slanger et al. (2004b) and implemented in the MAC model implicitly by increasing the association rates of $O_2(b, a, X)$ in the three-body recombination reactions. This was done implicitly because reactions including $O_2(^5\Pi)$ are not well known, e.g., compare Krasnopolsky (2011) and Krasnopolsky (1986). It should be noted that $O_2(^5\Pi)$ has a shorter lifetime and a higher energy compared to the other states $O_2(A, A', c, b, a, X)$ as it was also mentioned by Huestis (2002) and Slanger et al. (2004b).

The removal of the $O_2(^5\Pi)$–$O_2(A, A')$–group and the weak coupling of the $O_2(A, A')$ triplet states with the $O_2(c, b, a)$ singlet states enabled omitting the $O_2(A, A')$ states in the MAC model. There are three reasons for the weak coupling of the $O_2(A, A')$ triplet states with the $O_2(c, b, a)$ singlet states. Firstly, the $O_2(A)$ and $O_2(A')$ states are strongly coupled with each other because vibrational states of these triplet states are energetically very close to each other. Vibrational states of these triplet states and the $O_2(c)$ singlet state are also very close to each other, but the spin flip energy is required for transitions from these triplet states to the $O_2(c)$ singlet state. Secondly, the probability of transitions from $O_2(A, A')$ to $O_2(b, a)$ is supposed to be negligibly higher than that of transitions to $O_2(X)$ because of Franck-Condon factors. Considering internuclear distances (INDs) of the corresponding Franck-Condon factors, it should be emphasized that the difference in INDs between the excited $O_2(A)$ state and the ground $O_2(X)$ state is approximately equal to the difference in INDs between the excited $O_2(A')$ state and the ground $O_2(X)$ state. Additionally, the difference in INDs between the excited $O_2(b)$ state and the ground $O_2(X)$ state is approximately equal to the difference in INDs between the excited $O_2(a)$ state and the ground $O_2(X)$ state. Thirdly, the probability of transitions from $O_2(A, A')$ to $O_2(X)$ is supposed to be significantly higher than that of transitions to $O_2(b, a)$ because of a required spin flip. Note that data about INDs and Franck-Condon factors are used to calculate the transition intensities (Hollas, 2004). Therefore, we conclude that transitions from $O_2(A, A')$ to $O_2(X)$ are more probable than transitions from $O_2(A, A')$ to $O_2(c, b, a)$.

It should be kept in mind during the interpretation of the obtained results that the uncertainties of the ETON data sets are 10...20% in VER peak values, see Section 2). Varying the MAC input data within these uncertainty ranges significantly





influences the magnitude of products obtained with the MAC model. For example, the retrieved $[O(^3P)]$ peak values increase by up to 40% if VER values are increased by 10% due to the VER uncertainty, compare Figs. 6 and 3. Additionally, uncertainty in the *in-situ* $[O(^3P)]$ profile values of less than about 40% in $[O(^3P)]$ peak values is very high implying that novel *in-situ* data sets obtained with more accurate measurement techniques should be measured in the future. In fact, the ETON

*in-situ* measurements were used to tune unknown or poorly constrained rate values of the complementary processes, and the importance of precise *in-situ* measurements is tremendous. Nevertheless, rate values of the processes implemented in the MAC model are considered to be validated through a comparison of the *in-situ* and retrieved $[O(^3P)]$ profiles. In the following three sections we discuss the tuning based on the ETON data set.

### 4.1 Discussion of tuned rate values of quenching processes implemented in the MAC model

All processes of the MAC model are provided in Section 3.3. These processes were separated into four groups: those considered in the G-, M- and H-models as well as those considered as complementary processes completing the MAC model and denoted as C-processes. Unknown or poorly constrained reaction rate values of the C-processes were tuned comparing (1) retrieved and evaluated concentrations of excited chemical species, (2) *in-situ* and evaluated VER profiles as well as (3) *in-situ* and retrieved $[O(^3P)]$ profiles. The validation procedure is related to the comparison of $[O(^3P)]$ profiles, and the verification procedure is

related to the comparison of the other profiles, see Section 3.4. The verification and validation results support the use of the adjusted reaction rates provided in Tables 9 ... 12.

As it was mentioned in Section 2, unknown or poorly constrained reactions in the MAC model were tuned on the basis of the ETON *in-situ* measurements and applied to data sets measured during the WADIS-2, WAVE2000 and WAVE2004 campaigns, see the next article to be submitted. Dr. Fytterer and Dr. Sinnhuber from the Karlsruhe Institute of Technology suggested

the rate values of the reactions $R_{b2.1}$, $R_{b4.1}$, $R_{b6.0}$, $R_{r1.2}$ and $R_{r2.3}$ for the data sets of the WAVE2004 campaign. The other reaction rates were adjusted on the basis of the described verification and validation procedures. Particularly, the $R_{a2.2}$ reaction rate was also adjusted within the range provided by Burkholder et al. (2015), who gave the upper limit of this reaction. Rate values of the reactions $R_{t10.1}$, $R_{d9.1}$ and $R_{c2.1}$ regarding the $O(^1S)$ production were adjusted taking studies of Krasnopolsky (2011), Huang and George (2014), Steadman and Thrush (1994) and Torr et al. (1985) into account. The adjustment of rate

values of the three-body recombination reactions is described in Section 4.2.

The tuning of the rate coefficients was carried out by changing the values of dimensionless scaling factors (cTDu, cTCu, cTBu, cTAu, cDCu, cDBu, cDAu, cCBa, cCBm, cCAa, cCAm, cBAa, cBAm and cAXa shown in Tables 9 ... 12), which are multiplied with the corresponding rate coefficients and describe the strength of the coupling among $O_2$ states as follows:

1. cTDu is for coupling of $O_2(A)$ and $O_2(A')$, cTCu – $O_2(A)$ and $O_2(c)$, cTBu – $O_2(A)$ and $O_2(b)$, cTAu – $O_2(A)$ and

$O_2(a)$.

2. cDCu is for coupling of $O_2(A')$ and $O_2(c)$, cDBu – $O_2(A')$ and $O_2(b)$, cDAu – $O_2(A')$ and $O_2(a)$.





3.  cCBa is for coupling of $O_2(c)$ and $O_2(b)$ by quenching of $O_2(c)$ with $O(^3P)$, cCBm – $O_2(c)$ and $O_2(b)$ by quenching of $O_2(c)$ with $O_2(X)$, cCAa – $O_2(c)$ and $O_2(a)$ by quenching of $O_2(c)$ with $O(^3P)$, cCAm – $O_2(c)$ and $O_2(a)$ by quenching of $O_2(c)$ with $O_2(X)$.

4.  cBAa is for coupling of $O_2(b)$ and $O_2(a)$ by quenching of $O_2(a)$ with $O(^3P)$, cBAm – $O_2(b)$ and $O_2(a)$ by quenching of $O_2(b)$ with $O_2(X)$.

5.  cAXa is for coupling of $O_2(a)$ and $O_2(X)$ by quenching of $O_2(a)$ with $O(^3P)$.

Values of these scaling factors were altered to determine their influence on $[O^{\text{current}}]$ calculating differences with respect to $[O^{\text{reference}}]$ retrieved without adjusting these scaling factors. The differences were calculated according to Eq. (5) and used in the sensitivity analysis, the results of which are given by the interval of possible rate values shown in the third column of Tables 9 … 12. For instance, perturbations in cTDu values do not cause changes in retrieved and evaluated MAC output parameters. Therefore, the tested interval is shown as cTDu $\in [1 \cdot 10^{-30}, 1 \cdot 10^{30}]$ in Table 9, and cTDu is set to an arbitrary value of cTDu $= 1 \cdot 10^{-2}$.

Additionally, the rate value of the $R_{a2.2}$ reaction was adjusted in the interval cAXa $\in [1 \cdot 10^{-30}, 1 \cdot 10^{-2}]$ of possible values multiplied by $2 \cdot 10^{-16}$ molec$^{-1}$ cm$^3$ s$^{-1}$ and applied at the retrieval step 3.2 shown in Table 8. This adjustment of the scaling factor cAXa is allowed because $R_{a2.2} = 2 \cdot 10^{-16}$ molec$^{-1}$ cm$^3$ s$^{-1}$ is given by Burkholder et al. (2015) as the upper interval value. A $R_{a2.2}$ reaction rate of higher than $R_{a2.2} = 2 \cdot 10^{-17}$ molec$^{-1}$ cm$^3$ s$^{-1}$ seems by sight to cause higher $[O(^3P)]$ peak values than those obtained with $R_{a2.2} = 2 \cdot 10^{-18}$ molec$^{-1}$ cm$^3$ s$^{-1}$. Therefore, cAXa $= 1 \cdot 10^{-2}$ is used so that the $R_{a2.2}$ reaction rate equal to $2 \cdot 10^{-18}$ molec$^{-1}$ cm$^3$ s$^{-1}$ is employed in the MAC model.

## 4.2 Discussion of the obtained results regarding tuned rate values for implemented three-body recombination processes

This section continues reporting about adjusting unknown or poorly constrained reaction rate values.

The MAC model was proposed on the basis of the hypothesis of Huestis (2002) and Slanger et al. (2004b), who stressed that association rates of excited $O_2$ states in the three-body recombinations must be modified because $O_2$ molecules in various excited states collide with each other and other molecules so that an excitation transfer takes place. However, Huestis (2002) and Slanger et al. (2004b) did not provide modified association rates. This was also emphasized by Krasnopolsky (2011), who applied the two-step Barth excitation transfer scheme for each of the ETON VER profiles separately. Thus, Krasnopolsky (2011) substantially limited the number of the considered chemical reactions related to $O_2(^5\Pi)$ comparing to Krasnopolsky (1986). Because the lifetime of $O_2(^5\Pi)$ is less than $\sim 0.4\mu$s (Slanger and Copeland, 2003), it is impossible to determine a number of reaction rates involving $O_2(^5\Pi)$ in the laboratory. For this reason reactions involving $O_2(^5\Pi)$ cannot be adequately included in chemical-dynamic time-dependent atmospheric models. Nevertheless, the association rate values of $O_2$ states were tuned with the use of the hypothesis of Slanger et al. (2004b) to apply them in the MAC model as follows. Firstly, the theoretically known association rates (Bates, 1988a) were considered. Then, they were used to obtain the new association rate values of $O_2(b, a, X)$, see the respective yielding factors $bY$, $aY$ and $xY$ in Tables 10 and 11. Specifically, values of the known





association rates were increased using the association rate ($pY$) of $O_2(^5\Pi)$. For instance, the association rate of $O_2(b)$ was increased by an arbitrary value of 7% of the $pY$ value to determine a new value of $bY$. In a similar way, the association rates of $O_2(a)$ and $O_2(X)$ were increased by arbitrary values of 68% and 25% of the $pY$ value to determine new values of $aY$ and $xY$, respectively.

It should be noted that Bates (1988a) provided the association rates for $O_2(^5\Pi, A, A', c, b, a, X)$ applying the concept of a hard-sphere to the reaction rates in the three-body recombinations ($O(^3P) + O(^3P) + \{N_2, O_2\}$) as it was done by Bates (1951), Wraight (1982) and Smith (1984). It is remarkable that $N_2$ was used as the third body in laboratory studies and that the reaction rate of the three-body recombination updated by Smith and Robertson (2008) is lower than that one provided by Campbell and Gray (1973) above 200 K and higher below 200 K. Nevertheless, Campbell and Gray (1973) and Smith and Robertson (2008)

assumed the obtained reaction rate ($\chi_{N2}^{Px}$) to be equal to that one ($\chi_{O2}^{Px}$) considering $O_2$ as the third body because of the used hard-sphere concept. Unfortunately, neither $\chi_{O2}^{Px}$ nor $\chi_{N2}^{Px}$ is provided in the established studies on chemical kinetics, e.g. the Jet Propulsion Laboratory databases (Burkholder et al., 2015). It is worth being mentioned that Bates (1979) interpreted the Chapman excitation process as follows: two colliding $O(^3P)$ atoms create an electronically excited $O_2$ molecule, which is presumably in the upper Herzberg state (Greer et al., 1987), see Section 1 for details. This altogether implies that an interaction

of $O_2$ in the ground or excited states with one or more $O(^3P)$ atoms is a complicated process worth of further investigation, and the hard-sphere concept should be used with caution.

There are two main adjustments done in the MAC model with respect to the three-body recombinations. The first one is related to the increased association rates of $O_2(b, a, X)$ taking collisions of higher excited $O_2$ molecules with $O_2(^5\Pi)$ into account and being implicitly considered in the MAC model. The second one is related to the increase of $\chi_{O2}^{Px}$ compared

to $\chi_{N2}^{Px}$ of the reactions $R_{x1.2}$ and $R_{x1.1}$, respectively. This adjustment was done because the used hard-sphere concept is probably misleading and because other $O(^3P)$ loss processes were required to be implemented in the MAC model implicitly according to the verification and validation procedures. The origin of the required $O(^3P)$ loss processes is currently not known definitely because both photochemical and dynamical phenomena might contribute to the total $O(^3P)$ loss. Note that the $O_2$ photodissociation into $O(^3P)$ atoms has its maximum at ~120 km according to Solomon and Qian (2005), and Colegrove et al.

(1965) invoked eddy diffusion to describe the $O(^3P)$ loss by transport from the lower thermosphere downwards.

Two cases are considered adjusting rate values of the $R_{x1.1-2}$ reactions considered in the MAC model. In the first case the $\chi_{O2}^{Px}$ rate value is multiplied by $\sim 3.56 \cdot 10^4$, and the $\chi_{N2}^{Px}$ rate value is left to be equal to that one given by Smith and Robertson (2008). The first case is used as the standard case of using the $R_{x1.1-2}$ reactions in the MAC model. In the second possible case used optionally both rate values ($\chi_{O2}^{Px}$ and $\chi_{N2}^{Px}$) are multiplied by $7.67 \cdot 10^3$. The $R_{x1.1-2}$ reactions are only involved in the last

$[O(^3P)]$ retrieval step considering all chemical species. The rate values of the $R_{x1.1-2}$ reactions were tuned and applied on the basis of the *in-situ* data sets obtained during the ETON and WAVE2004 campaigns described in Section 2. The dependence of $[O(^3P)]$ values on $O(^3P)$ loss processes is very high at the last retrieval step 5.1 because the $R_{x1.1-2}$ reactions are taken into account in the continuity equations at this step.

It should be noted that $[O(^3P)]$ values retrieved at the steps 2.1, 2.2, 2.3, 3.1, 3.2 and 4.1 significantly depend on perturbations

in VER values. As for the last retrieval step 5.1, concentrations of chemical species are retrieved at this step at first, and



$[O(^3P)]$ values are retrieved afterwards. It follows from the discussion of Fig. 6 that the dependence of $[O(^3P)]$ values on VER values used at the current step, e.g. the step 4.1, at which VER values belong to the MAC input parameters, is lower than the dependence of $[O(^3P)]$ values on VER values used at the previous steps.

In summary, the verification and validation procedures based on the comparison of the *in-situ* $O(^3P)$ profile with several $O(^3P)$ profiles retrieved at the steps 2.1, 2.2, 2.3, 3.1, 3.2, 4.1 and 5.1 support the complementary reactions considered in the continuity equations, see Appendix A. This implies that additional $O(^3P)$ loss processes considered by the $R_{x1.1-2}$ reactions implicitly are supported by calculations carried out with the MAC model, see also the next section.

## 4.3 Discussion of the causes responsible for additional $O(^3P)$ loss processes

This section deals with additional $O(^3P)$ loss processes implicitly considered in the MAC model by the $R_{x1.1-2}$ reactions according to the description provided in the previous section. Continuity equations implemented in the MAC model include the production and loss terms of various chemical species. The mentioned additional $O(^3P)$ loss processes concluded using results obtained at the last $[O(^3P)]$ retrieval step 5.1 were validated on the basis of all results obtained with the MAC model at each of the retrieval steps. Unfortunately, there are not enough data to quantify contributions of the diffusive velocities (molecular and turbulent ones) and the Eulerian mean velocity in the considered continuity equations to the transport of various chemical species. For instance, the molecular diffusive velocity may contribute to the additional $O(^3P)$ loss processes. Processes of atmospheric dynamics (damped atmospheric gravity waves, tides and advection) significantly contribute to the transport processes (Smith et al., 1987; Johnson and Gottlieb, 1973). Atmospheric dynamics responsible for the Eulerian mean velocity and the turbulent diffusive velocity would influence all chemical species considered in a continuity equation. As far as temperature, $[N_2]$ and $[O_2]$ are obtained with the NRLMSISE-00 model for the ETON campaign, these transport velocity components are not discussed in this article. Instead, they are discussed in our next article describing the WADIS-2, WAVE2000 and WAVE2004 campaigns.

The maximum of the $O_2$ photodissociation into $O(^3P)$ atoms is at ~120 km (Solomon and Qian, 2005). Shematovich et al. (2011) and Wei et al. (2014) discussed the ionized $O(^3P)$ drag to outer space. This drag might play a relatively negligible role at normal solar activity and atmospheric conditions due to a low-rate production of the ionized $O(^3P)$ from inelastic collisions involving $O(^3P)$ atoms. Colegrove et al. (1965) discussed the downward $O(^3P)$ transport from the lower thermosphere. The total downward $O(^3P)$ transport was explained by Colegrove et al. (1965) to occur due to high values of the diffusive transport velocity. Note that Grygalashvyly et al. (2012) and Qian et al. (2009) also derived relatively high values of the diffusive transport velocity in the MLT region comparing to those of Swenson et al. (2018).

The molecular diffusion velocity was emphasized in Brasseur and Solomon (2005) on page 138 to occur because of elastic collisions between particles and taking into account the effect of thermal diffusion, whereas reactive collisions were neglected. The issue regarding reactive collisions was discussed in Section 1 with respect to difficulties calculating the respective rate coefficients. In fact, it is even difficult to address the static and combined quenching processes in the laboratory, where dynamic quenching processes are often studied with the use of the Stern-Volmer method Lakowicz (2006). For instance, a tetraoxygen molecule, the chemistry of which is not well known because it had been detected by Cacace et al. (2001) recently, may be



produced from reactive collisions involving $O(^3P)$. It can be concluded that reactive collisions are not considered in the steady state continuity equations applied in the MAC model, but they should be taken into account. Therefore, a temporary solution was introduced to implement possible $O(^3P)$ loss processes discussed in the previous section implicitly, i.e. simply increasing the rate value of the three-body recombination reaction with $O_2$ as a third body.

**4.4  Discussion of the obtained results regarding the $O(^1S)$ precursor**

Preliminary conclusions about the origin of the $O(^1S - {}^1D)$ green line emission are drawn on the basis of the VER and $[O(^3P)]$ profiles shown on the left in Figs. 3, 4 and 5 compared with each other in Section 3.4. As far as the shown VER and $[O(^3P)]$ profiles retrieved via different pathways indicated in Figs. 1 and 2 seem to be in perfect agreement with each other, it was concluded that the contribution of processes involving $O_2(A)$ and $O_2(A')$ to the $O(^1S)$ production is negligible, and $O_2(c)$
was considered to be the major $O(^1S)$ precursor.

  We start the discussion ragarding the $O(^1S)$ precursor with two main findings and finish with considering arguments published previously.

  Firstly, the MAC model is based mainly on the two-step Barth excitation transfer schemes which requires to consider the $O(^1S)$ precursor, see Sections 1 and A. The nature of the oxygen green line emission was investigated by many atmospheric
scientists on the basis of *in-situ* airglow measurements by tuning the reaction rates including the $O(^1S)$ precursor as a not identified $O_2^*$ state and the comparison of these rates with the ones measured in a ground-based laboratory. It can be assumed that the deduced $O_2^*$ corresponds to an excited $O_2$ in a specific state or a group of $O_2$ states according to Huestis (2002). However, the hypothesis of Huestis (2002) was refuted by Slanger et al. (2004b).

  Secondly, the Barth excitation transfer scheme was implemented in the MAC model sequentially considering $O_2(A)$, $O_2(A')$
and $O_2(c)$ as multiple $O(^1S)$ precursors according to Slanger et al. (2004b). It should be noted that $O_2(A)$, $O_2(A')$ and $O_2(X)$ are triplet states, and $O_2(c)$ is a singlet state. The verification and validation results shown in Section 3.4 enable separating MAC processes in two groups related to $O_2(^5\Pi, A, A', c, b, a, X)$ and $O(^1S, {}^1D\,{}^3P)$ as well as related to $O_2(^5\Pi, c, b, a, X)$ and $O(^1S, {}^1D\,{}^3P)$. This conclusion reflects the importance of the ETON rocket campaign (Greer et al., 1986) for identifying the $O(^1S)$ precursor.

$O_2(c)$ was proposed by Solheim and Llewellyn (1979), Llewellyn et al. (1980) and Krasnopolsky (1981) to be the $O(^1S)$ precursor on the basis of the electron-impact excitation spectrum of $O_2$ determined by Trajmar et al. (1972) and Stern-Volmer relations. As far as the results of Trajmar et al. (1972) are also valid for $O_2(^5\Pi)$, Krasnopolsky (1986) and Krasnopolsky (2011) proposed $O_2(^5\Pi)$ to be a possible $O(^1S)$ precursor. Nevertheless, $O_2(A)$ was concluded by Krasnopolsky (2011) to be the most probable $O(^1S)$ precursor according to experimental measurements of Stott and Thrush (1989) and Steadman and
Thrush (1994).

  Stott and Thrush (1989) excluded $O_2(^5\Pi)$, $O_2(A', \nu = 2 - 4)$ and $O_2(c, \nu = 0)$ from the list of possible $O(^1S)$ precursors and concluded that $O_2(A, \nu \geq 5)$ is the $O(^1S)$ precursor. Various arguments were provided Stott and Thrush (1989) on the basis of results obtained with the use of the Stern-Volmer relationship applied for each of the possible $O(^1S)$ precursors. Some of the arguments against $O_2(c)$ were based on the quenching of the triplet $O_2(A, A')$ states converting to the singlet $O_2(b, a)$



states. The validity of this argument was tested in the MAC model implementing the $O_2(A)$ quenching to $O_2(b)$ by using the $R_{t4.1-3}$ reactions, the $O_2(A)$ quenching to $O_2(c)$ by using the $R_{t3.1-3}$ reactions, and the $O_2(A)$ quenching to $O_2(a)$ by using the $R_{t6.1-3}$ reactions. The results of the sensitivity analysis discussed in Section 4.1 show that these reactions can be neglected in the MAC model, see Tables 5 and 9. Similarly, the $O_2(A')$ quenching to $O_2(c, b, a)$ implemented in the reactions $R_{d2.1-2}$,

$R_{d3.1-2}$ and $R_{d4.1-2}$ can be also neglected in the MAC model. Quenching of the triplet $O_2(A, A')$ states to the singlet $O_2(b, a)$ states requires the spin flip that is energetically not favorable, and the arguments of Stott and Thrush (1989) can be considered as refuted. Therefore, $O_2(c, \nu \geq 2)$ can be considered the $O(^1S)$ precursor.

Steadman and Thrush (1994) excluded $O_2(A', c)$ from the list of possible $O(^1S)$ precursors and concluded that $O_2(A, \nu \geq 6)$ is the $O(^1S)$ precursor. As for the Franck-Condon factors in the $O_2(A-X)$ transitions, they were emphasized by Krasnopol-

sky (2011) to be low, so that $O_2(A, \nu \leq 5)$ in low vibrational levels does not seem to be an effective transition path of producing $O(^1S)$ from $O(^3P)$. The arguments provided by Steadman and Thrush (1994) against $O_2(A', c)$ as the $O(^1S)$ precursors were based on the general idea that the $O_2(A'^3\Delta_u)$ and $O_2(c^1\Sigma_u^-)$ quenching to $O_2(X^3\Sigma_g^-)$ is not symmetry allowed, but the $O_2(A^3\Sigma_u^+)$ quenching to $O_2(X^3\Sigma_g^-)$ is symmetry allowed. The validity of this argument was tested in the MAC model implementing the $O_2(A', c)$ quenching to $O_2(X)$ by using the reactions $R_{d9.1}$ and $R_{c2.1}$.

Steadman and Thrush (1994) suggested that if $O_2(c)$ is considered to be the $O(^1S)$ precursor, then it is probably at the vibrational state $\nu = 8$ because of the favorable Frank-Condon factors for transitions to vibrational states of the electronic $O_2$ ground state $O_2(X)$. Krasnopolsky (1981) also considered $O_2(c)$ as the $O(^1S)$ precursor on the basis of observations in the atmospheres of Venus and Mars, where $O_2(c)$ is in the vibrational ground state $\nu = 0$. Krasnopolsky (1981) concluded that the activation energy of $2.1\,\mathrm{kcal\,mol^{-1}}$ is required for quenched $O_2(c, \nu = 0)$ molecules to produce $O(^1S)$. Altitude profiles of the

fractional $O_2(c)$ vibrational populations with $\nu = 3 \ldots 10$ are characterized by various peak altitude values in the altitude range $80 \ldots 120\,\mathrm{km}$, where they were derived by Llewellyn and McDade (1984) from a model by using reaction rate values given by Kenner and Ogryzlo (1983). The $[O_2(c, \nu = 6)]$ peak is at $94\,\mathrm{km}$, and the $[O_2(c, \nu = 8)]$ peak is at $103\,\mathrm{km}$ according to the results of atmospheric modeling shown in Fig. 5 in Llewellyn and McDade (1984). These results enable determining the peak of $[O_2(c, \nu = 7)]$ at about $97\,\mathrm{km}$, where the green line emission peak is, see Table 1. Additionally, the modeling results obtained

by López-González et al. (1992a) and shown in their Fig. 6c indicate that the $[O_2(c, \nu = 6)]$ peak is at about $97\,\mathrm{km}$. Stott and Thrush (1989) compared results obtained with laboratory experiments and atmospheric models (their Fig. 10) and found that the maximum of the relative vibrational $O_2(A)$ population is at $O_2(A, \nu = 2, 3)$ in laboratory experiments and at $O_2(A, \nu = 5)$ in model results. It follows that the maximum of the relative vibrational $O_2(c)$ population found in laboratory experiments might differ from the respective model results published in, e.g., Llewellyn and McDade (1984) and López-González et al. (1992a).

It is difficult to argue about the fractional $O_2(c)$ vibrational population of the $O(^1S)$ precursor because the $[O_2(c, \nu = 6, 7, 8)]$ peak might be at about $97\,\mathrm{km}$ altitude if the collisional activation energy mentioned by Krasnopolsky (1981) is neglected.




## 5    Conclusions

Photochemical processes in the altitude range $80\ldots105\,\mathrm{km}$ were modeled considering seven states of molecular oxygen, $O_2(^5\Pi, A, A', c, b, a, X)$, and three states of atomic oxygen, $O(^1S, {}^1D\,{}^3P)$. The Multiple Airglow Chemistry (MAC) model was proposed to explain the excitation mechanisms responsible for observed airglow. Processes of the photochemical models

discussed in Sections 3.1, 3.2.1 and 3.2.2 were combined with complementary processes suggested to complete the list of processes implemented in the MAC model. Additional processes were proposed to couple the mentioned $O_2$ states and to implement the $O_2(^5\Pi)$–$O_2(A, A', c)$–group in the MAC model according to the hypothesis of Slanger et al. (2004b). *In-situ* VER profiles obtained during the ETON campaign were applied to determine unknown or poorly constrained reaction rates and update known ones considered in the MAC model, see Sections 4.1, 4.2 and 4.3. Note that *in-situ* VER profiles obtained

during the WADIS-2, WAVE2000 and WAVE2004 campaigns were applied to validate these reaction rates used in calculations carried out with the MAC model, see future publications. We would like to emphasize that the agreement between $[O(^3P)]$ profiles obtained at various retrieval steps and the corresponding *in-situ* $[O(^3P)]$ profiles for these three campaigns is perceived as significantly better than that for the ETON campaign. The proposed algorithm enabled calculating concentrations of such coupled minor species as $O_2(A, A', c, b, a)$ and $O(^1S, {}^1D, {}^3P)$ for the first time.

The integrity of the $O_2$ electronic states' identity formulated in the hypothesis of Huestis (2002) was refuted by Slanger et al. (2004b) which hinders representing the $O(^1S)$ precursor by $O_2^*$ as it was done in Lednyts′kyy et al. (2015). Nevertheless, the $[O(^3P)]$ retrievals performed by Lednyts′kyy et al. (2015) according to the well-known and extended cubic equations were validated using the *in-situ* $[O(^3P)]$ measurements, see Sections 3.1 and 3.4. As for calculations with the MAC model, a consistent explanation of the origin of each of the considered airglow emissions, including the famous oxygen green line emission, was

proposed. Specifically, the precursors of $O_2(b)$, $O_2(a)$ and $O(^1S)$ were identified and confirmed during the verification and validation procedures provided in Section 3.4. Firstly, $O_2(c)$ and states of the $O_2(^5\Pi)$–$O_2(A, A')$–group were found to be the $O_2(b)$ precursors responsible for Atmospheric band emissions. Secondly, $O_2(c)$, $O_2(b)$ and states of the $O_2(^5\Pi)$–$O_2(A, A')$–group were found to be the $O_2(a)$ precursors responsible for Infrared Atmospheric band emissions. Finally, $O_2(c)$ was found to be the major $O(^1S)$ precursor responsible for the oxygen green line emission, whereas the contribution of $O_2(A, A')$ was found

to be negligible. Note that all states from the $O_2(^5\Pi)$–$O_2(A, A')$–group can be considered to be the $O_2(b, a, X)$ precursors because $O_2(^5\Pi)$ was implicitly used to calculate new association rate values of $O_2(b, a, X)$.

Convincing verification and validation results should be accepted critically because the tuned rate values were obtained on the basis of the *in-situ* measurements with uncertainties provided by Greer et al. (1986) and discussed in Section 2. The influence of variability was studied in various MAC input parameters, see Section 3.5. In summary, perturbations in temperature of 5%

cause variations in $[O(^3P)]$ of about 10%, but perturbations in atmospheric density of 5% cause about 3% $[O(^3P)]$ variations. Uncertainties in such VER values as $VER\{O_2(A-X)\}$ and $VER\{O_2(A'-a)\}$ at the retrieval steps 2.1 and 2.2 cause up to about 40% of $[O(^3P)]$ variations, uncertainties in such VER values as $VER\{O_2(b-X)\}$ and $VER\{O_2(a-X)\}$ at the retrieval steps 2.3 and 3.2 cause about 12% of $[O(^3P)]$ variations, but uncertainties in such VER values as $VER\{O(^1S-{}^1D)\}$ at the retrieval step 4.1 cause up to about 20% of $[O(^3P)]$ variations.





The following four key findings required to develop the MAC model were proposed for the first time to the best of our knowledge. Firstly, the algorithm was proposed without using *a priori* data applied to initiate calculations with the MAC model. Instead, sequent retrieval steps were applied to solve the system of continuity equations starting calculations from higher excited species, and providing concentrations of excited species at the following retrieval steps. Each polynomial equa-

tion was solved separately to obtain concentrations of chemical species required for the next polynomial equations, which were sequentially introduced and solved to retrieve $[O(^3P)]$ profiles, see Table 8 for retrieval steps applied using the MAC model. Secondly, participation of $O_2(^5\Pi)$ in chemical reactions was implemented implicitly adjusting the association rates of $O_2(b, a, X)$ (Bates, 1951) by using the value of the $O_2(^5\Pi)$ association rate. Thirdly, the singlet $O_2(c, b, a)$ excited states and the triplet $O_2(A, A', X)$ states as well as $O(^1S, {}^1D, {}^3P)$ states were identified and treated in the MAC model explicitly.

Fourthly, calculations carried out using the MAC model were consistently verified for each considered ETON VER profile, and validated for each $[O(^3P)]$ retrieval step, see Section 3.4. The proposed algorithm also enables applying the MAC model on the basis of a VER$\{O_2(b - X)\}$ profile only, as the $[O(^3P)]$ retrieval results show in Fig. 5.

The proposed algorithm used to solve the system of continuity equations also enabled introducing perturbations in tuned rate values that was studied in their the impact on the MAC output parameters. The results of the sensitivity analysis enable

neglecting not important processes coupling $O_2$ states, see the third column of Tables 9 ... 12 for interval values of the tuned reaction rates. For instance, transitions from the triplet $O_2(A, A')$ states to the singlet $O_2(c, b, a)$ states were found to be not intense and less probable than transitions from these excited triplet and singlet states to the triplet $O_2(X)$ ground state. This might be explained by the energy required for the spin flip during transitions between one triplet and one singlet states.

The following conclusions can be drawn from the results of the sensitivity analysis. Firstly, the triplet $O_2(A, A')$ states can

be neglected in the MAC model because of their strong coupling with the ground triplet $O_2(X)$ state. Then, the following correspondences among the violated electric dipole selection rules and the transition intensity seem to be established. Collisional deactivation processes implemented in the MAC model involving $O_2(A)$ and $O_2(A')$ were found (1) strong between $O_2(c^1\Sigma_u^-)$ and $O_2(b^1\Sigma_g^+)$, (2) weak between $O_2(A'^3\Delta_u)$ and $O_2(c^1\Sigma_u^-)$, (3) almost absent between $O_2(A^3\Sigma_u^+)$ and $O_2(A'^3\Delta_u)$ as well as $O_2(c^1\Sigma_u^-)$ and $O_2(a^1\Delta_g)$. These three findings are summarized with respect to the violation of the electric dipole selection

rules as follows: $\Sigma^+ \nleftrightarrow \Sigma^-$ and $\Delta\Lambda = 0, \pm 1$. In fact, the violation of the $\Sigma^+ \nleftrightarrow \Sigma^-$ selection rule (1) describes the change of the wave function's sign by reflection in a plane containing the internuclear axis (Bates, 1962) and (2) seems to lead to relatively intense transitions in the $O_2$ band systems. The violation of the $\Delta\Lambda = 0, \pm 1$ selection rule (1) describes the orbital (or azimuthal) angular momentum $\Lambda$) and (2) seems to lead to weak or no transitions in the $O_2$ band systems.

Two topics can be emphasized regarding open tasks of further research. Firstly, the MAC model should be extended to

consider various vibrational $O_2$ and $OH^*$ states because the MAC model was implemented with the use of the Local Thermodynamic Equilibrium (LTE) approximation and only a few $O_2$ and $OH^*$ vibrational states were considered analyzing the ETON multiple emissions. This requires the use of the non-LTE approximation, see Sections 3.2.1 and 4.2 for details, but will possibly result in the $O(^3P)$ loss. Specifically, the MAC model will be extended to consider the $[O(^1D)]$ and $[O(^3P)]$ retrieval on the basis of measured VER$\{O(^1D - {}^3P)\}$ profiles because of the role of the transient $HO_2^*$ complex emphasized in the end

of Section 3.4 and required to implement various $OH^*$ vibrational states.





*Code availability.* The algorithm described in this study is available to the community and may be obtained by contacting the lead author of this article.

**Appendix A: Algorithmic steps of the Multiple Airglow Chemistry model development**

The retrieval steps of $[O(^3P)]$ are closely related to the development of the MAC model. For instance, the well-known photo-
chemical model of McDade et al. (1986) is applied at the 1$^{st}$ substep of the 1$^{st}$ step (see Section A1.1) to retrieve $[O(^1S)]$ as a part of the prior retrieval procedure. Then simple continuity equations are applied in the prior retrieval procedure to retrieve $[O(^1D)]$ (see Section A1.2) as well as $[OH^*]$ and $[HO_2]$ (see Section A1.3). The next retrieval steps are provided with the results obtained from the prior retrieval procedure and described in the appendix starting from Section A2. An overview of these sequentially applied retrieval steps is provided in Section 3.4 in Table 8 for the MAC model.

Note that calculations of the prior retrieval procedure (see Section A1) are omitted in this study because neither the ETON campaign nor the NRLMSISE-00 model provide concentrations of chemical species required at this step. This implies that values of $[O_3]$, $[CO_2]$, $[O(^1D)]$, $[OH^*]$ and $[HO_2]$ included in calculations of the next retrieval steps (see Section A2 and the following sections) are equal to zero values.

Processes of the MAC model are described in the following sections according to the models, processes of which were
adopted in the MAC model. For instance processes of the M-model (see Section 3.2.1) are marked with a character M, those of the H-model (see Section 3.2.2) are marked with a character H and the other (complementary) processes completing the development of the MAC model are marked with a character C. The complementary (or completing) processes are related to processes of the G-model, see Section 3.1, processes introduced to implement the hypothesis of Slanger et al. (2004b) and other processes coupling $O_2$ states with each other, see Section 3.3. For instance, $O_2(A)$ is only considered in complementary
processes, and $[O_2(A)]$ are marked as $[O_2(A)$-C]. Production and loss terms of $O_2(A)$ are also marked with a character C as $P\{O_2(A)$-C\} and $L\{O_2(A)$-C\}, respectively.

There are three kinds of the MAC products:

1. Retrieved concentrations of chemical species obtained using all relevant reactions. Retrieved concentration profiles are marked with a character R, e.g. R-$[O_2(A)]$.

2. Evaluated concentrations of chemical species obtained dividing the R-VER profiles, which correspond to the particular chemical species, by the respective transition probability. Evaluated concentration profiles are marked with a character E, e.g. E-$[O_2(A)]$.

3. Evaluated VER values obtained dividing the retrieved concentrations of the respective same chemical species by the respective transition probability. Evaluated VER profiles are marked with a character E, e.g. E-VER$\{O_2(A - X)\}$.





## A1 The 1$^{st}$ retrieval step

The 1$^{st}$ retrieval step was performed at three substeps to calculate $[O(^1D)]$, $[OH^*]$ and $[HO_2]$ prior values. As for this study, this step was omitted for retrievals on the basis of measurements obtained *in-situ* during the ETON campaign, see Section 2, because profiles of temperature, $[N_2]$ and $[O_2]$ were obtained using the NRLMSISE-00 model. Nevertheless, measurements

obtained remotely and *in-situ* during the WAVE2004 campaign described in the next publication provide data sets required in the prior retrieval to apply the MAC model.

The processes shown in Tables 2 (see Section 3.1) and A1 were used for retrievals at this step. Processes marked with a character P in these tables were not used as complementary processes proposing the MAC model. The resulting concentration values obtained at the prior retrieval step are also marked with the character P.

### A1.1 Substep 1: prior retrieval of $[O(^1S)]$

The prior retrieval of $[O(^1S)]$ is performed according to the well-known cubic equation with empirical coefficients provided by McDade et al. (1986) on the basis of the *in-situ* $[O(^3P)]$ measurements.

### A1.2 Substep 2: prior retrieval of $[O(^1D)]$

The prior retrieval of $[O(^1D)]$ is performed according to the corresponding continuity equation applied on the basis of $[O_3]$

and $[CO_2]$ profiles.

The continuity equation for $[O(^1D)]$ includes the terms of the $[O(^1D)]$ production ($P\{O(^1D)\}$) and loss ($L\{O(^1D)\}$) as follows: $d[O(^1D)]/dt = P\{O(^1D)\} - L\{O(^1D)\} = 0$.

The production and loss terms were calculated according to the processes shown in Tables 2 and A1 as follows: $P\{O(^1D)\} = [O(^1S)](2R_{g1.1}[O(^3P)] + R_{g3.0}) + R_{s1.1-2}[O_2] + R_{s2.1}[O_3]$ and $L\{O(^1D)\} = [O(^1D)] \times D_r$, where the destruction term

$D_r = R_{r1.1-3}\{[O(^3P)], [O_3], [O_3]\} + R_{r2.1-4}\{[N_2], [O_2], [O_2], [CO_2]\} + R_{r3.0}$.

The prior retrieval results in $[O(^1D)]$ profile values as follows:

P-$[O(^1D)] = [O(^1D)] = ((2R_{g1.1}[O(^3P)] + R_{g3.0})[O(^1S)] + R_{s1.1-2}[O_2] + R_{s2.1}[O_3]) / D_r$.

### A1.3 Substep 3: prior retrieval of $[OH^*]$ and $[HO_2]$

The prior retrieval of $[OH^*]$ and $[HO_2]$ is performed according to the corresponding continuity equations applied on the basis of $[O_3]$, $[H]$ and $[O(^3P)]$ profiles.

The continuity equation for $[OH^*]$ including terms of the $[OH^*]$ production ($P\{OH^*\}$) and its loss ($L\{OH^*\}$) is as follows: $d[OH^*]/dt = P\{OH^*\} - L\{OH^*\} = 0$. The production and loss terms were calculated according to the processes shown in Tables 2 and A1 as follows: $P\{OH^*\} = [H]R_{h1.1}[O_3] + [H]2R_{h6.1}[HO_2] + [O(^3P)]R_{h4.1}[HO_2] = [OH^*] \times D_h$, where $D_h =$

$R_{h3.1}[O_3] + R_{h2.1}[O(^3P)]$.



The continuity equation for $[HO_2]$ including terms of the $[HO_2]$ production ($P\{HO_2\}$) and its loss ($L\{HO_2\}$) is as follows: $d[HO_2]/dt = P\{HO_2\} - L\{HO_2\} = 0$. The production and loss terms were calculated according to the processes shown in Tables 2 and A1 as follows: $P\{HO_2\} = [OH^*]R_{h3.1}[O_3] + [H][O_2]R_{h5.1-2}\{[N_2],[O_2]\} = [HO_2] \times D_2$, where $D_2 = R_{h6.1-3}[H] + R_{h4.1}[O(^3P)]$.

The system of continuity equations for $[OH^*]$ and $[HO_2]$ was transformed in the system of the two following equations: $P\text{-}[OH^*] = [OH^*] = \frac{[H]R_{h1.1}[O_3] + 2[H]R_{h6.1}[HO_2] + [O(^3P)]R_{h4.1}[HO_2]}{D_h}$ and $[HO_2] = \frac{[OH^*]R_{h3.1}[O_3] + [H][O_2]R_{h5.1-2}\{[N_2],[O_2]\}}{D_2}$ and solved for the values of $[HO_2]$. The obtained values of $[HO_2]$ were calculated as follows: $P\text{-}[HO_2] = [HO_2] = ([H]R_{h1.1}[O_3]\cdot R_{h3.1}[O_3] + [H][O_2]R_{h5.1-2}\{[N_2],[O_2]\}\cdot(R_{h3.1}[O_3] + R_{h2.1}[O(^3P)]))/(D_2D_h)$, where $D_2D_h = R_{h3.1}[O_3]\cdot R_{h6.1-3}[H] + R_{h2.1}[O(^3P)]\cdot(R_{h6.1-3}[H] + R_{h4.1}[O(^3P)])$.

## A2    The 2$^{\text{nd}}$ retrieval step

The 2$^{\text{nd}}$ retrieval step was performed within four substeps to calculate $[O_2(b)]$ values.

### A2.1    Substep 1: retrieval of $[O_2(A)]$

Herzberg I band emission measured at 320 nm was used to retrieve $VER\{O_2(A-X)\}$ values and then to retrieve $[O(^3P)]$ values according to the continuity equation for $[O_2(A)]$, i.e. the quadratic equation with respect to $[O(^3P)]$. Then, $[O_2(A)]$
values were retrieved (R-$[O_2(A)]$) on the basis of $[O(^3P)]$ values by using the continuity equation considering all relevant processes of the MAC model. The continuity equation for $[O_2(A)]$ including terms of the $[O_2(A)]$ production ($P\{O_2(A)\}$) and its loss ($L\{O_2(A)\}$) is as follows: $d[O_2(A)]/dt = P\{O_2(A)\} - L\{O_2(A)\} = 0$. The production and loss terms were calculated considering the processes shown in Tables 6 . . . 5 as follows: $P\{O_2(A)\} = P\{O_2(A)\text{-C}\} = [O(^3P)]^2 R_{t1.1-2}\{[N_2],[O_2]\}$ and $L\{O_2(A)\} = L\{O_2(A)\text{-C}\} = [O_2(A)] \times D_t$, where $D_t = (R_{t2.1-3} + R_{t3.1-3} + R_{t4.1-3} + R_{t6.1-3} + R_{t7.1-3})\{[O(^3P)],[N_2],[O_2]\} +$
$R_{t10.1}[O(^3P)] + R_{t5.0} + R_{t9.0}$. Complementary processes were used in the production and loss terms that is denoted with a character C. Therefore, R-$[O_2(A)]$ is also marked with the character C instead of the character R as follows: R-$[O_2(A)] = [O_2(A)\text{-C}] = P\{O_2(A)\text{-C}\}/D_t$. In the case when Herzberg I band emissions are not given, $[O_2(A)]$ values can be retrieved on the basis of already known $[O(^3P)]$ values.

$[O_2(A)]$ values were also evaluated (E-$[O_2(A)]$) on the basis of retrieved $VER\{O_2(A-X)\}$ values (R-$VER\{O_2(A-X)\}$),
so that only the respective transition probability was used: E-$[O_2(A)] = $ R-$VER\{O_2(A-X)\}/R_{t8.0}$.

Finally, $VER\{O_2(A-X)\}$ values were evaluated (E-$VER\{O_2(A-X)\}$) on the basis of retrieved $[O_2(A)]$ values and the respective transition probability: E-$VER\{O_2(A-X)\} = $ R-$[O_2(A)] \times R_{t8.0}$.

$[O_2(A)]$ values were retrieved and then evaluated to compare and verify these calculations. As for $VER\{O_2(A-X)\}$ values, they were also evaluated to compare them with retrieved values and so to verify calculations by using the MAC model, see
Section A2.4.



## A2.2 Substep 2: retrieval of $[O_2(A')]$

Chamberlain band emission measured at $370\,\text{nm}$ was used to retrieve $\text{VER}\{O_2(A'-a)\}$ values and then to retrieve $[O(^3P)]$ values according to the continuity equation for $[O_2(A')]$, i.e. the cubic equation with respect to $[O(^3P)]$. Note that $[O_2(A)]$ values retrieved at the previous step were used in the $[O(^3P)]$ retrieval at this step. Then, $[O_2(A')]$ values were retrieved (R-$[O_2(A')]$)

on the basis of $[O(^3P)]$ values by using the continuity equation considering all relevant processes of the MAC model. The continuity equation for $[O_2(A')]$ including terms of the $[O_2(A')]$ production ($P\{O_2(A')\}$) and its loss ($L\{O_2(A')\}$) is as follows: $d[O_2(A')]/dt = P\{O_2(A')\} - L\{O_2(A')\} = 0$. The production and loss terms were calculated considering the processes shown in Tables 6 … 5 as follows: $P\{O_2(A')\} = P\{O_2(A')\text{-C}\} = [O_2(A)]R_{t2.1-3}\{[O(^3P)],[N_2],[O_2]\} + [O(^3P)]^2 R_{d1.1-2}\{[N_2],[O_2]\}$ and $L\{O_2(A')\} = L\{O_2(A')\text{-C}\} = [O_2(A')] \times D_d$, where $D_d = (R_{d2.1-2} + R_{d3.1-2} + R_{d4.1-2} +$

$R_{d7.1-2})\{[O(^3P)],[O_2]\} + R_{d9.1}[O(^3P)] + R_{d6.0} + R_{d8.0}$. $[O_2(A')]$ profile values were retrieved as follows: R-$[O_2(A')] = [O_2(A')\text{-C}] = P\{O_2(A')\text{-C}\}/D_d$. In the case when Chamberlain band emissions are not given, $[O_2(A')]$ values can be retrieved on the basis of already known $[O(^3P)]$ values.

$[O_2(A')]$ values were also evaluated (E-$[O_2(')]$) on the basis of retrieved $\text{VER}\{O_2(A'-a)\}$ values (R-$\text{VER}\{O_2(A'-a)\}$), so that only the respective transition probability was used: E-$[O_2(A')] = \text{R-VER}\{O_2(A'-a)\}/R_{d5.0}$.

Finally, $\text{VER}\{O_2(A'-a)\}$ values were evaluated (E-$\text{VER}\{O_2(A'-a)\}$) on the basis of retrieved $[O_2(A')]$ values and the respective transition probability: E-$\text{VER}\{O_2(A'-a)\} = \text{R-}[O_2(A')] \times R_{d5.0}$.

$[O_2(A')]$ values were retrieved and then evaluated to compare and verify these calculations. As for $\text{VER}\{O_2(A'-a)\}$ values, they were also evaluated to compare them with retrieved given ones and so to verify calculations by using the MAC model, see Section A2.4.

## A2.3 Substep 3: retrieval of $[O_2(b)]$

Atmospheric band emission measured at $761.9\,\text{nm}$ was used to retrieve $\text{VER}\{O_2(b-X)\}$ values and then to retrieve $[O(^3P)]$ values according to the continuity equation for $[O_2(b)]$, i.e. the cubic equation with respect to $[O(^3P)]$. Note that $[O_2(A)]$ and $[O_2(A')]$ values retrieved at the previous steps were used in the $[O(^3P)]$ retrieval at this step. However, if the MAC model excluding $O_2(A)$ and $O_2(A')$ is used then $[O_2(A)]$ and $[O_2(A')]$ profile values are set to zero because these concentrations

were not retrieved at the previous retrieval steps. This is justified because $O_2$ in these triplet excited states is decoupled from the singlet excited states according to the hypothesis of Slanger et al. (2004b) used to propose the MAC model, which were verified and validated, (see Section 3.4. Then, $[O_2(b)]$ values were retrieved (R-$[O_2(b)]$) on the basis of $[O(^3P)]$ values by using the continuity equation considering all relevant processes of the MAC model. The continuity equation for $[O_2(b)]$ including terms of the $[O_2(b)]$ production ($P\{O_2(b)\}$) and its loss ($L\{O_2(b)\}$) is as follows: $d[O_2(b)]/dt = P\{O_2(b)\} - L\{O_2(b)\} = 0$. The

production and loss terms were calculated considering the processes shown in Tables 6 … 5.

The production term was calculated as follows: $P\{O_2(b)\} = P\{O_2(b)\text{-M}\} + P\{O_2(b)\text{-H}\} + P\{O_2(b)\text{-C}\}$, where $P\{O_2(b)\text{-M}\} = [O(^1D)]R_{r2.3}[O_2] + R_{s3.0}[O_2]$, $P\{O_2(b)\text{-H}\} = [O(^3P)]^2 R_{b1.1-2}\{[N_2],[O_2]\} + [O_2(c)]R_{c3.1-2}\{[O(^3P)],[O_2]\}$, $P\{O_2(b)\text{-C}\} = [O_2(A)]R_{t4.1-3}\{[O(^3P)],[N_2],[O_2]\} + [O_2(A)]R_{t5.0} + [O_2(A')]R_{d3.1-2}\{[O(^3P)],[O_2]\} + [O_2(c)]R_{c4.0}$ so



$$P\{O_2(b)\} = [O_2(A)]R_{t4.1-3}\{[O(^3P)], [N_2], [O_2]\} + [O_2(A)]R_{t5.0} + [O_2(A')]R_{d3.1-2}\{[O(^3P)], [O_2]\}$$
$$+ [O_2(c)]R_{c3.1-2}\{[O(^3P)], [O_2]\} + [O_2(c)]R_{c4.0} + [O(^3P)]^2 R_{b1.1-2}\{[N_2], [O_2]\} + [O(^1D)]R_{r2.3}[O_2] + R_{s3.0}[O_2].$$

The loss term was calculated as follows: $L\{O_2(b)\} = L\{O_2(b)\text{-}M\} + L\{O_2(b)\text{-}H\} + L\{O_2(b)\text{-}C\} = [O_2(b)] \times D_b$, where $L\{O_2(b)\text{-}M\} = [O_2(b)] \times (R_{b2.2-5}\{[O(^3P)], [N_2], [O_2], [CO_2]\} + R_{b3.0})$ is related to the M-model discussed in Section 3.2.1,

$L\{O_2(b)\text{-}H\} = [O_2(b)] \times (R_{b4.2-4}\{[O(^3P)], [N_2], [O_2]\} + R_{b6.0})$ is related to the H-model discussed in Section 3.2.2 and $L\{O_2(b)\text{-}C\} = [O_2(b)] \times (R_{b4.1,5-6}\{[CO_2], [O_3]\} + R_{b2.1}[O_3])$ corresponds to the complementary processes relevant here. Note that $D_b = (R_{b2.1-5} + R_{b4.1-6})\{[O_3], [O(^3P)], [N_2], [O_2], [CO_2], [O_3]\} + R_{b4.6}[O_3] + R_{b3.0} + R_{b6.0}$.

$[O_2(b)]$ values were retrieved taking M-, H- and C-processes into account as follows: R-$[O_2(b)] = [O_2(b)] = [O_2(b)\text{-}M] + [O_2(b)\text{-}H] + [O_2(b)\text{-}C]$, where $[O_2(b)\text{-}M] = P\{O_2(b)\text{-}M\}/(D_b D_c)$, $[O_2(b)\text{-}H] = P\{O_2(b)\text{-}H\}/(D_b D_c)$ and

$[O_2(b)\text{-}C] = P\{O_2(b)\text{-}C\}/(D_b D_c)$, so R-$[O_2(b)] = P\{O_2(b)\}/(D_b D_c)$. In the case when Atmospheric band emissions are not given, $[O_2(b)]$ values can be retrieved on the basis of already known $[O(^3P)]$ values.

$[O_2(b)]$ values were also evaluated (E-$[O_2(b)]$) on the basis of retrieved VER$\{O_2(b-X)\}$ values (R-VER$\{O_2(b-X)\}$), so that only the respective transition probability was used: E-$[O_2(b)] = $ R-VER$\{O_2(b-X)\}/R_{b5.0}$.

Finally, VER$\{O_2(b-X)\}$ values were evaluated (E-VER$\{O_2(b-X)\}$) on the basis of retrieved $[O_2(b)]$ values and the

respective transition probability: E-VER$\{O_2(b-X)\} = $ R-$[O_2(b)] \times R_{b5.0}$.

$[O_2(b)]$ values were retrieved and then evaluated to compare and verify these calculations. As for VER$\{O_2(b-X)\}$ values, they were also evaluated to compare them with retrieved given ones and so to verify calculations performed with the MAC model, see Section A2.4.

### A2.4    Substep 4: consistency tests in the retrieval of $[O_2(b)]$

The consistency tests in the retrievals performed with the MAC model are based on the comparison of the retrieved and evaluated values.

Calculations at the retrieval steps 2.1 and 2.2 are relevant for the MAC model involving $O_2(A)$ and $O_2(A')$, but calculations at the retrieval step 2.3 only are relevant for the MAC model excluding $O_2(A)$ and $O_2(A')$, see the following overview.

The retrieval step 2.1 described in Section A2.1 was carried out to retrieve R-$[O_2(A)]$ and $[O(^3P)]$ values on the basis of

R-VER$\{O_2(A-X)\}$ values. E-$[O_2(A)]$ values were also evaluated to compare them with R-$[O_2(A)]$ values. Additionally, E-VER$\{O_2(A-X)\}$ values were also evaluated to compare them with R-VER$\{O_2(A-X)\}$ values.

The retrieval step 2.2 described in Section A2.2 was carried out to retrieve R-$[O_2(A')]$ and $[O(^3P)]$ values on the basis of R-VER$\{O_2(A'-a)\}$ and R-$[O_2(A)]$ values. E-$[O_2(A')]$ values were also evaluated to compare them with R-$[O_2(A')]$ values. Additionally, E-VER$\{O_2(A'-a)\}$ values were also evaluated to compare them with R-VER$\{O_2(A'-a)\}$ values.

The retrieval step 2.3 described in Section A2.3 was carried out with the MAC model to retrieve R-$[O_2(b)]$ and $[O(^3P)]$ values on the basis of R-VER$\{O_2(b-X)\}$ values. E-$[O_2(b)]$ values were also evaluated to compare them with R-$[O_2(b)]$ values. Additionally, E-VER$\{O_2(b-X)\}$ values were also evaluated to compare them with R-VER$\{O_2(b-X)\}$ values.





### A3  The 3$^{rd}$ retrieval step

The 3$^{rd}$ retrieval step was performed at three substeps to calculate $[O_2(c)]$ and $[O_2(a)]$ values.

### A3.1  Substep 1: retrieval of $[O_2(c)]$

$[O_2(c)]$ values were retrieved (R-$[O_2(c)]$) on the basis of $[O_2(A)]$, $[O_2(A')]$ and $[O_2(b)]$ values (obtained at the retrieval steps

2.1, 2.2 and 2.3, respectively) as well as $[O(^3P)]$ values (obtained at the retrieval step 2.3) according to the continuity equation for $[O_2(c)]$ considering all relevant processes of the MAC model.

The continuity equation for $[O_2(c)]$ including terms of the $[O_2(c)]$ production ($P\{O_2(c)\}$) and its loss ($L\{O_2(c)\}$) is as follows: $d[O_2(c)]/dt = P\{O_2(c)\} - L\{O_2(c)\} = 0$. The production and loss terms were calculated considering the processes shown in Tables 6 ... 5.

The production term was calculated as follows: $P\{O_2(c)\} = P\{O_2(c)$-M$\} + P\{O_2(c)$-H$\} + P\{O_2(c)$-C$\}$, where $P\{O_2(c)$-M$\} = 0$, $P\{O_2(c)$-H$\} = [O(^3P)]^2 R_{c1.1-2}\{[N_2],[O_2]\}$ and $P\{O_2(c)$-C$\} = [O_2(A)]R_{t3.1-3}\{[O(^3P)],[N_2],[O_2]\}$ $+[O_2(A')]R_{d2.1-2}\{[O(^3P)],[O_2]\}$, so $P\{O_2(c)\} = [O_2(A)]R_{t3.1-3}\{[O(^3P)],[N_2],[O_2]\}+ [O_2(A')]R_{d2.1-2}\{[O(^3P)],[O_2]\}+$ $[O(^3P)]^2 R_{c1.1-2}\{[N_2],[O_2]\}$.

The loss term was calculated as follows: $L\{O_2(c)\} = L\{O_2(c)$-M$\} + L\{O_2(c)$-H$\} + L\{O_2(c)$-C$\} = [O_2(c)] \times D_c$, where

$L\{O_2(c)$-M$\} = 0$, $L\{O_2(c)$-H$\} = [O_2(c)] \times (R_{c2.1}[O(^3P)] + R_{c3.1-2}\{[O(^3P)],[O_2]\} + R_{c7.1}[O(^3P)] + R_{c8.0})$ and $L\{O_2(c)$-C$\} = [O_2(c)] \times (R_{c5.1-2}\{[O(^3P)],[O_2]\} + R_{c4.0} + R_{c6.0} + R_{c7.2}[O_2])$, so $D_c = R_{c2.1}[O(^3P)] + (R_{c3.1-2} + R_{c5.1-2} + R_{c7.1-2})\{[O(^3P)],[O_2]\} + R_{c4.0} + R_{c6.0} + R_{c8.0}$.

$[O_2(c)]$ values were retrieved taking M-, H- and C-processes into account as follows: R-$[O_2(c)] = [O_2(c)] = [O_2(c)$-M$] +$ $[O_2(c)$-H$]+[O_2(c)$-C$]$, where $[O_2(c)$-M$] = 0$, $[O_2(c)$-H$] = P\{O_2(c)$-H$\}/D_c$ and $[O_2(c)$-C$] = P\{O_2(c)$-C$\}/D_c$ so R-$[O_2(c)] =$

$P\{O_2(c)\}/D_c$.

### A3.2  Substep 2: retrieval of $[O_2(a)]$

Infrared Atmospheric band emission measured at $1.27\,\mu$m was used to retrieve VER$\{O_2(a - X)\}$ values and then to retrieve $[O(^3P)]$ values according to the continuity equation for $[O_2(a)]$, i.e. the cubic equation with respect to $[O(^3P)]$. Note that $[O_2(A)]$, $[O_2(A')]$, $[O_2(b)]$ and $[O_2(c)]$ values retrieved at the previous steps were used in the $[O(^3P)]$ retrieval at this step.

Then, $[O_2(a)]$ values were retrieved (R-$[O_2(a)]$) on the basis of $[O(^3P)]$ values by using the continuity equation considering all relevant processes of the MAC model. on the basis of VER$\{O_2(a - X)\}$ values. The continuity equation for $[O_2(a)]$ including terms of the $[O_2(a)]$ production ($P\{O_2(a)\}$) and its loss ($L\{O_2(a)\}$) is as follows: $d[O_2(a)]/dt = P\{O_2(a)\} - L\{O_2(a)\} = 0$. The production and loss terms were calculated considering the processes shown in Tables 6 ... 5.

The production term is compounded of terms related to the M-model discussed in Section 3.2.1 ($P\{O_2(a)$-M$\}$), to the H-

model discussed in Section 3.2.2 ($P\{O_2(a)$-H$\}$) and the complementary processes relevant here ($P\{O_2(a)$-C$\}$): $P\{O_2(a)\} =$ $P\{O_2(a)$-M$\} + P\{O_2(a)$-H$\} + P\{O_2(a)$-C$\}$, where $P\{O_2(a)$-M$\} = [O_2(b)]R_{b2.1-5}\{[O_3],[O(^3P)],[N_2],[O_2],[CO_2]\}$ $+ [O_2(b)]R_{b3.0} + R_{s2.3}[O_3]$, $P\{O_2(a)$-H$\} = 0$ and $P\{O_2(a)$-C$\} = [O_2(A)]R_{t6.1-3}\{[O(^3P)],[N_2],[O_2]\}$





$+ [O_2(A')](R_{d4.1-2}\{[O(^3P)], [O_2]\} + R_{d6.0}) + [O_2(c)](R_{c5.1-2}\{[O(^3P)], [O_2]\} + R_{c6.0})$

$+ [O(^3P)]^2 R_{a1.1-2}\{[N_2], [O_2]\} + [O(^1D)]R_{r2.2}[O_2] + R_{s2.1,5}[O_3]$. The production term was calculated as follows: $P\{O_2(a)\} = [O_2(A)]R_{t6.1-3}\{[O(^3P)], [N_2], [O_2]\} + [O_2(A')](R_{d4.1-2}\{[O(^3P)], [O_2]\} + R_{d6.0})$

$+ [O_2(c)](R_{c5.1-2}\{[O(^3P)], [O_2]\} + R_{c6.0}) + [O_2(b)]R_{b2.1-5}\{[O_3], [O(^3P)], [N_2], [O_2], [CO_2]\} + [O_2(b)]R_{b3.0} + R_{s2.1,3,5}[O_3]$

$+ [O(^3P)]^2 R_{a1.1-2}\{[N_2], [O_2]\} + [O(^1D)]R_{r2.2}[O_2]$.

The loss term was calculated as follows: $L\{O_2(a)\} = L\{O_2(a)\text{-M}\} + L\{O_2(a)\text{-H}\} + L\{O_2(a)\text{-C}\} = [O_2(a)] \times D_a$, where $L\{O_2(a)\text{-M}\} = [O_2(a)] \times (R_{a2.2-4}\{[O(^3P)], [N_2], [O_2]\} + R_{a4.0})$, $L\{O_2(a)\text{-H}\} = 0$ and $L\{O_2(a)\text{-C}\} = [O_2(a)] \times (R_{a2.1}[O_3])$, so $D_a = R_{a2.1-4}\{[O_3], [O(^3P)], [N_2], [O_2]\} + R_{a4.0}$.

$[O_2(a)]$ values were retrieved taking M-, H- and C-processes into account as follows: R-$[O_2(a)] = [O_2(a)] = [O_2(a)]\text{-M} +$

$[O_2(a)]\text{-H} + [O_2(a)]\text{-C}$, where $[O_2(a)]\text{-M} = P\{O_2(a)\text{-M}\}/(D_a D_c)$, $[O_2(a)]\text{-H} = P\{O_2(a)\text{-H}\}/(D_a D_c)$ and $[O_2(a)]\text{-C} = P\{O_2(a)\text{-C}\}/(D_a D_c)$, so R-$[O_2(a)] = P\{O_2(a)\}/(D_a D_c)$. In the case when Infrared Atmospheric band emissions are not given, $[O_2(a)]$ values can be retrieved on the basis of already known $[O(^3P)]$ values.

$[O_2(a)]$ values were also evaluated (E-$[O_2(a)]$) on the basis of retrieved VER$\{O_2(a - X)\}$ values (R-VER$\{O_2(a - X)\}$), so that only the respective transition probability was used: E-$[O_2(a)] = $ R-VER$\{O_2(a - X)\}/R_{a3.0}$.

Finally, VER$\{O_2(a - X)\}$ values were evaluated (E-VER$\{O_2(a - X)\}$) on the basis of retrieved $[O_2(a)]$ values and the respective transition probability: E-VER$\{O_2(a - X)\} = $ R-$[O_2(a)] \times R_{a3.0}$.

$[O_2(a)]$ values were retrieved and then evaluated to compare and verify these calculations. As for VER$\{O_2(a - X)\}$ values, they were also evaluated to compare them with retrieved given ones and so to verify calculations by using the MAC model, see Section A3.3.

## 20   A3.3   Substep 3: consistency tests in the retrieval of $[O_2(a)]$

The consistency tests in the retrievals performed with the MAC model are based on the comparison of the retrieved and evaluated values.

The retrieval step 3.1 described in Section A3.1 was carried out to retrieve R-$[O_2(c)]$ and $[O(^3P)]$ values. Calculations at the retrieval step 3.1 can not be tested for consistency because $[O_2(c)]$ was not retrieved on the basis of VER values, but con-

centrations available from the previous retrieval steps. Indeed, emissions in the Herzberg II band were not measured, whereas emissions in the New system from Keck I/II and the Richards-Johnson system are of low signal to noise ratio. Therefore, calculations at the retrieval step 3.2 only can be tested for consistency.

The retrieval step 3.2 described in Section A3.2 was carried out to retrieve R-$[O_2(a)]$ and $[O(^3P)]$ values on the basis of R-VER$\{O_2(a - X)\}$ values and concentrations of available excited chemical species. E-$[O_2(a)]$ values were also evaluated

to compare them with R-$[O_2(a)]$ values. Additionally, E-VER$\{O_2(a - X)\}$ values were also evaluated to compare them with R-VER$\{O_2(a - X)\}$ values.

## A4   The 4$^{th}$ retrieval step

The 4$^{th}$ retrieval step was performed at two substeps to calculate $[O(^1S)]$ values.



### A4.1 Substep 1: retrieval of $[\mathrm{O}(^1S)]$

Oxygen green line emission measured at 557.7 nm was used to retrieve $\mathrm{VER}\{\mathrm{O}(^1S-^1D)\}$ values and then to retrieve $[\mathrm{O}(^3P)]$ values according to the continuity equation for $[\mathrm{O}(^1S)]$, i.e. the cubic equation with respect to $[\mathrm{O}(^3P)]$. Note that $[\mathrm{O}_2(A)]$, $[\mathrm{O}_2(A')]$, $[\mathrm{O}_2(c)]$, $[\mathrm{O}_2(b)]$ and $[\mathrm{O}_2(a)]$ values retrieved at the previous steps were used in the $[\mathrm{O}(^3P)]$ retrieval at this step.

Then, $[\mathrm{O}(^1S)]$ values were retrieved (R-$[\mathrm{O}(^1S)]$) on the basis of $[\mathrm{O}(^3P)]$ values by using the continuity equation considering all relevant processes of the MAC model. The continuity equation for $[\mathrm{O}(^1S)]$ including terms of the $[\mathrm{O}(^1S)]$ production ($P\{\mathrm{O}(^1S)\}$) and its loss ($L\{\mathrm{O}(^1S)\}$) is as follows: $d[\mathrm{O}(^1S)]/dt = P\{\mathrm{O}(^1S)\} - L\{\mathrm{O}(^1S)\} = 0$. The production and loss terms were calculated considering the processes shown in Tables 6 … 5.

The production term is compounded of terms related to the M-model discussed in Section 3.2.1 ($P\{\mathrm{O}(^1S)\text{-M}\}$), to the H-
model discussed in Section 3.2.2 ($P\{\mathrm{O}(^1S)\text{-H}\}$) and the complementary processes relevant here ($P\{\mathrm{O}(^1S)\text{-C}\}$): $P\{\mathrm{O}(^1S)\} = P\{\mathrm{O}(^1S)\text{-M}\} + P\{\mathrm{O}(^1S)\text{-H}\} + P\{\mathrm{O}(^1S)\text{-C}\}$, where $P\{\mathrm{O}(^1S)\text{-M}\} = 0$, $P\{\mathrm{O}(^1S)\text{-H}\} = [\mathrm{O}(^3P)]R_{c2.1}[\mathrm{O}_2(c)]$ and $P\{\mathrm{O}(^1S)\text{-C}\} = [\mathrm{O}(^3P)](R_{t10.1}[\mathrm{O}_2(A)] + R_{d9.1}[\mathrm{O}_2(A')])$. The production term was calculated as follows: $P\{\mathrm{O}(^1S)\} = [\mathrm{O}(^3P)](R_{t10.1}[\mathrm{O}_2(A)] + R_{d9.1}[\mathrm{O}_2(A')]) + [\mathrm{O}(^3P)]R_{c2.1}[\mathrm{O}_2(c)]$.

The loss term was calculated as follows: $L\{\mathrm{O}(^1S)\} = L\{\mathrm{O}(^1S)\text{-M}\} + L\{\mathrm{O}(^1S)\text{-H}\} + L\{\mathrm{O}(^1S)\text{-C}\} = [\mathrm{O}(^1S)] \times D_g$, where
$L\{\mathrm{O}(^1S)\text{-M}\} = 0$, $L\{\mathrm{O}(^1S)\text{-H}\} = [\mathrm{O}(^1S)] \times (R_{g1.2}[\mathrm{O}_2] + R_{g3.0} + R_{g4.0})$ and $L\{\mathrm{O}(^1S)\text{-C}\} = [\mathrm{O}(^1S)] \times (R_{g1.1}[\mathrm{O}(^3P)] + R_{g1.3}[\mathrm{O}_3] + R_{g2.1-2}\{[\mathrm{N}_2],[\mathrm{O}_2(a)]\})$, so $D_g = R_{g1.1-3}\{[\mathrm{O}(^3P)],[\mathrm{O}_2],[\mathrm{O}_3]\} + R_{g2.1-2}\{[\mathrm{N}_2],[\mathrm{O}_2(a)]\} + R_{g3.0} + R_{g4.0}$.

$[\mathrm{O}(^1S)]$ values were retrieved taking M-, H- and C-processes into account as follows: R-$[\mathrm{O}(^1S)] = [\mathrm{O}(^1S)] = [\mathrm{O}(^1S)\text{-M}] + [\mathrm{O}(^1S)\text{-H}] + [\mathrm{O}(^1S)\text{-C}]$, where $[\mathrm{O}(^1S)\text{-M}] = 0$, $[\mathrm{O}(^1S)\text{-H}] = P\{\mathrm{O}(^1S)\text{-H}\}/(D_g D_c)$ and $[\mathrm{O}(^1S)\text{-C}] = 0$. In the case when oxygen green line emissions are not given, $[\mathrm{O}(^1S)]$ values can be retrieved on the basis of already known $[\mathrm{O}(^3P)]$ values.

$[\mathrm{O}(^1S)]$ values were also evaluated (E-$[\mathrm{O}(^1S)]$) on the basis of retrieved $\mathrm{VER}\{\mathrm{O}(^1S-^1D)\}$ values (R-$\mathrm{VER}\{\mathrm{O}(^1S-^1D)\}$), so that only the respective transition probability was used: E-$[\mathrm{O}(^1S)] = $ R-$\mathrm{VER}\{\mathrm{O}(^1S-^1D)\}/R_{g3.0}$.

Finally, $\mathrm{VER}\{\mathrm{O}(^1S-^1D)\}$ values were evaluated (E-$\mathrm{VER}\{\mathrm{O}(^1S-^1D)\}$) on the basis of retrieved $[\mathrm{O}_2(a)]$ values and the respective transition probability: E-$\mathrm{VER}\{\mathrm{O}(^1S-^1D)\} = $ R-$[\mathrm{O}_2(a)] \times R_{g3.0}$.

$[\mathrm{O}(^1S)]$ values were retrieved and then evaluated to compare and verify these calculations. As for $\mathrm{VER}\{\mathrm{O}(^1S-^1D)\}$
values, they were also evaluated to compare them with retrieved given ones and so to verify calculations by using the MAC model, see Section A4.2.

### A4.2 Substep 2: consistency tests in the retrieval of $[\mathrm{O}(^1S)]$

The consistency tests in the retrievals by using the MAC model is based on the comparison of the retrieved and evaluated values.

The retrieval step 4.1 described in Section A4.1 was carried out to retrieve R-$[\mathrm{O}(^1S)]$ and $[\mathrm{O}(^3P)]$ values on the basis of R-$\mathrm{VER}\{\mathrm{O}(^1S-^1D)\}$ values and concentrations of available excited chemical species. E-$[\mathrm{O}(^1S)]$ values were also evaluated to compare them with R-$[\mathrm{O}(^1S)]$ values. Additionally, E-$\mathrm{VER}\{\mathrm{O}(^1S-^1D)\}$ values were also evaluated to compare them with R-$\mathrm{VER}\{\mathrm{O}(^1S-^1D)\}$ values.





## A5    The 5$^{th}$ retrieval step

The 5$^{th}$ retrieval step was performed to calculate [O$_x$] ([O($^3P$)], [O($^1D$)] and [O$_3$]) values on the basis of concentrations of all relevant chemical species.

### A5.1    Substep 1: retrieval of [O($^3P$)] involving all relevant chemical species

[O($^3P$)] values were retrieved ([O($^3P$)-R]) on the basis of concentrations of atmospheric minor species obtained at the previous retrieval steps according to the continuity equation for [O($^3P$)]considering all relevant processes of the MAC model. For instance, values of [O$_2(A)$], [O$_2(A')$], [O$_2(b)$], [O$_2(c)$], [O$_2(a)$] and [O($^1S$)] were obtained at the retrieval steps 2.1, 2.2, 2.3, 3.1, 3.2 and 4.1, respectively.

The continuity equation for [O($^3P$)] including terms of the [O($^3P$)] production ($P\{$O($^3P$)$\}$) and loss ($L\{$O($^3P$)$\}$) is as fol-

lows: $d[$O($^3P$)$]/dt = P\{$O($^3P$)$\} - L\{$O($^3P$)$\} = 0$. The production and loss terms were calculated considering the processes shown in Tables 6 . . . 5.

The production term is compounded of terms related to the M-model discussed in Section 3.2.1 ($P\{$O($^3P$)-M$\}$), to the H-model discussed in Section 3.2.2 ($P\{$O($^3P$)-H$\}$) and the complementary processes relevant here ($P\{$O($^3P$)-C$\}$):
$P\{$O($^3P$)$\} = P\{$O($^3P$)-M$\} + P\{$O($^3P$)-H$\} + P\{$O($^3P$)-C$\}$, where $P\{$O($^3P$)-M$\} = [$O($^1D$)$]R_{r2.1,3}\{[$N$_2$], [O$_2$]$\} + (R_{s1.1-2} +$

$2R_{s1.3-4})[$O$_2$], $P\{$O($^3P$)-H$\} = [$O($^1S$)$](R_{g1.2}[$O$_2$] + R_{g4.0})$ and $P\{$O($^3P$)-C$\} = [$O$_2(b)]R_{b4.1}[$O$_3$] + [O$_2(a)]R_{a2.1}[$O$_3$]
$+ [$O($^1S$)$]R_{g2.1-2}\{[$N$_2$], [O$_2(a)$]$\} + [$O($^1D$)$](R_{r1.1}[$O($^3P$)$] + R_{r2.2,4}\{[$O$_2$], [CO$_2$]$\} + R_{r3.0} + 2R_{r1.2}[$O$_3$]) + 3R_{s2.2}[$O$_3$] + R_{s2.5-6}[$O$_3$] +$
$[$H$]R_{h6.3}[$HO$_2$]$. The production term was calculated as follows: $P\{$O($^3P$)$\} = ([$O$_2(b)]R_{b4.1} + [$O$_2(a)]R_{a2.1})[$O$_3$] + [O($^1S$)$](R_{g1.2}[$O$_2$]
$+ R_{g2.1-2}\{[$N$_2$], [O$_2(a)$]$\} + R_{g4.0}) + [$O($^1D$)$](R_{r1.1}[$O($^3P$)$] + 2R_{r1.2}[$O$_3$] + R_{r2.1-4}\{[$N$_2$], [O$_2$], [O$_2$], [CO$_2$]$\} + R_{r3.0}) + (R_{s1.1-2} +$
$2R_{s1.3-4})[$O$_2$] + 3R_{s2.2}[$O$_3$] + R_{s2.5-6}[$O$_3$] + [H]R_{h6.3}[$HO$_2$]$.

The loss term was calculated as follows: $L\{$O($^3P$)$\} = L\{$O($^3P$)-M$\} + L\{$O($^3P$)-H$\} + L\{$O($^3P$)-C$\} = [$O($^3P$)$] \times D_o$,
where $L\{$O($^3P$)-M$\} = [$O($^3P$)$] \times ([$O($^3P$)$]R_{a1.1-2}\{[$N$_2$], [O$_2$]$\})$, $L\{$O($^3P$)-H$\} = [$O($^3P$)$] \times ([$O($^3P$)$](R_{x1.1-2} + R_{c1.1-2} +$
$R_{b1.1-2})\{[$N$_2$], [O$_2$]$\} + R_{c2.1}[$O$_2(c)$])$, $L\{$O($^3P$)-C$\} = [$O($^3P$)$] \times (R_{t10.1}[$O$_2(A)] + R_{d9.1}[$O$_2(A')] + [$O($^3P$)$](R_{t1.1-2} +$
$R_{d1.1-2})\{[$N$_2$], [O$_2$]$\} + R_{x2.1}[$O$_3$] + [O$_2$]R_{x3.1-2}\{[$N$_2$], [O$_2$]$\} + R_{h2.1}[$OH$^*$] + R_{h4.1}[$HO$_2$])$, so $D_o = R_{t10.1}[$O$_2(A)] +$
$R_{d9.1}[$O$_2(A')] + [$O($^3P$)$](R_{x1.1-2} + R_{t1.1-2} + R_{d1.1-2} +$

$R_{c1.1-2} + R_{b1.1-2} + R_{a1.1-2})\{[$N$_2$], [O$_2$]$\} + R_{x2.1}[$O$_3$] + [O$_2$]R_{x3.1-2}\{[$N$_2$], [O$_2$]$\} + R_{h2.1}[$OH$^*$] + R_{h4.1}[$HO$_2$] + R_{c2.1}[$O$_2(c)]$.

[O($^3P$)] values were retrieved taking M-, H- and C-processes into account as follows: [O($^3P$)-R] = [O($^3P$)-M] + [O($^3P$)-H] +
[O($^3P$)-C], where [O($^3P$)-M] =
$([$O($^1D$)$]R_{r2.1,3}\{[$N$_2$], [O$_2$]$\} + (R_{s1.1-2} + 2R_{s1.3-4})[$O$_2$]) / D_o$, [O($^3P$)-H] $= ([$O($^1S$)$](R_{g1.2}[$O$_2$] + R_{g4.0})) / D_o$ and [O($^3P$)-C] =
$([$O$_2(b)]R_{b4.1}[$O$_3$] + [O$_2(a)]R_{a2.1}[$O$_3$] + [O($^1S$)$]R_{g2.1-2}\{[$N$_2$], [O$_2(a)$]$\} + 3R_{s2.2}[$O$_3$] + R_{s2.5-6}[$O$_3$]) / D_o$

$+ ([$O($^1D$)$](R_{r1.1}[$O($^3P$)$] + R_{r2.2,4}\{[$O$_2$], [CO$_2$]$\} + R_{r3.0} + 2R_{r1.2}[$O$_3$]) + [H]R_{h6.3}[$HO$_2$]) / D_o$.

The final equation for [O($^3P$)] is as follows: [O($^3P$)-R] = [O($^3P$)] $= ([$O$_2(a)]R_{a2.1}[$O$_3$] + [O$_2(b)]R_{b4.1}[$O$_3$]) / D_o$
$+ ([$O($^1S$)$](R_{g1.2}[$O$_2$] + R_{g2.1-2}\{[$N$_2$], [O$_2(a)$]$\} + R_{g4.0})) / D_o$



$+ \left([\mathrm{O}(^1D)](R_{r1.1}[\mathrm{O}(^3P)] + R_{r2.1-4}\{[\mathrm{N}_2],[\mathrm{O}_2],[\mathrm{O}_2],[\mathrm{CO}_2]\} + R_{r3.0} + 2R_{r1.2}[\mathrm{O}_3])\right)/D_o$

$+ \left((R_{s1.1-2} + 2R_{s1.3-4})[\mathrm{O}_2] + 3R_{s2.2}[\mathrm{O}_3] + R_{s2.5-6}[\mathrm{O}_3] + [\mathrm{H}]R_{h6.3}[\mathrm{HO}_2]\right)/D_o.$

## A5.2 Substep 2: retrieval of $[\mathrm{O}(^1D)]$ involving all relevant chemical species

$[\mathrm{O}(^1D)]$ values were retrieved ($[\mathrm{O}(^1D)]$-R]) on the basis of concentrations of atmospheric minor species obtained at the previ-

ous retrieval steps according to the continuity equation for $[\mathrm{O}(^1D)]$ considering all relevant processes of the MAC model.

The continuity equation for $[\mathrm{O}(^1D)]$ including terms of the $[\mathrm{O}(^1D)]$ production ($P\{\mathrm{O}(^1D)\}$) and loss ($L\{\mathrm{O}(^1D)\}$) is as follows: $d[\mathrm{O}(^1D)]/dt = P\{\mathrm{O}(^1D)\} - L\{\mathrm{O}(^1D)\} = 0$.

The production and loss terms were calculated considering the processes shown in Tables 5, 6 and 7.

The calculation of the production term was based on the considered M-, H- and C-processes as follows: $P\{\mathrm{O}(^1D)\} =$

$P\{\mathrm{O}(^1D)\text{-M}\} + P\{\mathrm{O}(^1D)\text{-H}\} + P\{\mathrm{O}(^1D)\text{-C}\}$, where $P\{\mathrm{O}(^1D)\text{-M}\} = R_{s1.1-2}[\mathrm{O}_2] + R_{s2.3}[\mathrm{O}_3]$,
$P\{\mathrm{O}_2(^1D)\text{-H}\} = R_{g3.0}[\mathrm{O}(^1S)]$ and $P\{\mathrm{O}_2(^1D)\text{-C}\} = [\mathrm{O}(^1S)]2R_{g1.1}[\mathrm{O}(^3P)] + R_{s2.4}[\mathrm{O}_3]$, so $P\{\mathrm{O}(^1D)\} = [\mathrm{O}(^1S)](2R_{g1.1}[\mathrm{O}(^3P)]+$
$R_{g3.0}) + R_{s1.1-2}[\mathrm{O}_2] + R_{s2.3-4}[\mathrm{O}_3]$.

The calculation of the loss term was based on the considered M-, H- and C-processes as follows: $L\{\mathrm{O}(^1D)\} = L\{\mathrm{O}(^1D)\text{-M}\}+$
$L\{\mathrm{O}(^1D)\text{-H}\} + L\{\mathrm{O}(^1D)\text{-C}\} = [\mathrm{O}(^1D)] \times D_r$, where $L\{\mathrm{O}(^1D)\text{-M}\} = R_{r2.1,3}\{[\mathrm{N}_2],[\mathrm{O}_2]\}$, $L\{\mathrm{O}(^1D)\text{-H}\} = 0$ and

$L\{\mathrm{O}(^1D)\text{-C}\} = R_{r1.1-3}\{[\mathrm{O}(^3P)],[\mathrm{O}_3],[\mathrm{O}_3]\} + R_{r2.2,4}\{[\mathrm{O}_2],[\mathrm{CO}_2]\} + R_{r3.0}$, so $D_r = R_{r1.1-3}\{[\mathrm{O}(^3P)],[\mathrm{O}_3],[\mathrm{O}_3]\}+$
$R_{r2.1-4}\{[\mathrm{N}_2],[\mathrm{O}_2],[\mathrm{O}_2],[\mathrm{CO}_2]\} + R_{r3.0}$.

$[\mathrm{O}(^1D)]$ values were retrieved taking M-, H- and C-processes into account as follows: $[\mathrm{O}(^1D)]\text{-R}] = [\mathrm{O}(^1D)] = [\mathrm{O}(^1D)\text{-M}]+$
$[\mathrm{O}(^1D)\text{-H}] + [\mathrm{O}(^1D)\text{-C}]$, where $[\mathrm{O}(^1D)\text{-M}] = (R_{s1.1-2}[\mathrm{O}_2] + R_{s2.3}[\mathrm{O}_3])/D_r$, $[\mathrm{O}(^1D)\text{-H}] = (R_{g3.0}[\mathrm{O}(^1S)])/D_r$ and
$[\mathrm{O}(^1D)\text{-C}] = ([\mathrm{O}(^1S)]2R_{g1.1}[\mathrm{O}(^3P)])/D_r$.

The final equation for $[\mathrm{O}(^1D)]$ is as follows: $[\mathrm{O}(^1D)\text{-R}] = ((2R_{g1.1}[\mathrm{O}(^3P)] + R_{g3.0})[\mathrm{O}(^1S)] + R_{s1.1-2}[\mathrm{O}_2] + R_{s2.3-4}[\mathrm{O}_3])/D_r$.

## A5.3 Substep 3: retrieval of $[\mathrm{O}_3]$ involving all relevant chemical species

$[\mathrm{O}_3]$ values were retrieved ($[\mathrm{O}_3\text{-R}]$) on the basis of concentrations of atmospheric minor species obtained at the previous retrieval steps according to the continuity equation for $[\mathrm{O}_3]$ considering all relevant processes of the MAC model.

The continuity equation for $[\mathrm{O}_3]$ including terms of the $[\mathrm{O}_3]$ production ($P\{\mathrm{O}_3\}$) and loss ($L\{\mathrm{O}_3\}$) is as follows: $d[\mathrm{O}_3]/dt =$

$P\{\mathrm{O}_3\} - L\{\mathrm{O}_3\} = 0$.

The production and loss terms were calculated considering the processes shown in Tables 5, 6 and 7.

The calculation of the production term was based on the considered M-, H- and C-processes as follows: $P\{\mathrm{O}_3\} = P\{\mathrm{O}_3\text{-M}\}+$
$P\{\mathrm{O}_3\text{-H}\} + P\{\mathrm{O}_3\text{-C}\}$, where $P\{\mathrm{O}_3\text{-M}\} = 0$, $P\{\mathrm{O}_3\text{-H}\} = 0$ and $P\{\mathrm{O}_3\text{-C}\} = P\{\mathrm{O}_3\} = [\mathrm{O}(^3P)][\mathrm{O}_2]R_{x3.1-2}\{[\mathrm{N}_2],[\mathrm{O}_2]\}$.

The calculation of the loss term was based on the considered M-, H- and C-processes as follows: $L\{\mathrm{O}_3\} = L\{\mathrm{O}_3\text{-M}\}+$

$L\{\mathrm{O}_3\text{-H}\} + L\{\mathrm{O}_3\text{-C}\} = [\mathrm{O}_3] \times D_3$, where $L\{\mathrm{O}_3\text{-M}\} = R_{s2.3}$, $L\{\mathrm{O}_3\text{-H}\} = 0$, $L\{\mathrm{O}_3\text{-C}\} = R_{x2.1}[\mathrm{O}(^3P)] + R_{b4.1}[\mathrm{O}_2(b)]+$
$R_{a2.1}[\mathrm{O}_2(a)] + R_{g1.3}[\mathrm{O}(^1S)] + R_{r1.2-3}[\mathrm{O}(^1D)] + R_{h1.1}[\mathrm{H}] + R_{h3.1}[\mathrm{OH}^*] + R_{s2.1-2,4-6}$, so $D_3 = R_{x2.1}[\mathrm{O}(^3P)] + R_{b4.1}[\mathrm{O}_2(b)]+$
$R_{a2.1}[\mathrm{O}_2(a)] + R_{g1.3}[\mathrm{O}(^1S)] + R_{r1.2-3}[\mathrm{O}(^1D)] + R_{s2.1-6} + R_{h1.1}[\mathrm{H}] + R_{h3.1}[\mathrm{OH}^*]$.





[$O_3$] values were retrieved taking M-, H- and C-processes into account as follows: $[O_3\text{-R}] = [O_3\text{-M}] + [O_3\text{-H}] + [O_3\text{-C}]$,
where $[O_3\text{-M}] = 0$, $[O_3\text{-H}] = 0$ and $[O_3\text{-C}] = \left([O(^3P)][O_2]R_{x3.1-2}\{[N_2],[O_2]\}\right)/D_3$.

The final equation for [$O_3$] is as follows: $[O_3\text{-R}] = [O_3] = \left([O(^3P)][O_2]R_{x3.1-2}\{[N_2],[O_2]\}\right)/D_3$.

*Author contributions.* Olexandr Lednyts'kyy worked out the concept of the MAC approach proposed by Torr *et al.* (1985), developed corre-
5  sponding software, performed needed computations and prepared the manuscript of the article. Christian von Savigny contributed to planning
the work activities regarding the article, discussed the results, contributed to the manuscript of the article, corrected and edited it.

*Competing interests.* The authors declare that they have no conflict of interests.

*Acknowledgements.* The authors acknowledge the financial support provided by the German Research Foundation (German: DFG) through
the grant SA 1351/6-1 and thank Dr. Sinnhuber and Dr. Fytterer for the corresponding helpful discussions. The authors acknowledge a
10  positive stimulating influence of Edward Llewellyn on working out the doctoral thesis by Olexandr Lednyts'kyy under the supervision of
Christian von Savigny. Olexandr Lednyts'kyy also acknowledges the financial support provided by the University of Greifswald and the
International Helmholtz Graduate School for Plasma Physics.



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





**Table 9.** Rate values of processes considered in the MAC model involving $O_2(A)$ and $O_2(A')$. These reaction rates are followed by those shown in Tables 10, 11 and 12. Processes of the provided here rate values are shown in Table 5. References: r01 - Smith and Robertson (2008), r02 - Bates (1988a), r03 - Lednyts′kyy and von Savigny (2016), r04 - Rodrigo et al. (1988), r05 - Bates (1988b), r06 - Krasnopolsky (2011), r07 - Kenner and Ogryzlo (1984), r08 - Stegman and Murtagh (1991), r09 - López-González et al. (1992a).

| R$_\#$ | $\Delta$H (eV) | Rate value | Rate unit | Ref. |
|---|---|---|---|---|
| $R_{t1.1}$ | | $\theta_{N2}^{Pt} = tY \cdot 3 \cdot 10^{-33}(300/T)^{3.25}$ | molec$^{-2}$ cm$^6$ s$^{-1}$ | r01 |
| $R_{t1.2}$ | | $\theta_{O2}^{Pt} = tY \cdot 3 \cdot 10^{-33}(300/T)^{3.25}$ | molec$^{-2}$ cm$^6$ s$^{-1}$ | r01 |
| | | $tY = 0.06$ | 1 | r02 |
| $R_{t2.1-3}$ | | $\theta_{3P}^{td} = cTDu \cdot \theta_{3P}^{tx}, \theta_{N2}^{td} = cTDu \cdot \theta_{N2}^{tx}, \theta_{O2}^{td} = cTDu \cdot \theta_{O2}^{tx}$ | molec$^{-1}$ cm$^3$ s$^{-1}$ | r03 |
| | | cTDu $= 1 \cdot 10^{-2}$ as cTDu $\in [1 \cdot 10^{-30}, 1 \cdot 10^{30}]$ | 1 | r03 |
| $R_{t3.1-3}$ | | $\theta_{3P}^{tc} = cTCu \cdot \theta_{3P}^{tx}, \theta_{N2}^{tc} = cTCu \cdot \theta_{N2}^{tx}, \theta_{O2}^{tc} = cTCu \cdot \theta_{O2}^{tx}$ | molec$^{-1}$ cm$^3$ s$^{-1}$ | r03 |
| | | cTCu $= 1 \cdot 10^{-2}$ as cTCu $\in [1 \cdot 10^{-30}, 1 \cdot 10^{-2}]$ | 1 | r03 |
| $R_{t4.1-3}$ | | $\theta_{3P}^{tb} = cTBu \cdot \theta_{3P}^{tx}, \theta_{N2}^{tb} = cTBu \cdot \theta_{N2}^{tx}, \theta_{O2}^{tb} = cTBu \cdot \theta_{O2}^{tx}$ | molec$^{-1}$ cm$^3$ s$^{-1}$ | r03 |
| | | cTBu $= 1 \cdot 10^{-2}$ as cTBu $\in [1 \cdot 10^{-30}, 1 \cdot 10^{-2}]$ | 1 | r03 |
| $R_{t5.0}$ | | $\theta_{BG}^{A} = 0.13$ | s$^{-1}$ | r04 |
| $R_{t6.1-3}$ | | $\theta_{3P}^{ta} = cTAu \cdot \theta_{3P}^{tx}, \theta_{N2}^{ta} = cTAu \cdot \theta_{N2}^{tx}, \theta_{O2}^{ta} = cTAu \cdot \theta_{O2}^{tx}$ | molec$^{-1}$ cm$^3$ s$^{-1}$ | r03 |
| | | cTAu $= 1 \cdot 10^{-2}$ as cTAu $\in [1 \cdot 10^{-30}, 1 \cdot 10^{-2}]$ | 1 | r03 |
| $R_{t7.1}$ | | $\theta_{3P}^{tx} = 1.3 \cdot 10^{-11}$ | molec$^{-1}$ cm$^3$ s$^{-1}$ | r07 |
| $R_{t7.2}$ | | $\theta_{N2}^{tx} = 1.2 \cdot 10^{-11}$ | molec$^{-1}$ cm$^3$ s$^{-1}$ | r03 |
| $R_{t7.3}$ | | $\theta_{O2}^{tx} = 1.3 \cdot 10^{-13}$ | molec$^{-1}$ cm$^3$ s$^{-1}$ | r06 |
| $R_{t8.0}$ | | $\theta_{320n}^{A} = 11$ | s$^{-1}$ | r08 |
| $R_{t9.0}$ | | $\theta_{HI}^{A} = 11$ | s$^{-1}$ | r05 |
| $R_{t10.1}$ | | $\theta_{1S}^{tx} = 1 \cdot 10^{-14}$ as $\theta_{1S}^{tx} \in [1 \cdot 10^{-30}, 1 \cdot 10^{-14}]$ | molec$^{-1}$ cm$^3$ s$^{-1}$ | r03 |
| $R_{d1.1}$ | | $\delta_{N2}^{Pd} = dY \cdot 3 \cdot 10^{-33}(300/T)^{3.25}$ | molec$^{-2}$ cm$^6$ s$^{-1}$ | r01 |
| $R_{d1.2}$ | | $\delta_{O2}^{Pd} = dY \cdot 3 \cdot 10^{-33}(300/T)^{3.25}$ | molec$^{-2}$ cm$^6$ s$^{-1}$ | r01 |
| | | $dY = 0.18$ | 1 | r02 |
| $R_{d2.1-2}$ | | $\delta_{3P}^{dc} = cDCu \cdot \delta_{3P}^{tx}, \delta_{O2}^{dc} = cDCu \cdot \delta_{O2}^{tx}$ | molec$^{-1}$ cm$^3$ s$^{-1}$ | r03 |
| | | cDCu $= 1 \cdot 10^{-2}$ close to cDCu $\in [1 \cdot 10^{-30}, 1 \cdot 10^{-3}]$ | 1 | r03 |
| $R_{d3.1-2}$ | | $\delta_{3P}^{db} = cDBu \cdot \delta_{3P}^{tx}, \delta_{O2}^{db} = cDBu \cdot \delta_{O2}^{tx}$ | molec$^{-1}$ cm$^3$ s$^{-1}$ | r03 |
| | | cDBu $= 1 \cdot 10^{-2}$ as cDBu $\in [1 \cdot 10^{-30}, 1 \cdot 10^{-2}]$ | 1 | r03 |
| $R_{d4.1-2}$ | | $\delta_{3P}^{da} = cDAu \cdot \delta_{3P}^{dx}, \delta_{O2}^{da} = cDAu \cdot \delta_{O2}^{dx}$ | molec$^{-1}$ cm$^3$ s$^{-1}$ | r03 |
| | | cDAu $= 1 \cdot 10^{-2}$ as cDAu $\in [1 \cdot 10^{-30}, 1 \cdot 10^{-2}]$ | 1 | r03 |
| $R_{d5.0}$ | | $\delta_{370n}^{A} = 0.85$ | s$^{-1}$ | r08 |
| $R_{d6.0}$ | | $\delta_{Cha}^{A} = 0.85$ | s$^{-1}$ | r05 |
| $R_{d7.1}$ | | $\delta_{3P}^{dx} = 1.3 \cdot 10^{-11}$ | molec$^{-1}$ cm$^3$ s$^{-1}$ | r06 |
| $R_{d7.2}$ | | $\delta_{O2}^{dx} = 1.7 \cdot 10^{-11}$ | molec$^{-1}$ cm$^3$ s$^{-1}$ | r09 |
| $R_{d8.0}$ | | $\delta_{HIII}^{A} = 0.9$ | s$^{-1}$ | r05 |
| $R_{d9.1}$ | | $\delta_{1S}^{dx} = 1 \cdot 10^{-14}$ as $\delta_{1S}^{dx} \in [1 \cdot 10^{-30}, 1 \cdot 10^{-14}]$ | molec$^{-1}$ cm$^3$ s$^{-1}$ | r03 |





**Table 10.** Rate values of processes are related to those shown in Table 9 and followed by those shown in Tables 11 and 12. Processes of the provided here rate values are shown in Table 6. References: r10 - Predoi-Cross et al. (2008), r11 - Slanger (1978), r12 - Kenner and Ogryzlo (1983), r13 - Burkholder et al. (2015), r14 - Minaev and Ågren (1997). Labels r01, r02, r03, r06, r08 were used in Table 9. The enthalpy change ($\Delta H$) was determined at standard temperature and pressure, see Table 12 for abbreviations.

| R# | $\Delta H$ (eV) | Rate value | Rate unit | Ref. |
|---|---|---|---|---|
| $R_{c1.1-2}$ | | $\varsigma_{N2}^{Pc} = \varsigma_{O2}^{Pc} = cY \cdot 3 \cdot 10^{-33}(300/T)^{3.25}$ | $\mathrm{molec^{-2}\,cm^6\,s^{-1}}$ | r01 |
| | | $cY = 0.04$ | 1 | r02 |
| $R_{c2.1}$ | | $\varsigma_{1S}^{cx} = 1.4 \cdot 10^{-8}$ | $\mathrm{molec^{-1}\,cm^3\,s^{-1}}$ | r03 |
| $R_{c3.1}$ | | $\varsigma_{3P}^{cb} = cCBa \cdot \varsigma_{3P}^{cx}$ | $\mathrm{molec^{-1}\,cm^3\,s^{-1}}$ | r03 |
| $R_{c3.2}$ | | $\varsigma_{O2}^{cb} = cCBm \cdot \varsigma_{O2}^{cx}$ | $\mathrm{molec^{-1}\,cm^3\,s^{-1}}$ | r03 |
| | | $cCBa = 5.8 \cdot 10^4$ | 1 | r03 |
| | | $cCBm = 1 \cdot 10^{-1}$ as $cCBm \in [1 \cdot 10^{-30}, 1 \cdot 10^{-1}]$ | 1 | r03 |
| $R_{c4.0}$ | | $\varsigma_{cbK}^{A} = \varsigma_{RJ}^{A}/10$ | $\mathrm{s^{-1}}$ | r03 |
| $R_{c5.1}$ | | $\varsigma_{3P}^{ca} = cCAa \cdot \varsigma_{3P}^{cx}$ | $\mathrm{molec^{-1}\,cm^3\,s^{-1}}$ | r03 |
| $R_{c5.2}$ | | $\varsigma_{O2}^{ca} = cCAm \cdot \varsigma_{O2}^{cx}$ | $\mathrm{molec^{-1}\,cm^3\,s^{-1}}$ | r03 |
| | | $cCAa = 1 \cdot 10^{-1}$ close to $cCAa \in [1 \cdot 10^{-30}, 1 \cdot 10^{+3}]$ | 1 | r03 |
| | | $cCAm = 1 \cdot 10^{-1}$ close to $cCAm \in [1 \cdot 10^{-30}, 1]$ | 1 | r03 |
| $R_{c6.0}$ | | $\varsigma_{RJ}^{A} = 0.073$ | $\mathrm{s^{-1}}$ | r11 |
| $R_{c7.1}$ | | $\varsigma_{3P}^{cx} = 6 \cdot 10^{-12}$ | $\mathrm{molec^{-1}\,cm^3\,s^{-1}}$ | r12 |
| $R_{c7.2}$ | | $\varsigma_{O2}^{cx} = 1.8 \cdot 10^{-11}$ | $\mathrm{molec^{-1}\,cm^3\,s^{-1}}$ | r06 |
| $R_{c8.0}$ | | $\varsigma_{HII}^{A} = 0.66$ | $\mathrm{s^{-1}}$ | r08 |
| $R_{b1.1-2}$ | -3.49[E] | $\beta_{N2}^{Pb} = \beta_{O2}^{Pb} = bY \cdot 3 \cdot 10^{-33}(300/T)^{3.25}$ | $\mathrm{molec^{-2}\,cm^6\,s^{-1}}$ | r01 |
| | | $bY = 0.03 + pY \cdot 0.07$ | 1 | r03 |
| | | $pY = 0.5$ (for $O_2(^5\Pi)$) | 1 | r02 |
| $R_{b2.1}$ | -0.65[A] | $\beta_{O3}^{ba} = 0.15 \cdot 3.5 \cdot 10^{-11}\exp(-135/T)$ | $\mathrm{molec^{-1}\,cm^3\,s^{-1}}$ | r13 |
| $R_{b2.2}$ | -0.65[E] | $\beta_{3P}^{ba} = cBAa \cdot \beta_{3P}^{bx}$ | $\mathrm{molec^{-1}\,cm^3\,s^{-1}}$ | r03 |
| $R_{b2.3}$ | -0.65[A] | $\beta_{N2}^{ba} = cBAm \cdot \beta_{N2}^{bx}$ | $\mathrm{molec^{-1}\,cm^3\,s^{-1}}$ | r03 |
| $R_{b2.4}$ | -0.65[A] | $\beta_{O2}^{ba} = cBAm \cdot \beta_{O2}^{bx}$ | $\mathrm{molec^{-1}\,cm^3\,s^{-1}}$ | r03 |
| $R_{b2.5}$ | -0.65[E] | $\beta_{C2}^{ba} = cBAm \cdot \beta_{C2}^{bx}$ | $\mathrm{molec^{-1}\,cm^3\,s^{-1}}$ | r03 |
| | | $cBAa = cBAm = 1 \cdot 10^{-1}$ as $cBAa, cBAm \in [1 \cdot 10^{-30}, 1 \cdot 10^{-1}]$ | 1 | r03 |
| $R_{b3.0}$ | 0.65[E] | $\beta_{Nox}^{A} = 0.0014$ | $\mathrm{s^{-1}}$ | r14 |
| $R_{b4.1}$ | -1.63[A] | $\beta_{O3}^{bx} = 0.7 \cdot 3.5 \cdot 10^{-11}\exp(-135/T)$ | $\mathrm{molec^{-1}\,cm^3\,s^{-1}}$ | r13 |
| $R_{b4.2}$ | -1.63[E] | $\beta_{3P}^{bx} = 8 \cdot 10^{-14}$ | $\mathrm{molec^{-1}\,cm^3\,s^{-1}}$ | r13 |
| $R_{b4.3}$ | -1.63[A] | $\beta_{N2}^{bx} = 1.8 \cdot 10^{-15}\exp(45/T)$ | $\mathrm{molec^{-1}\,cm^3\,s^{-1}}$ | r13 |
| $R_{b4.4}$ | -1.63[A] | $\beta_{O2}^{bx} = 3.9 \cdot 10^{-17}$ | $\mathrm{molec^{-1}\,cm^3\,s^{-1}}$ | r13 |
| $R_{b4.5}$ | -1.63[E] | $\beta_{C2}^{bx} = 4.2 \cdot 10^{-13}$ | $\mathrm{molec^{-1}\,cm^3\,s^{-1}}$ | r13 |
| $R_{b4.6}$ | -1.63[A] | $\beta_{O3}^{bx} = 0.15 \cdot 3.5 \cdot 10^{-11}\exp(-135/T)$ | $\mathrm{molec^{-1}\,cm^3\,s^{-1}}$ | r13 |
| $R_{b5.0}$ | 1.63[E] | $\beta_{762}^{A} = 0.079$ | $\mathrm{s^{-1}}$ | r08 |
| $R_{b6.0}$ | 1.63[E] | $\beta_{Atm}^{A} = 0.083$ | $\mathrm{s^{-1}}$ | r10 |





**Table 11.** Rate values of processes are related to those shown in Tables 9 and 10 and followed by those shown in Table 12. Processes of the provided here rate values are shown in Tables 6 and 7. References: r15 - Pendleton et al. (1996), r16 - Krauss and Neumann (1975), r17 - Capetanakis et al. (1993), r18 - Gordiets et al. (1995), r19 - Atkinson and Welge (1972), r20 - Kenner and Ogryzlo (1982), r21 - Kramida et al. (2015), r22 - Pinheiro et al. (1998), r23 - Sakai et al. (2014). Labels r01, r02, r03 were used in Table 9, and labels r13, r14 were used in Table 10. The enthalpy change ($\Delta H$) was determined at standard temperature and pressure, see Table 12 for abbreviations.

| $R_\#$ | $\Delta H$ (eV) | Rate value | Rate unit | Ref. |
|---|---|---|---|---|
| $R_{a1.1}$ | -4.14[E] | $\alpha^{\mathrm{Pa}}_{\mathrm{N2}} = aY \cdot 3 \cdot 10^{-33}(300/T)^{3.25}$ | $\mathrm{molec}^{-2}\,\mathrm{cm}^6\,\mathrm{s}^{-1}$ | r01 |
| $R_{a1.2}$ | -4.14[E] | $\alpha^{\mathrm{Pa}}_{\mathrm{O2}} = aY \cdot 3 \cdot 10^{-33}(300/T)^{3.25}$ | $\mathrm{molec}^{-2}\,\mathrm{cm}^6\,\mathrm{s}^{-1}$ | r01 |
| | | $aY = 0.07 + pY \cdot 0.68$ | 1 | r03 |
| | | $pY = 0.5$ (for $\mathrm{O}_2(^5\Pi)$) | 1 | r02 |
| $R_{a2.1}$ | 0.13[A] | $\alpha^{\mathrm{ax}}_{\mathrm{O3}} = 5.2 \cdot 10^{-11}\exp(-2840/T)$ | $\mathrm{molec}^{-1}\,\mathrm{cm}^3\,\mathrm{s}^{-1}$ | r13 |
| $R_{a2.2}$ | -0.98[E] | $\alpha^{\mathrm{ax}}_{\mathrm{3P}} = \mathrm{cAXa} \cdot 2 \cdot 10^{-16}$ | $\mathrm{molec}^{-1}\,\mathrm{cm}^3\,\mathrm{s}^{-1}$ | r13 |
| | | $\mathrm{cAXa} = 1 \cdot 10^{-2}$ as $\mathrm{cAXa} \in [1 \cdot 10^{-30}, 1 \cdot 10^{-2}]$ | 1 | r03 |
| $R_{a2.3}$ | -0.98[A] | $\alpha^{\mathrm{ax}}_{\mathrm{N2}} = 1 \cdot 10^{-20}$ | $\mathrm{molec}^{-1}\,\mathrm{cm}^3\,\mathrm{s}^{-1}$ | r13 |
| $R_{a2.4}$ | -0.98[A] | $\alpha^{\mathrm{ax}}_{\mathrm{O2}} = 3.6 \cdot 10^{-18}\exp(-220/T)$ | $\mathrm{molec}^{-1}\,\mathrm{cm}^3\,\mathrm{s}^{-1}$ | r13 |
| $R_{a3.0}$ | 0.98[E] | $\alpha^{\mathrm{A}}_{\mathrm{1u27}} = 2.8 \cdot 10^{-4}$ | $\mathrm{molec}^{-1}\,\mathrm{cm}^3\,\mathrm{s}^{-1}$ | r15 |
| $R_{a4.0}$ | 0.98[E] | $\alpha^{\mathrm{A}}_{\mathrm{IRA}} = 1.9 \cdot 10^{-4}$ | $\mathrm{s}^{-1}$ | r14 |
| $R_{g1.1}$ | -2.20[E] | $\gamma^{\mathrm{SP}}_{\mathrm{1D}} = 2 \cdot 10^{-14}$ | $\mathrm{molec}^{-1}\,\mathrm{cm}^3\,\mathrm{s}^{-1}$ | r16, r20 |
| $R_{g1.2}$ | -4.17[E] | $\gamma^{\mathrm{SP}}_{\mathrm{O2}} = 2.32 \cdot 10^{-12}\exp(-811.88/T + 0.001816 \cdot T)$ | $\mathrm{molec}^{-1}\,\mathrm{cm}^3\,\mathrm{s}^{-1}$ | r17 |
| $R_{g1.3}$ | -6.26[E] | $\gamma^{\mathrm{SP}}_{\mathrm{O3}} = 6 \cdot 10^{-10}$ | $\mathrm{molec}^{-1}\,\mathrm{cm}^3\,\mathrm{s}^{-1}$ | r18 |
| $R_{g2.1}$ | -4.17[E] | $\gamma^{\mathrm{SP}}_{\mathrm{N2}} = 5 \cdot 10^{-17}$ | $\mathrm{molec}^{-1}\,\mathrm{cm}^3\,\mathrm{s}^{-1}$ | r19 |
| $R_{g2.2}$ | -4.17[E] | $\gamma^{\mathrm{SP}}_{\mathrm{Oa}} = 2.6 \cdot 10^{-10}$ | $\mathrm{molec}^{-1}\,\mathrm{cm}^3\,\mathrm{s}^{-1}$ | r20 |
| $R_{g3.0}$ | 2.20[E] | $\gamma^{\mathrm{A}}_{\mathrm{557n7}} = 1.26$ | $\mathrm{s}^{-1}$ | r21 |
| $R_{g4.0}$ | 4.17[E] | $\gamma^{\mathrm{A}}_{\mathrm{1S3Pe}} = A295n8 + A297n$ | $\mathrm{s}^{-1}$ | r21 |
| | | $A295n8 = 2.42 \cdot 10^{-4}$ | $\mathrm{s}^{-1}$ | r21 |
| | | $A297n2 = 7.54 \cdot 10^{-2}$ | $\mathrm{s}^{-1}$ | r21 |
| $R_{r1.1}$ | -1.97[E] | $\rho^{\mathrm{DP}}_{\mathrm{3P}} = 8 \cdot 10^{-12}$ | $\mathrm{molec}^{-1}\,\mathrm{cm}^3\,\mathrm{s}^{-1}$ | r22 |
| $R_{r1.2}$ | -0.86[A] | $\chi^{\mathrm{DP}}_{\mathrm{2P}} = 1.2 \cdot 10^{-10}$ | $\mathrm{molec}^{-1}\,\mathrm{cm}^3\,\mathrm{s}^{-1}$ | r13 |
| $R_{r1.3}$ | -6.03[A] | $\rho^{\mathrm{DP}}_{\mathrm{O2}} = 1.2 \cdot 10^{-10}$ | $\mathrm{molec}^{-1}\,\mathrm{cm}^3\,\mathrm{s}^{-1}$ | r13 |
| $R_{r2.1}$ | -1.97[A] | $\rho^{\mathrm{DP}}_{\mathrm{N2}} = 2.15 \cdot 10^{-11}\exp(110/T)$ | $\mathrm{molec}^{-1}\,\mathrm{cm}^3\,\mathrm{s}^{-1}$ | r13 |
| $R_{r2.2}$ | -0.99[A] | $\rho^{\mathrm{DP}}_{\mathrm{Oa}} = 0.2 \cdot 3.3 \cdot 10^{-11}\exp(55/T)$ | $\mathrm{molec}^{-1}\,\mathrm{cm}^3\,\mathrm{s}^{-1}$ | r13 |
| $R_{r2.3}$ | -0.34[A] | $\rho^{\mathrm{DP}}_{\mathrm{Ob}} = 0.8 \cdot 3.3 \cdot 10^{-11}\exp(55/T)$ | $\mathrm{molec}^{-1}\,\mathrm{cm}^3\,\mathrm{s}^{-1}$ | r13 |
| $R_{r2.4}$ | -1.97[E] | $\rho^{\mathrm{DP}}_{\mathrm{C2}} = 7.5 \cdot 10^{-11}\exp(115/T)$ | $\mathrm{molec}^{-1}\,\mathrm{cm}^3\,\mathrm{s}^{-1}$ | r13 |
| $R_{r3.0}$ | 1.97[E] | $\rho^{\mathrm{A}}_{\mathrm{1D3Pe}} = A630n0 + A636n4$ | $\mathrm{s}^{-1}$ | r23 |
| | | $A630n0 = 5.63 \cdot 10^{-3}$ | $\mathrm{s}^{-1}$ | r23 |
| | | $A636n4 = 1.82 \cdot 10^{-3}$ | $\mathrm{s}^{-1}$ | r23 |





**Table 12.** Rate values of processes are related to those shown in Tables 9, 10 and 11. Processes of the provided here rate values are shown in Table 7. References: r24 - Nicolet (1971), r25 - Nicolet et al. (1989), r26 - Nicolet and Kennes (1988), r27 - Nicolet (1989), r28 - Mlynczak et al. (1993), r29 - Atkinson et al. (1997), r30 - Khomich et al. (2008).

Labels r01, r02, r03 were used in Table 9, and the label r13 was used in Table 10. The exothermic reaction energy content was determined for each reaction at standard temperature and pressure, see column "ΔH" for the enthalpy change. ΔH values were read out in the units of eV from Roble (2013), they are marked with a character R, and in the units of $kJ\,mol^{-1}$ from Atkinson et al. (1997), they are marked with a character A. Additionally, ΔH values were evaluated, they are marked with a character E.

| $R_\#$ | ΔH (eV) | Rate value | Rate unit | Ref. |
|---|---|---|---|---|
| $R_{s1.1}$ | 8.98[A] | $\sigma_{PS}^{UV} = 3\cdot 10^{-9}$ (Day: $\lambda <$132 nm) | $s^{-1}$ | r03 |
| $R_{s1.2}$ | 6.83[A] | $\sigma_{PD}^{LA} = 3\cdot 10^{-9}$ (Day: Lyman-$\alpha$ emission) | $s^{-1}$ | r24 |
| $R_{s1.3}$ | 6.83[A] | $\sigma_{PD}^{Sc} = 3.7\cdot 10^{-7}$ (Day: Schumann-Runge cont.) | $s^{-1}$ | r24 |
| $R_{s1.4}$ | 4.94[A] | $\sigma_{PP}^{Sb} = 1.25\cdot 10^{-7}$ (Day: Schumann-Runge B.) | $s^{-1}$ | r25 |
| $R_{s1.5}$ | 4.94[A] | $\sigma_{PP}^{Hc} = 5.8\cdot 10^{-10}$ (Day: Herzberg continuum) | $s^{-1}$ | r26 |
| $R_{s2.1}$ | | $\sigma_{aS}^{UV} = 2.5\cdot 10^{-3}$ (Day: $\lambda$=193 nm) | $s^{-1}$ | r13 |
| $R_{s2.2}$ | 5.95[A] | $\sigma_{PP}^{Ha} = 1\cdot 10^{-2}$ (Day: Hartley bands) | $s^{-1}$ | r03 |
| $R_{s2.3}$ | 3.86[A] | $\sigma_{aD}^{Ha} = 1\cdot 10^{-2}$ (Day: Hartley bands) | $s^{-1}$ | r30 |
| $R_{s2.4}$ | 2.91[A] | $\sigma_{xD}^{Hu} = 1\cdot 10^{-4}$ (Day: Huggins bands) | $s^{-1}$ | r24 |
| $R_{s2.5}$ | 1.96[A] | $\sigma_{aP}^{Ch} = 3\cdot 10^{-4}$ (Day: Chappuis band) | $s^{-1}$ | r24 |
| $R_{s2.6}$ | 1.01[A] | $\sigma_{xP}^{Ch} = 3\cdot 10^{-4}$ (Day: Chappuis band) | $s^{-1}$ | r27 |
| $R_{s3.1}$ | | $\sigma_{b1}^{O2} = 5.35\cdot 10^{-9}$ (In sunlight conditions) | $s^{-1}$ | r28 |
| $R_{x1.1}$ | -5.12[R] | $\chi_{N2}^{Px} = cPXn\cdot xY\cdot 3\cdot 10^{-33}(300/T)^{3.25}$ | $molec^{-2}\,cm^6\,s^{-1}$ | r01 |
| $R_{x1.2}$ | -5.12[R] | $\chi_{O2}^{Px} = cPXm\cdot xY\cdot 3\cdot 10^{-33}(300/T)^{3.25}$ | $molec^{-2}\,cm^6\,s^{-1}$ | r01 |
| | | $xY = 0.12 + pY\cdot 0.25$ | 1 | r03 |
| | | $pY = 0.5$ (for $O_2(^5\Pi)$) | 1 | r02 |
| | | Optional: $cPXm = 7.67\cdot 10^3$ for $cPXn = cPXm$ | 1 | r03 |
| | | Current use: $cPXm \approx 3.56\cdot 10^4$ for $cPXn = 1$ | 1 | r03 |
| $R_{x2.1}$ | -4.06[A] | $\chi_{O2}^{3P} = 8\cdot 10^{-12}exp(-2060/T)$ | $molec^{-1}\,cm^3\,s^{-1}$ | r13 |
| $R_{x3.1}$ | -1.10[A] | $\chi_{N2}^{P3} = 6\cdot 10^{-34}(300/T)^{2.4}$ | $molec^{-2}\,cm^6\,s^{-1}$ | r13 |
| $R_{x3.2}$ | -1.10[A] | $\chi_{O2}^{P3} = 6\cdot 10^{-34}(300/T)^{2.4}$ | $molec^{-2}\,cm^6\,s^{-1}$ | r13 |
| $R_{h1.1}$ | -3.34[R] | $\eta_{OH}^{H} = 1.4\cdot 10^{-10}exp(-470/T)$ | $molec^{-1}\,cm^3\,s^{-1}$ | r13 |
| $R_{h2.1}$ | -0.73[A] | $\eta_{OH}^{3P} = 1.8\cdot 10^{-11}exp(180/T)$ | $molec^{-1}\,cm^3\,s^{-1}$ | r13 |
| $R_{h3.1}$ | -1.74[A] | $\eta_{HO2}^{OH} = 1.7\cdot 10^{-12}exp(-940/T)$ | $molec^{-1}\,cm^3\,s^{-1}$ | r13 |
| $R_{h4.1}$ | -2.33[A] | $\eta_{HO2}^{3P} = 3.0\cdot 10^{-11}exp(200/T)$ | $molec^{-1}\,cm^3\,s^{-1}$ | r13 |
| $R_{h5.1}$ | -2.11[A] | $\eta_{N2}^{H} = 4.4\cdot 10^{-32}(300/T)^{1.3}$ | $molec^{-2}\,cm^6\,s^{-1}$ | r13 |
| $R_{h5.2}$ | -2.11[A] | $\eta_{O2}^{H} = 4.4\cdot 10^{-32}(300/T)^{1.3}$ | $molec^{-2}\,cm^6\,s^{-1}$ | r13 |
| $R_{h6.1}$ | -1.60[A] | $\eta_{OH}^{HO2} = 7.2\cdot 10^{-11}$ | $molec^{-1}\,cm^3\,s^{-1}$ | r13 |
| $R_{h6.2}$ | -2.41[A] | $\eta_{H2}^{HO2} = 6.9\cdot 10^{-12}$ | $molec^{-1}\,cm^3\,s^{-1}$ | r13 |
| $R_{h6.3}$ | -2.33[A] | $\eta_{H2O}^{HO2} = 1.6\cdot 10^{-12}$ | $molec^{-1}\,cm^3\,s^{-1}$ | r13 |




**Table A1.** Processes of the prior retrieval and continued to shown in Table 2.

| $R_\#$ | Odd oxygen processes related to $O(^1S)$ |
|---|---|
| $R_{r1.1-3}$ | $O(^1D) + \{O(^3P), O_3, O_3\} \xrightarrow{\rho_{3P}^{DP}, \rho_{2P}^{DP}, \rho_{O2}^{DP}} \{2\,O(^3P), 2\,O(^3P) + O_2, 2\,O_2\}$ |
| $R_{r2.1-4}$ | $O(^1D) + \{N_2, O_2, O_2, CO_2\} \xrightarrow{\rho_{N2}^{DP}, \rho_{Oa}^{DP}, \rho_{Ob}^{DP}, \rho_{C2}^{DP}} O(^3P) + \{N_2, O_2(a), O_2(b), CO_2\}$ |
| $R_{r3.0}$ | $O(^1D) \xrightarrow{\rho_{1D3Pe}^{A}} O(^3P) + h\nu$ |

| $R_\#$ | Odd oxygen processes related to absorption and the catalytic ozone destruction |
|---|---|
| $R_{s1.1-5}$ | $O_2 + h\nu \xrightarrow{\sigma_{PS}^{UV}, \sigma_{PD}^{LA}, \sigma_{PD}^{Sc}, \sigma_{PP}^{Sb}, \sigma_{PP}^{Hc}} O(^3P) + \{O(^1S), O(^1D), O(^1D), O(^3P), O(^3P)\}$ |
| $R_{s2.1-6}$ | $O_3 + h\nu \xrightarrow{\sigma_{aS}^{UV}, \sigma_{PP}^{Ha}, \sigma_{aD}^{Hu}, \sigma_{xD}^{Hu}, \sigma_{aP}^{Ch}, \sigma_{xP}^{Ch}} \{O(^1S) + O_2(a), 3\,O, O(^1D) + O_2(a), O(^1D) + O_2, O + O_2(a), O + O_2\}$ |
| $R_{s3.1}$ | $O_2 + h\nu(\lambda = 762\,\mathrm{nm}) \xrightarrow{\sigma_{b1}^{O2}} O_2(b)$ |
| $R_{x1.1-2}$ | $O(^3P) + O(^3P) + \{N_2, O_2\} \xrightarrow{\chi_{N2}^{Px}, \chi_{O2}^{Px}} O_2 + \{N_2, O_2\}$ |
| $R_{x2.1}$ | $O(^3P) + O_3 \xrightarrow{\chi_{O2}^{3P}} 2\,O_2$ |
| $R_{x3.1-2}$ | $O_2 + O(^3P) + \{N_2, O_2\} \xrightarrow{\chi_{N2}^{P3}, \chi_{O2}^{P3}} O_3 + \{N_2, O_2\}$ |

| $R_\#$ | Odd hydrogen processes |
|---|---|
| $R_{h1.1}$ | $H + O_3 \xrightarrow{\eta_{OH}^{H}} OH(5 \leq \nu \leq 9) + O_2$ |
| $R_{h2.1}$ | $OH^* + O(^3P) \xrightarrow{\eta_{OH}^{3P}} H + O_2$ |
| $R_{h3.1}$ | $OH^* + O_3 \xrightarrow{\eta_{HO2}^{OH}} HO_2 + O_2$ |
| $R_{h4.1}$ | $HO_2 + O(^3P) \xrightarrow{\eta_{HO2}^{3P}} OH(\nu \leq 6) + O_2$ |
| $R_{h5.1-2}$ | $H + O_2 + \{N_2, O_2\} \xrightarrow{\eta_{N2}^{H}, \eta_{O2}^{H}} HO_2 + \{N_2, O_2\}$ |
| $R_{h6.1-3}$ | $H + HO_2 \xrightarrow{\eta_{OH}^{HO2}, \eta_{H2}^{HO2}, \eta_{H2O}^{HO2}} \{OH^* + OH^*, H_2 + O_2, O(^3P) + H_2O\}$ |