# Peer review of "Photochemical modeling of molecular and atomic oxygen based on multiple *in-situ* emissions measured during the Energy Transfer in the Oxygen Nightglow rocket campaign"

_Atmospheric Chemistry and Physics, 2019_

## Referee Comment (RC1) · Anonymous Referee #1 · 24 May 2019

**Referee Report**

**Manuscript no.: acp-2019-221**

**Title:** Photochemical modeling of molecular and atomic oxygen based on multiple *in-situ* emissions measured during the Energy Transfer in the Oxygen Nightglow rocket campaign.

Authors: Olexandr Lednyts'kyy and Christian von Savigny

In this paper the authors present a new airglow model (MAC, Multiple Airglow Chemistry model) that includes electronically excited states of molecular and atomic oxygen (six of O2 and two of O) and their ground states. The model is based on the measurements and findings of the ETON sounding rocket campaign conducted from South Uist, Scotland in March 1982 and extends this with later efforts by several authors to model the photochemistry of the MLT (Mesosphere/Lower Thermosphere) region, and updated reaction rates. Unfortunately, the *in situ* measurements of the atmospheric neutral temperature during the ETON campaign were not successful. Instead, the temperature (and neutral density) were taken from the NRLMSISE-O0 model in the current study. A sensitivity study was conducted by the authors to investigate the influence of changes in temperature and neutral density in the retrieval of the different excited and ground states of molecular and atomic oxygen.

To take this model (and our knowledge of airglow photochemistry) further, dedicated simultaneous *in situ* measurements of relevant airglow emissions, atomic oxygen, neutral temperature and density are needed.

**General comments**

The paper presents an extensive model to explain the excitation mechanisms responsible for the observed airglow emissions from the MLT region of the Earth's atmosphere. It is a nice review of the current knowledge of airglow photochemistry, and it constrain the precursors responsible for the Atmospheric band, Infrared Atmospheric band and the Oxygen Green Line emissions. It is an important contribution to the scientific community.

However, in some parts of the manuscript, in particular section 3, the language (sentence structure) needs to be improved as I sometimes find it difficult to read certain sentences. Sections 4 and 5 are much easier to read. The manuscript would greatly benefit from being read and corrected by someone fluent in English.

Since "*in situ*" is a Latin phrase with a specific meaning it should be written "*in situ*", not "*in-situ*". It is, however, very common, and seems widely accepted, to write in-situ or even *in-situ* (both italicised and hyphenated) and I suppose it is ok as long as it is consistently done so throughout the manuscript (which it is).

Sometimes it is written "datasets" and sometimes (most of the times) "data sets". Both are widely used but it should be consistent in the manuscript.

Remove "the" in front of "step" or "steps" when discussing the retrieval steps.

**Specific suggestions/comments/questions in order of appearance**

Title:"...and atomic oxygen based on multiple nightglow emissions measured *in situ* during the Energy Transfer..."

Page 1, lines 3-4: "The MAC model combines chemical processes of well-known photochemical models..."

Page 1, line 6: "...the multiple nightglow emissions measured *in situ* during the Energy Transfer..."

Page 1, line 7: "...obtain concentrations of these minor species in the MLT region is implemented..."

Page 1, lines 10-11: "...considered in the MAC model are identified and validated."

Page 1, line 22: "...active MLT trace gas and a critical component..."

Page 2, lines 25-26: "Dynamic quenching reduces the apparent fluorescent lifetime, while static quenching rather reduces the apparent concentration..."

Page 2, line 28: "...the dynamic quenching, and can introduce difficulties..."

Page 3, line 25: "The ETON multiple airglow emissions described in Section 2 can be applied..."

Page 3, line 30: "...scattered in time and place and might have stopped Torr et al. (1985) from combining..."

Page 4, lines 19-20: "All VER profiles related to... ...by two ETON rockets. The Infrared Atmospheric band emission at 1.27  $\mu$ m was measured..."

Page 4, lines 28-29: "...ground state  $[O({}^{3}P)]$  were carried out by the P232H and P234H rockets launched at..."

Page 4, lines 32-33: "...(where peak  $[O(^{3}P)]$  values were measured)... ...(where low  $[O(^{3}P)]$  values were measured)"

Page 5, lines 1-6: I suggest adding a reference to the NRLMSISE-00 model here (Picone, J. M., A. E. Hedin, D. P. Drob, and A. C. Aikin, NRLMSISE-00 empirical model of the atmosphere: Statistical comparisons and scientific issues, J. Geophys. Res., 107(A12), 1468, doi:10.1029/2002JA009430, 2002.), and possibly also the MSIS-83 model (Hedin, A. E., A revised thermospheric model based on mass spectrometer and incoherent scatter data: MSIS-83, J. Geophys. Res., 88, A12, 1983)

Page 5, line 10: "Some of the O2 transitions..."

Page 5, line 19: "...on the basis of the data sets obtained..."

Page 5, line 22: "..., see the next publication."?? Is there a manuscript in preparation with a title and author list that can be referenced here?

Page 6, Table 1 caption: The reference "na – Nagy et al. (2008)" is written twice.

Page 7, line 6: "...equation of McDade et al. (1986) provided below in the full form..."

Page 7, line 8: "The cubic equation in the full form is as follows:"

Page 7, equation 2: Why can't the equation be in a single line, as equation 1?

Page 7, line 26: "...using semi-empirical models, including MSIS-83, that are no longer available."

Page 7, line 29: "The lowest obtained values of C(0), C(1) and C(2), related to the O( $^{1}S$ ) precursor, were found..."

Page 7, line 31: "...their highest values were found to be..."

Page 9, line13: "...Kenner and Ogryzlo (1982). However, Johnston and Broadfoot (1993)..."

Page 10, line 23-24: "As for the well-known cubic Eq. (2), it was solved..."

Page 10, line27-28: "...in this study, the values of the reaction rates and empirical coefficients used are the ones provided by Murtagh et al. (1990)."

Page 10, line 32: "...photochemical models was provided in Section 1."

Page 10, line 35: "...second model, see section 3.2.2, developed using available data sets."

Page 12, line 6: Is "contiguously" the correct word to be used here or should "continuously" be used instead?

Page 13, line 8: "...(2014) (see Table 3 in section 3.2.1):"

Page 14, line 2-3: "...MAC model and are referred to as M-processes"

Page 14, line 10: "...MAC model and are referred to as H-processes"

Page 14, line 16: "...G-model are referred to as G-processes."

Page 14, line 23-25: Reformulate this sentence, e.g. "Although the Barth excitation transfer scheme was formulated with  $O_2^*$  considered as one not identified  $O_2$  states, a group of many

not identified  $O_2$  states coupled in a cascade of de-excitation reactions is also possible." Or did I not understand the meaning of this sentence?

Page 14, lines 26-27: How can Slanger et al. (2004b) have refuted the hypothesis by Huestis (2002) based on laboratory measurements discussed by Pejakovic at al. (2007)? A paper that was published 3 years after? I suggest to remove the reference of Pejakovic et al. (2007), or reformulate this section.

Page 14, line 29: "...the de-excitation of the  $O_2$  states does not occur in a cascade-like process."

Page 14, line 32: "...removed by conversion to very high vibrational..."

Page 14, line 32 to page 15, line 1: "...that  ${}^{5}\Pi$  is an electronically excited O2 state with higher energy than..."

Page 15, lines 16-19: Reformulate.

Page 16, line 8: "The advantage of the ETON campaign compared to other rocket campaigns..."

Page 16, line19: "...model are described in Section 2 and include VER profiles..."

Page 17, lines 6-10: Remove all "the" in front of "step".

Page 18, lines 9-10: "...each retrieval step. These profiles also seem ... "

Page 19, line 1: "...profile) enables the conclusion that all..."

Page 20-21, Figures 1 and 2: Only SCH04 is somewhat defined in the caption of figure 1, none of the other anywhere in the manuscript.

Page 20-21: What is the meaning of defining MMG+86 and LSE+15 in the text here? They are used later, and defined in the caption, in figure 4.

Page 21, line 7: Is "equidistant" the correct word to use here?

Page 21, line 8: "..two profiles of extreme values. ...averaging of the extreme..."

Page 21, line 9,: "...see the violet crosses on the left in both figures."?? There are violet crosses in both panels in figure 4, what do you mean?

Page 22, line 1: "...in relation to reaction rates in which..."

Page 22, line 2: "Note that processes..."

Page 22, line 8: "...retrieved at step 4.1 on the basis..."

Page 22, line 15: "...are equal to zero, whereas..."

Page 22, line 16-17: "...can not be shown in Fig. 5 because... ...the retrieval steps 2.1 and 2.2."

Page 23, lines 3-5: Reformulate this sentence.

Page 23, line 8: "...values retrieved at step 3.2 on the basis..."

Page 23, line 9: "...variable with a variability higher than those..."

Page 24, Figure 4 caption: "...(see the violet crosses on the left in this figure)? Do you mean "the violet crosses in the left panel of this figure"?

Page 24, line 1: "...values retrieved at step 3.2 are in agreement..."

Page 24, line 5: "...in all possible levels v' in..."

Page 25, line 8: "Unfortunately, it would not be enough..."

Page 26, line 7: "...at each of the retrieval steps listed in Table 8."

Page 27, Figure 6 caption: "The retrievals were performed at steps..."

Page 28, line 34: "...peak values, see Section 2. Varying..."

Page 29, line 19: "..., see the next article to be submitted."?? Is there a manuscript in preparation with a title and author list that can be referenced here?

Page 32, lines 20-21: Is there a manuscript in preparation with a title and author list that can be referenced here?

Page 32, line 34: "...not well known because it has recently been discovered by Cacace et al., (2001), may be..."

Page 33, lines 22-23: "," missing between  ${}^{1}D$  and  ${}^{3}P$  in two places.

Page 35, line 5: "...were combined with suggested complementary processes to complete the list..."

Page 37, line 9: "...provided in Section 3.4 and in Table 8."

Page 37, line 13: "...are equal to zero."

Page 37, line 14: "...according to the processes of the different models adopted in the MAC model."

Page 37, line 5: "...in the next publication..."?? Is there a manuscript in preparation with a title and author list that can be referenced here?

Page 40, line 30: "...considering the processes shown in Tables 6...5"? Do you mean Tables 5 and 6, or Tables 6 to some other number higher than 6?

Page 42, line 9: "...considering the processes shown in Tables 6...5"? Do you mean Tables 5 and 6, or Tables 6 to some other number higher than 6?

Page 42, line 28: "...considering the processes shown in Tables 6...5"? Do you mean Tables 5 and 6, or Tables 6 to some other number higher than 6?

Page 44, line 8: "...considering the processes shown in Tables 6...5"? Do you mean Tables 5 and 6, or Tables 6 to some other number higher than 6?

Page 45, line 11: "...considering the processes shown in Tables 6...5"? Do you mean Tables 5 and 6, or Tables 6 to some other number higher than 6?

All references in the text are listed in the reference list and vice versa.

Page 56, Table 9 caption: "Processes of the provided rate values are shown in Table..."

Page 57, Table 10 caption: "Processes of the provided rate values are shown in Table..."

Page 58, Table 11 caption: "Processes of the provided rate values are shown in Tables..."

Page 59, Table 12 caption: "Processes of the provided rate values are shown in Tables..."

---

## Referee Comment (RC2) · Anonymous Referee #2 · 3 Jul 2019

I believe that this paper contains a valuable contribution to the field and as such is publishable. However the paper is extremely difficult to read and requires major revision of the structure and language before it can be published. The paper is attempting a complex task and may be helped by some flow charts show in the procedures, the relationships between various products etc.

As I have managed to understand the paper: The authors first review a substantial

literature on many of the airglow emissions contrasting and comparing relevant groups and then make selections as to which reactions to maintain in the MAC model. Some tests are then carried out but I was unable to follow exactly what was done. I believe that Figure 3 is supposed to show the agreement between the model after tuning with the various ETON emission profiles. I do not see the point in showing both the concentration of the emitter and the intensity of the emissions as the conversion between them is trivial.

The sensitivity analysis in section 3.5 is a useful contribution. I might note that the temperature cannot actually be varied independently of the pressure because of hydrostatic equilibrium.

Much of the content of section 4 -discussion should appear earlier in the paper to explain what is actually done in the model. For instance the tuning of parameters is described first in section 4.1.

The discussion of dynamics on page 32 seems somewhat superfluous to the aim of the paper and could be excluded.

The use of some wording also confuses the reader. I believe that the use of "continuity equation" in for instance line 9 page 32 refers to a steady state chemical balance equation although some wording earlier in the paper may suggest that time dependent equations are being solved.

The use of the word retrieve when I believe calculate would be better is also a problem. Eg line 10 page 38, line 12 page 39

A few minor points Page 4 line 22 - The reference to Greer et al 1986 about the atomic oxygen measurement technique would be better replaced but a reference to one of the Dickinson papers.

Page 12 second line of the caption to table 3. I believe it should be the character E that marks the equations excluded not M

Page 28 . In the discussion of the vibrational distribution of the Herzberg states a reference to some of the ground based work by Slanger or Stegman might be appropriate.

---

## Author Comment (AC2) · 26 Jul 2019

***Response regarding comments of the Anonymous Referee #2 on acp-2019-221:*** **Photochemical modeling of molecular and atomic oxygen based on multiple *in-situ* emissions measured during the Energy Transfer in the Oxygen Nightglow rocket campaign**

[Figure]

***Comments of the Anonymous Referee #2***
***Feedback to the pdf-paper:***
I believe that this paper contains a valuable contribution to the field and as such is publishable. However the paper is extremely difficult to read and requires major revision of the structure and language before it can be published.

***General comments to the pdf-paper:***
The paper is attempting a complex task and may be helped by some flow charts show in the procedures, the relationships between various products etc.

As I have managed to understand the paper: The authors first review a substantial literature on many of the airglow emissions contrasting and comparing relevant groups and then make selections as to which reactions to maintain in the MAC model.

***List of corrections regarding general comments of the Anonymous Referee #2***
The authors of the manuscript are grateful to the Anonymous Referee #2 for the general comments. We agree that the paper was difficult to follow. We edited the paper and changed the text substantially. The paper is now hopefully easier to follow. Also, following the reviewer's suggestion a flow chart was added as Fig. A1. Changes in the manuscript were highlighted by bold font.

***Page 37, line 14, regarding the new third paragraph:***
Retrieval steps resulting in $[\mathrm{O}(^3P)]$ and carried out according to the proposed algorithm are illustrated in the flow chart in Fig. A1.

*Additionally, the new Fig. A1 was added on page 37. The figure is saved as the supplement file acp-2019-221-AC2-supplement.pdf. Text of the figure caption is as follows:*
The flow chart shows retrieval steps resulting in $[\mathrm{O}(^3P)]$ and carried out according to

[Figure]

the proposed MAC approach. The start and end states are denoted by filled black circles on the top and bottom of the figure, respectively. Decisions and processes are denoted by rhombs and rectangles, respectively. Connectors are denoted by empty circles. The flow chart is read following lines with arrows from one flow chart symbol to another. The prior retrieval procedure is described by the text shown in blue. If the prior retrieval procedure can be omitted (as it is the case for the ETON campaign), then the corresponding decision "Not important" (shown in violet) near a rhomb is to be taken that is denoted by "optionally" (shown in violet in a rectangle) being relevant for the optional calculation result (shown in blue in a rectangle). The optional procedure carried out to retrieve $[O_2(A)]$ and $[O_2(A')]$ is described by the text shown in green. If emissions in the Herzberg I and Chamberlain bands are not available (see "No" shown in violet near the respective rhombs) or optional (see "Not important" shown in violet near the respective rhombs), then this optional procedure can be omitted at steps 2.3 and 3.1, see $[O_2(A)]$ and $[O_2(A')]$ shown in green in the respective rectangles. Note that the $[O(^3P)]$ retrieval can be carried out most accurately if values of VER$\{O_2(b-X)\}$ are available which is indicated by the text shown in red.

*Additionally, the following relevant change was carried out in Table 9 on page 56, Table 10 on page 57, Table 11 on page 58, Table 12 on page 59 and Table A1 on page 60. These tables were sorted and placed not after the bibliography as was done earlier, but in the respective sections.*

*Additionally, the following relevant change was carried out in Table 12 on page 27:*
*The content of Table 12 was revised and compressed representing concentrations of many excited species, e.g. $[O_2(A, A')]$ instead of $[O_2(A)]$, $[O_2(A')]$ and so on. Note that this change was not highlighted.*

*Additionally, the following relevant change was carried out in Table 7 on page*

*19:*

$\ldots R_{s2.1-6} \ldots$

was changed to:

$\ldots R_{s2.1-3} \ldots R_{s2.4-6} \ldots$

**List of corrections regarding specific comments of the Anonymous Referee #2**

The authors of the manuscript are grateful to the Anonymous Referee #2 for the specific comments. The manuscript was substantially revised and the english has been hopefully improved. We hope that the paper is now easier to follow. Changes in the manuscript were highlighted by bold font.

**Specific comment:**

Some tests are then carried out but I was unable to follow exactly what was done. I believe that Figure 3 is supposed to show the agreement between the model after tuning with the various ETON emission profiles. I do not see the point in showing both the concentration of the emitter and the intensity of the emissions as the conversion between them is trivial.

**Response:**

Thank you for reminding about the triviality of the conversion between the concentration of the emitter and the intensity of the emissions. We would like to keep the text about this conversion in Appendix A describing steps in sections A2.1, A2.2, A2.3, A3.2 and A4.1. The corresponding consistency tests are described in Appendix A in Sections A2.4, A3.3 and A4.2. We would also like to keep both panels in Fig. 3 as the result of the carried out consistency tests. We substantiate our choice by the fact that Huang and George (2014) verified their calculations comparing concentration profiles

in their Fig. 1. Therefore, we also show concentration profiles in the right panel of our Fig. 3 to allow the comparison of calculation results shown by Huang and George (2014) in their Fig. 1 and our results shown in the right panel of our Fig. 3 easier, i.e. without additional calculations required for this conversion. In fact, the MAC model includes reactions proposed by Huang and George (2014) as well as many other reactions. Note that we employ updated reaction rate values considered by Huang and George (2014) in their Table 1. This makes the comparison of concentration profiles obtained by us and Huang and George (2014) necessary. Additionally, we compare VER profiles in the left panel of our Fig. 3 as was done by Murtagh et al. (1990).

*Following your comment, the following relevant change was carried out in caption of Fig. 3 on page 23:*
. . . are coherent with measurements. The abbreviations . . .
was changed to:
. . . are coherent with measurements. The corresponding consistency tests are described in Sections A2.4, A3.3 and A4.2. Note that the conversion between profile values of VER and concentrations is based on trivial, but required calculations provided in Sections A2.1, A2.2, A2.3, A3.2 and A4.1. The abbreviations . . .

**Specific comment:**
The sensitivity analysis in section 3.5 is a useful contribution. I might note that the temperature cannot actually be varied independently of the pressure because of hydrostatic equilibrium.

**Response:**
The reviewer is certainly correct that temperature and pressure are directly connected. However, it may occur that one of the two quantities is measured with a very high accuracy, and the other one is not. We would like to keep the sensitivity analysis considering the mentioned parameters, i.e. $[N_2]$ and $[O_2]$ (calculated using pressure)

as well as temperature, separately because of two reasons. Firstly, instruments aboard the WADIS-2 rocket and the SABER instrument provided data required to obtain pressure and temperature. The sensitivity analysis carried out by Lednyts'kyy et al. (2019) is based on pressure and temperature considered independently from each other. The sensitivity analysis discussed in our Section 3.5 is carried out in a similar way as was done by Lednyts'kyy et al. (2019). Finally, the mentioned parameters are also considered independently from each other in the continuity equations employed here. In summary, we would like to keep the applicability of the sensitivity analysis because it contains more information than treating temperature and pressure in a combined way.

***Specific comment:***
Much of the content of section 4 - discussion should appear earlier in the paper to explain what is actually done in the model. For instance the tuning of parameters is described first in section 4.1.
*The text on page 28, lines 12-17, was used to extend the text on page 14, line 32:*
. . . states. It should be noted that . . .
was changed to:
. . . states. In fact, the removal of the $O_2(^5\Pi)$–$O_2(A, A')$–group through collisions was suggested by Slanger et al. (2004b) and implemented in the MAC model implicitly by increasing the association rates of $O_2(b, a, X)$ in the three-body recombination reactions. This was done implicitly because reactions including $O_2(^5\Pi)$ are not well known, e.g., compare Krasnopolsky (2011) and Krasnopolsky (1986). It should be noted that $O_2(^5\Pi)$ has a shorter lifetime and a higher energy compared to the other states $O_2(A, A', c, b, a, X)$ as it was also mentioned by Huestis (2002) and Slanger et al. (2004b). It should be noted that . . .

*Additionally, the following relevant change was carried out in the title of Section 4.1 on page 29, line 9:*
Discussion of tuned rate values of quenching processes implemented in the MAC

model.
was changed to:
Tuning rate values of quenching processes implemented in the MAC model.

*Additionally, the text of Section 4.1 on pages 29 and 30 was written after Section 3.3 on page 16. Note that the added text was not highlighted.*

**Specific comment:**
The discussion of dynamics on page 32 seems somewhat superfluous to the aim of the paper and could be excluded.
**Response: The mentioned text on page 32, lines 16-21, and on page 5, lines 14-17, was deleted.**

*Additionally, the reference to Smith et al. (1987) on page 54, lines 19-21, was deleted. The reference to Johnson and Gottlieb (1973) on page 50, lines 17-18, was also deleted.*

**Specific comment:**
The use of some wording also confuses the reader. I believe that the use of "continuity equation" in for instance line 9 page 32 refers to a steady state chemical balance equation although some wording earlier in the paper may suggest that time dependent equations are being solved.
*Page 32, line 10:*
Continuity equations implemented . . .
was changed to:
Steady state chemical balance equations (also referred to as continuity equations) implemented . . .

*Additionally, a similar change was carried out on page 2, line 30:*

. . . the resulting mass conservation (continuity) equation . . .
was changed to:
. . . the resulting steady state chemical balance equation (hereafter referred to as continuity equation) . . .

*Additionally, a similar change was carried out on page 37, line 6:*
Then simple continuity equations . . .
was changed to:
Then simple steady state chemical balance equations (referred to as continuity equations) . . .

**Specific comment:**
The use of the word retrieve when I believe calculate would be better is also a problem. Eg line 10 page 38, line 12 page 39.
*The following change was carried out in titles of each subsection of the Appendix A (e.g. A2.1, A2.2, A2.3 and A2.4, but not A2) and for references in Appendix A to the mentioned sections:*
. . . retrieval . . .
was changed to (and highlighted in bold italic font):
. . . calculation . . .

**Specific comment:**
A few minor points Page 4 line 22 - The reference to Greer et al 1986 about the atomic oxygen measurement technique would be better replaced but a reference to one of the Dickinson papers.
**Page 4, line 30:**
. . . at ∼130 nm (Greer et al., 1986) and . . .
was changed to:
. . . at ∼130 nm (Dickinson et al., 1980) and . . .

***Specific comment:***
Page 12 second line of the caption to table 3. I believe it should be the character E that marks the equations excluded not M.

***Page 5, line 2, caption of Table 3:***
The processes marked with a character $M$ are not considered . . .
was changed to:
The processes marked with a character $E$ are not considered . . .

***Specific comment:***
Page 28. In the discussion of the vibrational distribution of the Herzberg states a reference to some of the ground based work by Slanger or Stegman might be appropriate.

***Page 28, line 21:***
. . . energetically very close to each other. Vibrational states . . .
was changed to:
. . . energetically very close to each other. Vibronic energy levels of $O_2(A, A', c, b, a, X)$ are shown in Fig. 8 by Goodman and Brus (1977). The atlas of terrestrial nightglow emission lines in the range 314. . .1043 nm including emission lines of these $O_2$ states is provided in Table 3 as a compressed form of the electronic supplement of Cosby et al. (2006). Vibrational states . . .

*Additionally, the following relevant change was carried out on page 28, line 10:*
. . . coupled with $O_2(c, b, a, X)$ which was also adopted in the MAC model.
was changed to:
. . . coupled with $O_2(c, b, a, X)$.

*Discussion with C. von Savigny resulted in the following changes.*

*The following change was carried out in Table 2 on page 8, Table 3 on page 12, Table 4 on page 13, Table 5 on page 17, Table 6 on page 18, Table 7 on page 19, and Table A1 on page 60:*

$\dots h\nu(\,\dots$

was changed to:

$\dots h\nu\,(\,\dots$

*Additionally, the following change was carried out on page 4, lines 19-20:*

The maximal number of VER profiles related to various $O_2$ and $O(^3P)$ transitions were obtained by two ETON rockets, which are discussed here. As for the Infrared Atmospheric band emissions at 1.27 $\mu$m, they were measured $\dots$

was changed to:

All VER profiles considered in the MAC model were measured during flights of two ETON rockets. The Infrared Atmospheric band emission at 1.27 $\mu$m was measured $\dots$

*Additionally, the following change was carried out on page 11, lines 17-19:*

In contrast to the YM2011 model, reaction rates in the modified kinetic model of Mlynczak et al. (1993) refer to a specific portion of vibrational states from their statistical equilibrium in each $O_2$ electronic state.

was changed to:

Rate values of reactions involving $O_2(b)$ and $O_2(a)$ in the modified kinetic model of Mlynczak et al. (1993) do not directly correspond to rate values of reactions involving various vibrational states of $O_2(b)$ and $O_2(a)$ in the YM2011 model because vibrational states are not identified in the modified kinetic model of Mlynczak et al. (1993).

*Additionally, the following change was carried out on page 30, lines 9-10:*

$\dots$ the results of which are given by the interval of possible rate values shown in the third column of Tables 8$\dots$11.

was changed to:

… see the third column of Tables 8, 9, 10 and 11 for a summary.

*Additionally, the following change was carried out on page 28, lines 24-25:*
Considering internuclear distances (INDs) of the corresponding Franck-Condon factors, it should be emphasized that the difference in INDs between the excited …
was changed to:
Considering Franck-Condon factors and the corresponding internuclear distances (INDs), it should be emphasized that the difference in INDs between the excited …

*Additionally, the following change was carried out on page 34, lines 30-32:*
It is difficult to argue about the fractional $O_2(c)$ vibrational population of the $O(^1S)$ precursor because the $[O_2(c, \nu=6, 7, 8)]$ peak might be at about 97 km altitude if the collisional activation energy mentioned by Krasnopolsky (1981) is neglected.
was changed to:
In summary, the exact role of the vibrational excitation of $O_2(c)$ as a precursor of $O(^1S)$ is still not well understood and should be investigated in future studies.
*Additionally, the following change was carried out on page 36, lines 19-21:*
Then, the following correspondences among the violated electric dipole selection rules and the transition intensity seem to be established. … (3) almost absent between $O_2(A^3\Sigma_u^+)$ and $O_2(A'^3\Delta_u)$ as well as $O_2(c^1\Sigma_u^-)$ and $O_2(a^1\Delta_g)$.
was changed to:
Then, the following correspondences regarding the selection rules for chemical reactions were established. … (3) almost nearly absent between $O_2(A^3\Sigma_u^+)$ and $O_2(A'^3\Delta_u)$ as well as between $O_2(c^1\Sigma_u^-)$ and $O_2(a^1\Delta_g)$.

*Additionally, the following change was carried out on page 40, lines 25-27:*
This is justified because $O_2$ in these triplet excited states is decoupled from the singlet excited states according to the hypothesis of Slanger et al. (2004b) used to propose the MAC model, which were verified and validated, (see Section 3.5.

[Figure]

was changed to:

This is justified because the hypothesis of Slanger et al. (2004b) was adopted to propose the MAC model. Note that the MAC calculations were verified and validated, see Section 3.5 for details.

*Additionally, the following change was carried out on page 43, lines 24-25:*
. . . was not retrieved on the basis of VER values, but concentrations available from the previous calculation steps.
was changed to:
. . . was retrieved on the basis of concentrations available from the previous calculation steps, whereas VER profiles were not employed for the $[O_2(c)]$ retrievals directly.

*Additionally, discussion with C. von Savigny resulted in the update of references on page 48 line 10, page 51 line 10, page 51 line 16, page 51 line 18, and page 51 line 22.*

*Additionally, discussion with C. von Savigny resulted in numerous changes regarding English grammar and style. These relevant changes were carried out and highlighted in the manuscript.*

Please also note the supplement to this comment:
https://www.atmos-chem-phys-discuss.net/acp-2019-221/acp-2019-221-AC2-supplement.pdf

**Supplement:**

---

## Author Response (AR1)

*Final response regarding comments of the Anonymous Referees #1 and #2 on acp-2019-221:*
**Photochemical modeling of molecular and atomic oxygen based on multiple *in-situ* emissions measured during the Energy Transfer in the Oxygen Nightglow rocket campaign**

*Comments of the Anonymous Referee #1*
*Feedback to the pdf-paper:*
In this paper the authors present a new airglow model (MAC, Multiple Airglow Chemistry model) that includes electronically excited states of molecular and atomic oxygen (six of $O_2$ and two of O) and their ground states. The model is based on the measurements and findings of the ETON sounding rocket campaign conducted from South Uist, Scotland in March 1982 and extends this with later efforts by several authors to model the photochemistry of the MLT (Mesosphere/Lower Thermosphere) region, and updated reaction rates. Unfortunately, the *in situ* measurements of the atmospheric neutral temperature during the ETON campaign were not successful. Instead, the temperature (and neutral density) were taken from the NRLMSISE-00 model in the current study. A sensitivity study was conducted by the authors to investigate the influence of changes in temperature and neutral density in the retrieval of the different excited and ground states of molecular and atomic oxygen.
To take this model (and our knowledge of airglow photochemistry) further, dedicated simultaneous *in situ* measurements of relevant airglow emissions, atomic oxygen, neutral temperature and density are needed.
*General comments to the pdf-paper:*
The paper presents an extensive model to explain the excitation mechanisms responsible for the observed airglow emissions from the MLT region of the Earth's atmosphere. It is a nice review of the current knowledge of airglow photochemistry, and it constrain the precursors responsible for the Atmospheric band, Infrared Atmospheric band and the Oxygen Green Line emissions. It is an important contribution to the scientific community.
However, in some parts of the manuscript, in particular section 3, the language (sentence structure) needs to be improved as I sometimes find it difficult to read certain sentences. Sections 4 and 5 are much easier to read. The manuscript would greatly benefit from being read and corrected by someone fluent in English.
Since "*in situ*" is a Latin phrase with a specific meaning it should be written "*in situ*", not "*in-situ*". It is, however, very common, and seems widely accepted, to write in-situ or even *in-situ* (both italicised and hyphenated) and I suppose it is ok as long as it is consistently done so throughout the manuscript (which it is).
Sometimes it is written "datasets" and sometimes (most of the times) "data sets". Both are widely used but it should be consistent in the manuscript.
Remove "the" in front of "step" or "steps" when discussing the retrieval steps.

**List of corrections regarding general comments of the Anonymous Referee #1**

The authors of the manuscript are grateful to the Anonymous Referee #1 for the general comments. The manuscript was substantially revised and the english has been hopefully improved. We hope that the paper is now easier to follow. Changes in the manuscript are highlighted by bold font.

**Title:**
... and atomic oxygen based on multiple *in-situ* emissions measured during the Energy Transfer ...
was not highlighted, but it was changed to:
... and atomic oxygen based on multiple nightglow emissions measured *in situ* during the Energy Transfer ...

*Everywhere in the manuscript:*
in-situ
was not highlighted, but it was changed to:
*in situ*

**Page 5, line 17:**
datasets
was changed to:
data sets

**Page 22, line 14:**
Profiles of VER and $[O(^3P)]$ obtained at the retrieval steps 2.3, 3.1, 3.2, 4.1 and 5.1 are shown ...
was changed to:
Profiles of VER and $[O(^3P)]$ obtained at steps 2.3, 3.1, 3.2, 4.1 and 5.1 are shown ...

*Additionally, a similar change was carried out in caption of Fig. 4 on page 24:*
Although the retrieval steps 2.1, 2.2, 2.3 and 3.2 applied on the basis of some ETON VER profiles result in lower $[O(^3P)]$ values compared to the *in situ* ones, the retrieval step 4.1 applied on the basis of VER$\{O(^1S - {}^1D)\}$ results in higher values.
was changed to:
Although steps 2.1, 2.2, 2.3 and 3.2 applied on the basis of some ETON VER profiles result in lower $[O(^3P)]$ values compared to the *in situ* ones, the $[O(^3P)]$ retrieval carried out at step 4.1 on the basis of VER$\{O(^1S - {}^1D)\}$ results in higher values.

*Additionally, a similar change was carried out in caption of Fig. 5 on page 25:*

... at the retrieval steps 2.3, 3.2, 4.1 and 5.1 ...
was changed to:
... at steps 2.3, 3.2, 4.1 and 5.1 ...

*Additionally, a similar change was carried out on page 30, line 14:*
... at the retrieval step 3.2 shown in Table 8. ...
was changed to:
... at step 3.2 shown in Table 8. ...

*Additionally, a similar change was carried out on page 31, lines 31-33:*
The dependence of $[O(^3P)]$ values on $O(^3P)$ loss processes is very high at the last retrieval step 5.1 because the $R_{x1.1-2}$ reactions are taken into account in the continuity equations at this step.
was changed to:
The $[O(^3P)]$ retrieval carried out at step 5.1 exhibits the dependence of the retrieved $[O(^3P)]$ values on the additional $O(^3P)$ loss processes implemented at this step, whereas the $[O(^3P)]$ retrievals carried out at steps 2.1, 2.2, 2.3, 3.1, 3.2 and 4.1 do not involve the $R_{x1.1-2}$ reactions in the corresponding steady state chemical balance equations.

*Additionally, a similar change was carried out on page 35, lines 31-35:*
Uncertainties in such VER values as VER$\{O_2(A-X)\}$ and VER$\{O_2(A'-a)\}$ at the retrieval steps 2.1 and 2.2 cause up to about 40% of $[O(^3P)]$ variations, uncertainties in such VER values as VER$\{O_2(b-X)\}$ and VER$\{O_2(a-X)\}$ at the retrieval steps 2.3 and 3.2 cause about 12% of $[O(^3P)]$ variations, but uncertainties in such VER values as VER$\{O(^1S-{}^1D)\}$ at the retrieval step 4.1 cause up to about 20% of $[O(^3P)]$ variations.
was changed to:
Uncertainties in values of VER$\{O_2(A-X)\}$ and VER$\{O_2(A'-a)\}$ cause $[O(^3P)]$ variations of up to about 40% at steps 2.1 and 2.2, respectively; uncertainties in values of VER$\{O_2(b-X)\}$ and VER$\{O_2(a-X)\}$ cause $[O(^3P)]$ variations of about 12% at steps 2.3 and 3.2, respectively; whereas uncertainties in values of VER$\{O(^1S-{}^1D)\}$ cause $[O(^3P)]$ variations of up to about 20% at step 4.1.

*Additionally, a similar change was carried out on page 37, line 4:*
The retrieval steps of $[O(^3P)]$ are closely related to the development of the MAC model.
was changed to:
The development and application of the MAC model is closely related to the retrieval steps required to obtain $[O(^3P)]$ profiles.

*Additionally, a similar change was carried out on page 41, lines 22-23:*
Calculations at the retrieval steps 2.1 and 2.2 are relevant for the MAC model involving $O_2(A)$ and $O_2(A')$, but calculations at the retrieval step 2.3 only are relevant for the MAC model excluding $O_2(A)$ and $O_2(A')$, see the following

overview.

was changed to:

Calculations carried out at steps 2.1 and 2.2 are relevant for the MAC model involving $O_2(A)$ and $O_2(A')$, but calculations carried out at step 2.3 only are relevant for the MAC model excluding $O_2(A)$ and $O_2(A')$, see the following overview.

*Additionally, a similar change was carried out on page 42, lines 4-6:*

$[O_2(c)]$ values were retrieved (R-$[O_2(c)]$) on the basis of $[O_2(A)]$, $[O_2(A')]$ and $[O_2(b)]$ values (obtained at the retrieval steps 2.1, 2.2 and 2.3, respectively) as well as $[O(^3P)]$ values (obtained at the retrieval step 2.3) according to the continuity equation for $[O_2(c)]$ considering all relevant processes of the MAC model.

was changed to:

$[O_2(c)]$ values were retrieved (R-$[O_2(c)]$) on the basis of $[O_2(A)]$, $[O_2(A')]$ and $[O_2(b)]$ values (obtained at steps 2.1, 2.2 and 2.3, respectively) as well as $[O(^3P)]$ values (obtained at step 2.3) according to the continuity equation for $[O_2(c)]$ considering all relevant processes of the MAC model.

*Additionally, a similar change was carried out on page 43, lines 23-24:*

Calculations at the retrieval step 3.1 can not be tested for consistency because . . .

was changed to:

The corresponding calculations carried out at step 3.1 could not be tested for consistency because . . .

*Additionally, a similar change was carried out on page 43, line 27:*

. . . calculations at the retrieval step 3.2 only can be tested for consistency.

was changed to:

. . . only calculations carried out at step 3.2 are tested for consistency.

*Additionally, a similar change was carried out on page 45, lines 7-8:*

. . . at the retrieval steps 2.1, 2.2, 2.3, 3.1, 3.2 and 4.1, respectively.

was changed to:

. . . at steps 2.1, 2.2, 2.3, 3.1, 3.2 and 4.1, respectively.

***Page 26, line 1:***

. . . retrieved at the step 3.2 . . .

was changed to:

. . . retrieved at step 3.2 . . .

*Additionally, a similar change was carried out on page 31, line 34:*

. . . retrieved at the steps 2.1, 2.2, 2.3, 3.1, 3.2 and 4.1 . . .

was changed to:

. . . retrieved at steps 2.1, 2.2, 2.3, 3.1, 3.2 and 4.1 . . .

*Additionally, a similar change was carried out on page 32, line 2:*

... e.g. the step 4.1, ...
was changed to:
... e.g. step 4.1, ...

*Additionally, a similar change was carried out on page 32, line 5:*
... retrieved at the steps 2.1, 2.2, 2.3, 3.1, 3.2, 4.1 and 5.1 ...
was changed to:
... retrieved at steps 2.1, 2.2, 2.3, 3.1, 3.2, 4.1 and 5.1 ...

**List of corrections regarding specific comments of the Anonymous Referee #1**

The authors of the manuscript are grateful to the Anonymous Referee #1 for the specific comments. The manuscript was substantially revised and the english has been hopefully improved. We hope that the paper is now easier to follow. Changes in the manuscript are highlighted by bold font.

***Page 1, lines 3-4:***
The MAC model is proposed combining chemical processes of the well-known photochemical models ...
was changed to:
The MAC model combines chemical processes of well-known photochemical models ...

***Page 1, line 6:***
... the multiple *in situ* nightglow emissions measured during the Energy Transfer
was changed to:
... the multiple nightglow emissions measured *in situ* during the Energy Transfer ...

***Page 1, line 7:***
... obtain concentrations of these MLT minor species is implemented ...
was changed to:
... obtain concentrations of these minor species in the MLT region is implemented ...

***Page 1, lines 10-11:***
... considered in the MAC model are identified and validated by calculations with the MAC model.
was changed to:
... considered in the MAC model are identified and validated.

***Page 1, line 22:***
... active MLT trace gas which is a critical component of ...
was changed to:
... active MLT trace gas and a critical component for ...

***Page 2, lines 25-26:***
Dynamic quenching reduces the apparent fluorescent lifetime, and static quenching rather reduces the apparent concentration . . .
was changed to:
Dynamic quenching reduces the apparent fluorescent lifetime, while static quenching rather reduces the apparent concentration . . .

***Page 2, line 28:***
. . . the dynamic quenching that can introduce difficulties . . .
was changed to:
. . . the dynamic quenching, and can introduce difficulties . . .

***Page 3, line 25:***
The ETON multiple airglow emissions are described in Section 2, they can be applied . . .
was changed to:
The ETON multiple airglow emissions described in Section 2 can be applied . . .

***Page 3, line 30:***
. . . scattered in the time and place that might have stopped Torr et al. (1985) combining . . .
was changed to:
. . . scattered in time and place and might have stopped Torr et al. (1985) from combining . . .

***Page 4, lines 19-20:***
The maximal number of VER profiles related to various $O_2$ and $O(^3P)$ transitions were obtained by two ETON rockets, which are discussed here. As for the Infrared Atmospheric band emissions at $1.27\,\mu$m, they were measured . . .
was changed to:
All VER profiles considered in the MAC model were measured during flights of two ETON rockets. The Infrared Atmospheric band emission at $1.27\,\mu$m was measured . . .

***Page 4, lines 28-29:***
. . . ground state $([O(^3P)])$ were carried out directly by the rockets P232H and P234H launched at . . .
was changed to:
. . . ground state $[O(^3P)]$ were carried out by the P232H and P234H rockets launched at . . .

***Page 4, lines 32-33:***
. . . (where $[O(^3P)]$ peak values are measured) . . . (where $[O(^3P)]$ low values are measured) . . .

was changed to:
... (where peak [O($^3P$)] values were measured) ... (where low [O($^3P$)] values were measured) ...

**_Specific comment:_**
I suggest adding a reference to the NRLMSISE-00 model here ..., and possibly also the MSIS-83 model ...

**_Page 5, lines 1-6:_**
The most recent version of the MSIS model, NRLMSISE-00 (Naval Research Laboratory MSIS Extended, 2000) ...
was changed to:
The most recent version of the MSIS model, NRLMSISE-00 (Naval Research Laboratory MSIS Extended, 2000, see Picone et al. (2002)) ...

**_Page 5, lines 1-6:_**
... the MSIS-83 model, which is not available anymore.
was changed to:
... the MSIS-83 model (Hedin, 1983) that is no longer available.

**_Page 5, line 10:_**
Some of O$_2$ transitions ...
was changed to:
Some of the O$_2$ transitions ...

**_Page 5, line 19:_**
... on the basis of data sets obtained ...
was changed to:
... on the basis of the data sets obtained ...

**_Specific comment:_**
Is there a manuscript in preparation with a title and author list that can be referenced here?

**_Page 5, lines 20-22:_**
Then, the MAC model was applied on the basis of data sets obtained during three campaigns as follows: the WADIS-2 (WAve propagation and DISsipation in the middle atmosphere), WAVE2000 (WAVes in airglow structures Experiment, 2000) and WAVE2004 campaigns, see the next publication.
was changed to:
Then, the MAC model was applied by Lednyts'kyy et al. (2019) on the basis of data sets obtained during the following three campaigns: the WADIS-2 (WAve propagation and DISsipation in the middle atmosphere), WAVE2000 (WAVes in airglow structures Experiment, 2000) and WAVE2004 campaigns.

**_Specific comment:_**
The reference "na – Nagy et al. (2008)" is written twice.

***Page 6, Table 1 caption:***
md – McDade (1998), na – Nagy et al. (2008), kh – Khomich et al. (2008).
was changed to:
md – McDade (1998), kh – Khomich et al. (2008).

***Page 7, line 6:***
... equation of McDade et al. (1986) provided here in the full form and in the short form ...
was changed to:
... equation of McDade et al. (1986) provided below in the full form ...

***Page 7, line 8:***
The well-known cubic equation provided by McDade et al. (1986) in the full form is as follows:
was changed to:
The cubic equation in the full form is as follows:

***Specific comment:***
Why can't the equation be in a single line, as equation 1?
***Response: Equation 2 on page 7 was changed to the equation in a single line. Additionally, Eqs. 3 and 4 on page 9 were changed to the equations in a single line. The changed equations were not highlighted.***

***Page 7, line 26:***
... using the semi-empirical models including MSIS-83, which is not available nowadays.
was changed to:
... using semi-empirical models, including MSIS-83 (Hedin, 1983), that are no longer available.

***Page 7, line 29:***
The minimal values of $C(0)$, $C(1)$ and $C(2)$ from all obtained ones, which are ...
was changed to:
The lowest obtained values of $C(0)$, $C(1)$ and $C(2)$, ...

***Page 7, line 31:***
... their maximal values were found to be equal to ...
was changed to:
... their highest values were found to be ...

***Page 9, line 13:***
... Kenner and Ogryzlo (1982), but Johnston and Broadfoot (1993) ...
was changed to:
... Kenner and Ogryzlo (1982). However, Johnston and Broadfoot (1993) ...

***Page 10, line 23-24:***
As for the well-known cubic Eq. (2), it solved . . .
was changed to:
As for the well-known cubic Eq. (2), it was solved . . .

***Page 10, line 27-28:***
. . . in this study, values of reaction rates and empirical coefficients provided by
. . .
was changed to:
. . . in this study, the values of the reaction rates and empirical coefficients used
are the ones provided by . . .

***Page 10, line 32:***
. . . photochemical models is provided in Section 1.
was changed to:
. . . photochemical models was provided in Section 1.

***Page 10, line 35:***
. . . second model, see Section 3.2.1, developed using available data sets.
was changed to:
. . . second model, see Section 3.2.2.

***Specific comment:***
Is "contiguously" the correct word to be used here or should "continuously" be
used instead?
***Page 12, line 6:***
and contiguously in the wavelength range
was changed to:
in the wavelength range

***Page 13, line 8:***
. . . (2014), see Table 3 in Section 3.2.1. These processes are related to:
was changed to:
. . . (2014) (see Table 3 in Section 3.2.1):

*Additionally, a similar change was carried out on page 14, line 5:*
. . . (1997), see Table 4 in Section 3.2.2. These processes are related to:
was changed to:
. . . (1997) (see Table 4 in Section 3.2.2):

*Additionally, a similar change was carried out on page 15, lines 11-12:*
. . . (1992), see Table 2 in Section 3.1. These processes are related to:
was changed to:
. . . (1992) (see Table 2 in Section 3.1):

***Page 14, lines 2-3:***
... MAC model, they are also referred to as M-processes.
was changed to:
... MAC model and are referred to as M-processes.

***Page 14, line 10:***
... MAC model, they are also referred to as H-processes.
was changed to:
... MAC model and are referred to as H-processes.

***Page 14, line 16:***
... G-model are also referred to as G-processes.
was changed to:
... G-model are referred to as G-processes.

***Specific comment:***
Reformulate this sentence, e.g. "Although the Barth excitation transfer scheme was formulated with $O_2^*$ considered as one not identified $O_2$ states, a group of many not identified $O_2$ states coupled in a cascade of de-excitation reactions is also possible." Or did I not understand the meaning of this sentence?
***Page 14, line 23-25:***
In fact, the Barth excitation transfer scheme was formulated with $O_2^*$ considering it as one not identified $O_2$ state or a one group of many not identified $O_2$ states coupled in a cascade of de-excitation reactions is also possible.
was changed to:
Although the Barth excitation transfer scheme was formulated with $O_2^*$ considered as one not identified $O_2$ state, a group of many not identified $O_2$ states coupled in a cascade of de-excitation reactions is also possible.

***Specific comment:***
How can Slanger et al. (2004b) have refuted the hypothesis by Huestis (2002) based on laboratory measurements discussed by Pejakovic at al. (2007)? A paper that was published 3 years after? I suggest to remove the reference of Pejakovic et al. (2007), or reformulate this section.
***Page 14, lines 26-27:***
... discussed by Huestis (2002), Slanger et al. (2004b) and Pejakovic et al. (2007) stating that energetically nearly resonant intermolecular processes are responsible for conversions of higher to lower excited $O_2$ electronic states according to Slanger and Copeland (2003).
was changed to:
... discussed by Huestis (2002) and Slanger et al. (2004b), and summarized by Pejakovic et al. (2007). Slanger and Copeland (2003) stated that energetically nearly resonant intermolecular processes are responsible for conversions of higher to lower excited $O_2$ electronic states.

***Page 14, line 29:***

... the de-excitation of $O_2$ states occurs not in a cascade-like process.
was changed to:
... the de-excitation of the $O_2$ states does not occur in a cascade-like process.

**Page 14, line 32:**
... removed converting to very high vibrational ...
was changed to:
... removed by conversion to very high vibrational ...

**Page 14, line 32 to page 15, line 1:**
... that $^5\Pi$ is the electronically excited $O_2$ state with the higher energy than
...
was changed to:
... that $^5\Pi$ is an electronically excited $O_2$ state with higher energy than ...

**Specific comment:**
Reformulate.
**Page 15, lines 16-19:**
These C-processes are shown here in Table 5, they were considered and discussed
by Lednyts'kyy et al. (2018). The corresponding reaction rates are shown in
Table 9. The C-processes related to the G-, M- and H-processes complete the
coupling of $O_2(^5\Pi, c, b, a, X)$ with each other and $O(^1S, {}^1D, {}^3P)$, they are re-
lated to:
was changed to:
These C-processes and the corresponding reaction rates are provided in Tables
5 and 9, respectively. The C-processes related to the G-, M- and H-processes
complete the coupling of $O_2(^5\Pi, c, b, a, X)$ with each other and $O(^1S, {}^1D, {}^3P)$:

**Page 16, line 8:**
The advantage of the ETON campaign compared to another rocket campaigns
...
was changed to:
The advantage of the ETON campaign compared to other rocket campaigns ...

**Page 16, line 19:**
... model are described in Section 2, they are: VER profiles ...
was changed to:
... model are described in Section 2 and include VER profiles ...

**Specific comment:**
Remove all "the" in front of "step".
**Page 17, lines 6-10:**
on the basis of R-VER$\{O_2(A-X)\}$ by using all relevant processes of the MAC
model. This retrieval step is shown as the step 2.1 in Table 8 and the step
2.1 described in Section A2.1 in Appendix A. Then, the verification of calculations at the step 2.1 is carried out comparing R-VER$\{O_2(A-X)\}$ with E-VER$\{O_2(A-X)\}$ and R-$[O_2(A)]$ with E-$[O_2(A)]$. The cubic equation is solved at the step 2.2 on the basis of T, $[N_2]$, $[O_2]$, R-VER$\{O_2(A'-a)\}$ and R-$[O_2(A)]$. Then, the verification of calculations at the step 2.2 is carried out comparing R-VER$\{O_2(A'-a)\}$ with E-VER$\{O_2(A'-a)\}$ and R-$[O_2(A')]$ with E-$[O_2(A')]$. was changed to:

on the basis of R-VER$\{O_2(A-X)\}$ using all relevant processes of the MAC model. This retrieval step is shown as step 2.1 in Table 8 and step 2.1 described in Section A2.1 in Appendix A. Then, the verification of calculations at step 2.1 is carried out comparing R-VER$\{O_2(A-X)\}$ with E-VER$\{O_2(A-X)\}$ and R-$[O_2(A)]$ with E-$[O_2(A)]$. The cubic equation is solved at step 2.2 on the basis of T, $[N_2]$, $[O_2]$, R-VER$\{O_2(A'-a)\}$ and R-$[O_2(A)]$. Then, the verification of calculations at step 2.2 is carried out comparing R-VER$\{O_2(A'-a)\}$ with E-VER$\{O_2(A'-a)\}$ and R-$[O_2(A')]$ with E-$[O_2(A')]$.

***Page 18, lines 9-10:***
... each retrieval step; these profiles also seem ...
was changed to:
... each retrieval step. These profiles also seem ...

***Page 19, line 1:***
... profiles) enables concluding that all ...
was changed to:
... profile) enables the conclusion that all ...

***Specific comment:***
Only SCH04 is somewhat defined in the caption of figure 1, none of the other anywhere in the manuscript.
***Page 20-21, Figures 1 and 2:***
All considered processes of the MAC model are provided in Tables 5, 6 and 7. Three-body recombination ...
was changed to:
All considered processes of the MAC model are provided in Tables 5, 6 and 7. Greer et al. (1981) (GLS$^+$81) and Huang and George (2014) (HG14) considered the G-processes, Mlynczak et al. (1993) (MSZ93) and Sharp et al. (2014) (SZB$^+$14) – the M-processes, and Lednyts 0 kyy and von Savigny (2016) (LvS16) – the C-processes. Three-body recombination ...

***Specific comment:***
What is the meaning of defining MMG$^+$86 and LSE$^+$15 in the text here? They are used later, and defined in the caption, in figure 4.
***Page 20, line 3-21:***
... McDade et al. (1986) (MMG$^+$86) was applied ...
was changed to:
... McDade et al. (1986) was applied ...

*Additionally, a similar change was carried out on page 21, line 2:*
. . . Lednyts'kyy et al. (2015) (LSE$^+$15) and . . .
was changed to:
. . . Lednyts'kyy et al. (2015) and . . .

**Specific comment:**
Is "equidistant" the correct word to use here? Suggestion: ". . . two profiles of extreme values."
***Page 21, lines 6-8:***
The [O($^3P$)] profile values retrieved according to the well-known and extended cubic equations are almost equidistant with respect to the *in situ* [O($^3P$)] profile values, and can be considered as two profiles of extrema values.
was changed to:
The [O($^3P$)] profile values retrieved according to the well-known and extended cubic equations can be considered as two profiles of extreme values because the *in situ* [O($^3P$)] profile values seem to be equidistant with respect to the retrieved ones.

**Specific comment:**
Suggestion: ". . . averaging of the extreme . . .". There are violet crosses in both panels in figure 4, what do you mean? ". . . see the violet crosses on the left in both figures."??
***Page 21, lines 8-9:***
One could assume that the arithmetical averaging of the extrema [O($^3P$)] profile values might be appropriate to finalize the [O($^3P$)] retrievals, see the violet crosses on the left in both figures.
was changed to:
One could assume that arithmetical averaging of the extreme [O($^3P$)] profile values might be appropriate to finalize the retrievals resulting in [O($^3P$)] profile values denoted by the violet crosses shown in both figures.

***Page 22, line 1:***
. . . in relation to rates of reactions in which . . .
was changed to:
. . . in relation to reaction rates in which . . .

***Page 22, line 2:***
Note processes . . .
was changed to:
Note that processes . . .

***Page 22, line 8:***
. . . retrieved at the pre-last step 4.1 on the basis of . . .
was changed to:
. . . retrieved at step 4.1 on the basis . . .

**Page 22, line 15:**
... are equal zero, whereas ...
was changed to:
... are equal to zero, whereas ...

**Page 22, line 16-17:**
... can not be shown because of the division by transition probabilities set to zero at the retrieval steps 2.1 and 2.2 for Fig. 5.
was changed to:
... can not be shown in Fig. 5 because of the division by transition probabilities set to zero at steps 2.1 and 2.2.

**Specific comment:**
Reformulate this sentence.
**Page 23, lines 3-5:**
Comparing VER and $[O(^3P)]$ profiles shown on the left in Figs. 3 and 5 with each other, it can be concluded that all calculations carried out using the MAC model excluding or involving $O_2(A)$ and $O_2(A')$ are all consistent with each other and coherent with measurements.
was changed to:
Values of VER profiles were compared with each other for two cases: (1) using the MAC model involving $O_2(A)$ and $O_2(A')$, see the left panel of Fig. 3, and (2) using the MAC model excluding $O_2(A)$ and $O_2(A')$, see the left panel of Fig. 5. This comparison enables concluding that the carried out calculations are consistent with each other leading to results coherent with measurements in both cases.

**Page 23, line 8:**
... values retrieved at the step 3.2 on the basis ...
was changed to:
... values retrieved at step 3.2 on the basis ...

**Page 23, line 9:**
... variable, and variabilities are higher than those ...
was changed to:
... variable with a variability higher than those ...

**Specific comment:**
"... (see the violet crosses on the left in this figure), ..."? Do you mean "... the violet crosses in the left panel of this figure, ..."?
**Page 24, Figure 4 caption:**
... (see the violet crosses on the left in this figure), ...
was changed to:
... (see the violet crosses in the left panel of this figure), ...

**Page 24, lines 1-2:**
[O($^3P$)] profile values retrieved at the other steps are in good agreement with those of the *in situ* ETON [O($^3P$)] profile, but [O($^3P$)] profile values retrieved at the step 3.2 are in disagreement with all [O($^3P$)] profile values mentioned here.
was changed to:
[O($^3P$)] profile values retrieved at step 3.2 do not agree with the *in situ* ETON [O($^3P$)] profile values to the degree the [O($^3P$)] profile values retrieved at the other steps agree.

**Page 24, line 5:**
It should be mentioned that the vibrational population of OH($\nu'$) has to be known in order to consider the reaction $R_{h2.1}$ shown in Table 7 (OH$^*$+O($^3P$) $\xrightarrow{\eta_{OH}^{3P}}$ H + O$_2$, where OH$^*$ describes the hydroxyl radical in all possible levels $\nu'$) in the MAC model.
was changed to:
The reaction $R_{h2.1}$ implemented in the MAC model and shown in Table 7 is similar to that considered by Llewellyn and Solheim (1978): OH$^*$ + O($^3P$) $\xrightarrow{\eta_{OH}^{3P}}$ H + O$_2$, where OH$^*$ describes the hydroxyl radical in all possible levels $\nu'$. It should be mentioned that it would be possible to retrieve [O($^3P$)] if the vibrational population of OH($\nu'$) were known.

**Page 25, line 8:**
Unfortunately, it would be not enough . . .
was changed to:
Unfortunately, it would not be enough . . .

**Page 26, line 7:**
. . . at each of the following retrieval steps provided in Table 8.
was changed to:
. . . at each of the retrieval steps listed in Table 8.

**Page 27, Figure 6 caption:**
The retrievals were performed at the steps . . .
was changed to:
The retrievals were performed at steps . . .

**Page 28, line 34:**
. . . peak values, see Section 2). Varying . . .
was changed to:
. . . peak values, see Section 2. Varying . . .

**Specific comment:**
". . ., see the next article to be submitted."?? Is there a manuscript in preparation with a title and author list that can be referenced here?

***Page 29, line 19:***
. . ., see the next article to be submitted.
was changed to:
. . ., see Lednyts'kyy et al. (2019) for details.

***Specific comment to page 32, lines 20-21:***
Is there a manuscript in preparation with a title and author list that can be referenced here?
***Response: The mentioned sencence was deleted.***

***Page 32, line 34:***
. . . not well known because it had been detected by Cacace et al. (2001) recently, may be . . .
was changed to:
. . . not well known because it has only recently been discovered by Cacace et al. (2001), may be . . .

***Specific comment:***
"," missing between $^1D$ and $^3P$ in two places.
***Page 33, lines 22-23:***
. . . and $O(^1S, \, ^1D\,^3P)$ as well as related to $O_2(^5\Pi, \, c, \, b, \, a, \, X)$ and $O(^1S, \, ^1D\,^3P)$.
was changed to:
. . . and $O(^1S, \, ^1D, \, ^3P)$ as well as related to $O_2(^5\Pi, \, c, \, b, \, a, \, X)$ and $O(^1S, \, ^1D, \, ^3P)$.

*Additionally, a similar change was carried out on page 35, line 2:*
. . . atomic oxygen, $O(^1S, \, ^1D\,^3P)$.
was changed to:
. . . atomic oxygen, $O(^1S, \, ^1D, \, ^3P)$.

***Page 35, line 5:***
. . . were combined with complementary processes suggested to complete the list . . .
was changed to:
. . . were combined with suggested complementary processes to complete the list . . .

***Page 37, line 9:***
. . . provided in Section 3.4 in Table 8 for the MAC model.
was changed to:
. . . provided in Section 3.4 and in Table 8.

***Page 37, line 13:***
. . . are equal to zero values.
was changed to:
. . . are equal to zero.

***Page 37, line 14:***
. . . according to the models, processes of which were adopted in the MAC model.
was changed to:
. . . according to the processes of the different models adopted in the MAC model.

***Specific comment:***
". . . in the next publication . . ."?? Is there a manuscript in preparation with a title and author list that can be referenced here?
***Page 37, line 5:***
. . . described in the next publication provide data sets required in the prior retrieval to apply the MAC model.
was changed to:
. . . represent data sets required at the prior retrieval step applied by Lednyts'kyy et al. (2019).

***Specific comment:***
". . . considering the processes shown in Tables 6. . .5 . . ."? Do you mean Tables 5 and 6, or Tables 6 to some other number higher than 6?
***Page 40, line 30:***
. . . considering the processes shown in Tables 6. . .5 . . .
was changed to:
. . . considering the processes shown in Tables 5 and 6 . . .

*Additionally, a similar change was carried out on page 39, line 18:*
. . . considering the processes shown in Tables 6. . .5 . . .
was changed to:
. . . considering the processes shown in Tables 5 and 6 . . .

***Specific comment:***
". . . considering the processes shown in Tables 6. . .5 . . ."? Do you mean Tables 5 and 6, or Tables 6 to some other number higher than 6?
***Page 42, lines 8-9:***
. . . considering the processes shown in Tables 6. . .5 . . .
was changed to:
. . . considering the processes shown in Tables 5 and 6 . . .

***Specific comment:***
". . . considering the processes shown in Tables 6. . .5 . . ."? Do you mean Tables 5 and 6, or Tables 6 to some other number higher than 6?
***Page 42, line 28:***
. . . considering the processes shown in Tables 6. . .5 . . .
was changed to:
. . . considering the processes shown in Tables 5 and 6 . . .

***Specific comment:***

"... considering the processes shown in Tables 6...5 ..."? Do you mean Tables 5 and 6, or Tables 6 to some other number higher than 6?

***Page 44, line 8:***
... considering the processes shown in Tables 6...5 ...
was changed to:
... considering the processes shown in Tables 5 and 6 ...

***Specific comment:***
"... considering the processes shown in Tables 6...5 ..."? Do you mean Tables 5 and 6, or Tables 6 to some other number higher than 6?

***Page 45, line 11:***
... considering the processes shown in Tables 6...5 ...
was changed to:
... considering the processes shown in Tables 5 and 6 ...

***Specific comment:***
All references in the text are listed in the reference list and vice versa.

***Response regarding the word "References" shown in page 48, line 1:***
The authors of the article thank the Anonymous Referee #1 for the suggestion, but will not adopt this suggestion. Instead, the authors follow the JASTP suggestion regarding the names in the reference list.

***Page 56, Table 9 caption:***
Processes of the provided here rate values are shown in Table 5.
was changed to:
Rate values of the processes listed in ...

***Page 57, Table 10 caption:***
Processes of the provided here rate values are shown in Table 6.
was changed to:
Rate values of the processes listed in ...

***Page 58, Table 11 caption:***
Processes of the provided here rate values are shown in Tables 6 and 7.
was changed to:
Rate values of the processes listed in ...

***Page 59, Table 12 caption:***
Processes of the provided here rate values are shown in Tables 7.
was changed to:
Rate values of the processes listed in ...

*Comments of the Anonymous Referee #2*
*Feedback to the pdf-paper:*
I believe that this paper contains a valuable contribution to the field and as such is publishable. However the paper is extremely difficult to read and requires major revision of the structure and language before it can be published.
*General comments to the pdf-paper:*
The paper is attempting a complex task and may be helped by some flow charts show in the procedures, the relationships between various products etc.
As I have managed to understand the paper: The authors first review a substantial literature on many of the airglow emissions contrasting and comparing relevant groups and then make selections as to which reactions to maintain in the MAC model.

*List of corrections regarding general comments of the Anonymous Referee #2*
The authors of the manuscript are grateful to the Anonymous Referee #2 for the general comments. We agree that the paper was difficult to follow. We edited the paper and changed the text substantially. The paper is now hopefully easier to follow. Also, following the reviewer's suggestion a flow chart was added as Fig. A1. Changes in the manuscript were highlighted by bold font.

*Page 37, line 14, regarding the new third paragraph:*
Retrieval steps resulting in $[O(^3P)]$ and carried out according to the proposed algorithm are illustrated in the flow chart in Fig. A1.

*Additionally, the new Fig. A1 was added on page 37. This figure is shown as Fig. 1 in the Final Response and as the supplement file at* https://www.atmos-chem-phys-discuss.net/acp-2019-221/acp-2019-221-AC2-supplement.pdf.

*Additionally, the following relevant change was carried out in Table 9 on page 56, Table 10 on page 57, Table 11 on page 58, Table 12 on page 59 and Table A1 on page 60. These tables were sorted and placed not after the bibliography as was done earlier, but in the respective sections.*

*Additionally, the following relevant change was carried out in Table 12 on page 27:*
*The content of Table 12 was revised and compressed representing concentrations of many excited species, e.g. $[O_2(A, A')]$ instead of $[O_2(A)]$, $[O_2(A')]$ and so on. Note that this change was not highlighted.*

*Additionally, the following relevant change was carried out in Table 7 on page 19:*
*$\ldots R_{s2.1-6} \ldots$*

was changed to:
$\ldots R_{s2.1-3} \ldots R_{s2.4-6} \ldots$

**List of corrections regarding specific comments of the Anonymous Referee #2**

The authors of the manuscript are grateful to the Anonymous Referee #2 for the specific comments. The manuscript was substantially revised and the english has been hopefully improved. We hope that the paper is now easier to follow. Changes in the manuscript were highlighted by bold font.

**Specific comment:**

Some tests are then carried out but I was unable to follow exactly what was done. I believe that Figure 3 is supposed to show the agreement between the model after tuning with the various ETON emission profiles. I do not see the point in showing both the concentration of the emitter and the intensity of the emissions as the conversion between them is trivial.

**Response:**

Thank you for reminding about the triviality of the conversion between the concentration of the emitter and the intensity of the emissions. We would like to keep the text about this conversion in Appendix A describing steps in sections A2.1, A2.2, A2.3, A3.2 and A4.1. The corresponding consistency tests are described in Appendix A in Sections A2.4, A3.3 and A4.2. We would also like to keep both panels in Fig. 3 as the result of the carried out consistency tests. We substantiate our choice by the fact that Huang and George (2014) verified their calculations comparing concentration profiles in their Fig. 1. Therefore, we also show concentration profiles in the right panel of our Fig. 3 to allow the comparison of calculation results shown by Huang and George (2014) in their Fig. 1 and our results shown in the right panel of our Fig. 3 easier, i.e. without additional calculations required for this conversion. In fact, the MAC model includes reactions proposed by Huang and George (2014) as well as many other reactions. Note that we employ updated reaction rate values considered by Huang and George (2014) in their Table 1. This makes the comparison of concentration profiles obtained by us and Huang and George (2014) necessary. Additionally, we compare VER profiles in the left panel of our Fig. 3 as was done by Murtagh et al. (1990).

*Following your comment, the following relevant change was carried out in caption of Fig. 3 on page 23:*
... are coherent with measurements. The abbreviations ...
was changed to:
... are coherent with measurements. The corresponding consistency tests are described in Sections A2.4, A3.3 and A4.2. Note that the conversion between profile values of VER and concentrations is based on trivial, but required calculations provided in Sections A2.1, A2.2, A2.3, A3.2 and A4.1. The abbreviations
...

***Specific comment:***
The sensitivity analysis in section 3.5 is a useful contribution. I might note
that the temperature cannot actually be varied independently of the pressure
because of hydrostatic equilibrium.

***Response:***
The reviewer is certainly correct that temperature and pressure are directly
connected. However, it may occur that one of the two quantities is measured
with a very high accuracy, and the other one is not. We would like to keep
the sensitivity analysis considering the mentioned parameters, i.e. $[N_2]$ and
$[O_2]$ (calculated using pressure) as well as temperature, separately because of
two reasons. Firstly, instruments aboard the WADIS-2 rocket and the SABER
instrument provided data required to obtain pressure and temperature. The
sensitivity analysis carried out by Lednyts'kyy et al. (2019) is based on pres-
sure and temperature considered independently from each other. The sensitivity
analysis discussed in our Section 3.5 is carried out in a similar way as was done
by Lednyts'kyy et al. (2019). Finally, the mentioned parameters are also consid-
ered independently from each other in the continuity equations employed here.
In summary, we would like to keep the applicability of the sensitivity analysis
because it contains more information than treating temperature and pressure
in a combined way.

***Specific comment:***
Much of the content of section 4 - discussion should appear earlier in the paper
to explain what is actually done in the model. For instance the tuning of pa-
rameters is described first in section 4.1.
*The text on page 28, lines 12-17, was used to extend the text on page 14, line
32:*
... states. It should be noted that ...
was changed to:
... states. In fact, the removal of the $O_2(^5\Pi)$–$O_2(A, A')$–group through colli-
sions was suggested by Slanger et al. (2004b) and implemented in the MAC
model implicitly by increasing the association rates of $O_2(b, a, X)$ in the three-
body recombination reactions. This was done implicitly because reactions in-
cluding $O_2(^5\Pi)$ are not well known, e.g., compare Krasnopolsky (2011) and
Krasnopolsky (1986). It should be noted that $O_2(^5\Pi)$ has a shorter lifetime and
a higher energy compared to the other states $O_2(A, A', c, b, a, X)$ as it was also
mentioned by Huestis (2002) and Slanger et al. (2004b). It should be noted
that ...

*Additionally, the following relevant change was carried out in the title of Section
4.1 on page 29, line 9:*

Discussion of tuned rate values of quenching processes implemented in the MAC model.
was changed to:
Tuning rate values of quenching processes implemented in the MAC model.

*Additionally, the text of Section 4.1 on pages 29 and 30 was written after Section 3.3 on page 16. Note that the added text was not highlighted.*

**Specific comment:**
The discussion of dynamics on page 32 seems somewhat superfluous to the aim of the paper and could be excluded.
**Response: The mentioned text on page 32, lines 16-21, and on page 5, lines 14-17, was deleted.**

*Additionally, the reference to Smith et al. (1987) on page 54, lines 19-21, was deleted. The reference to Johnson and Gottlieb (1973) on page 50, lines 17-18, was also deleted.*

**Specific comment:**
The use of some wording also confuses the reader. I believe that the use of "continuity equation" in for instance line 9 page 32 refers to a steady state chemical balance equation although some wording earlier in the paper may suggest that time dependent equations are being solved.
*Page 32, line 10:*
Continuity equations implemented . . .
was changed to:
Steady state chemical balance equations (also referred to as continuity equations) implemented . . .

*Additionally, a similar change was carried out on page 2, line 30:*
. . . the resulting mass conservation (continuity) equation . . .
was changed to:
. . . the resulting steady state chemical balance equation (hereafter referred to as continuity equation) . . .

*Additionally, a similar change was carried out on page 37, line 6:*
Then simple continuity equations . . .
was changed to:
Then simple steady state chemical balance equations (referred to as continuity equations) . . .

**Specific comment:**
The use of the word retrieve when I believe calculate would be better is also a problem. Eg line 10 page 38, line 12 page 39.
*The following change was carried out in titles of each subsection of the Appendix A (e.g. A2.1, A2.2, A2.3 and A2.4, but not A2) and for references in Appendix*

*A to the mentioned sections:*
... retrieval ...
was changed to (and highlighted in bold italic font):
... calculation ...

**Specific comment:**
A few minor points Page 4 line 22 - The reference to Greer et al 1986 about the atomic oxygen measurement technique would be better replaced but a reference to one of the Dickinson papers.
**Page 4, line 30:**
... at $\sim$130 nm (Greer et al., 1986) and ...
was changed to:
... at $\sim$130 nm (Dickinson et al., 1980) and ...

**Specific comment:**
Page 12 second line of the caption to table 3. I believe it should be the character E that marks the equations excluded not M.
**Page 5, line 2, caption of Table 3:**
The processes marked with a character $M$ are not considered ...
was changed to:
The processes marked with a character $E$ are not considered ...

**Specific comment:**
Page 28. In the discussion of the vibrational distribution of the Herzberg states a reference to some of the ground based work by Slanger or Stegman might be appropriate.
**Page 28, line 21:**
... energetically very close to each other. Vibrational states ...
was changed to:
... energetically very close to each other. Vibronic energy levels of $O_2(A, A', c, b, a, X)$ are shown in Fig. 8 by Goodman and Brus (1977). The atlas of terrestrial nightglow emission lines in the range 314...1043 nm including emission lines of these $O_2$ states is provided in Table 3 as a compressed form of the electronic supplement of Cosby et al. (2006). Vibrational states ...

*Additionally, the following relevant change was carried out on page 28, line 10:*
... coupled with $O_2(c, b, a, X)$ which was also adopted in the MAC model.
was changed to:
... coupled with $O_2(c, b, a, X)$.

*Discussion with C. von Savigny resulted in the following changes.*

*The following change was carried out in Table 2 on page 8, Table 3 on page 12, Table 4 on page 13, Table 5 on page 17, Table 6 on page 18, Table 7 on page 19, and Table A1 on page 60:*

$\ldots h\nu(\ldots$
was changed to:
$\ldots h\nu(\ldots$

*Additionally, the following change was carried out on page 4, lines 19-20:*
The maximal number of VER profiles related to various $O_2$ and $O(^3P)$ transitions were obtained by two ETON rockets, which are discussed here. As for the Infrared Atmospheric band emissions at $1.27\,\mu m$, they were measured $\ldots$
was changed to:
All VER profiles considered in the MAC model were measured during flights of two ETON rockets. The Infrared Atmospheric band emission at $1.27\,\mu m$ was measured $\ldots$

*Additionally, the following change was carried out on page 11, lines 17-19:*
In contrast to the YM2011 model, reaction rates in the modified kinetic model of Mlynczak et al. (1993) refer to a specific portion of vibrational states from their statistical equilibrium in each $O_2$ electronic state.
was changed to:
Rate values of reactions involving $O_2(b)$ and $O_2(a)$ in the modified kinetic model of Mlynczak et al. (1993) do not directly correspond to rate values of reactions involving various vibrational states of $O_2(b)$ and $O_2(a)$ in the YM2011 model because vibrational states are not identified in the modified kinetic model of Mlynczak et al. (1993).

*Additionally, the following change was carried out on page 30, lines 9-10:*
$\ldots$ the results of which are given by the interval of possible rate values shown in the third column of Tables 8$\ldots$11.
was changed to:
$\ldots$ see the third column of Tables 8, 9, 10 and 11 for a summary.

*Additionally, the following change was carried out on page 28, lines 24-25:*
Considering internuclear distances (INDs) of the corresponding Franck-Condon factors, it should be emphasized that the difference in INDs between the excited $\ldots$
was changed to:
Considering Franck-Condon factors and the corresponding internuclear distances (INDs), it should be emphasized that the difference in INDs between the excited $\ldots$

*Additionally, the following change was carried out on page 34, lines 30-32:*
It is difficult to argue about the fractional $O_2(c)$ vibrational population of the $O(^1S)$ precursor because the $[O_2(c,\,\nu=6,\,7,\,8)]$ peak might be at about $97\,km$ altitude if the collisional activation energy mentioned by Krasnopolsky (1981) is neglected.
was changed to:
In summary, the exact role of the vibrational excitation of $O_2(c)$ as a precursor

of $O(^1S)$ is still not well understood and should be investigated in future studies.

*Additionally, the following change was carried out on page 36, lines 19-21:*
Then, the following correspondences among the violated electric dipole selection rules and the transition intensity seem to be established. ... (3) almost absent between $O_2(A^3\Sigma_u^+)$ and $O_2(A'^3\Delta_u)$ as well as $O_2(c^1\Sigma_u^-)$ and $O_2(a^1\Delta_g)$.
was changed to:
Then, the following correspondences regarding the selection rules for chemical reactions were established. ... (3) almost nearly absent between $O_2(A^3\Sigma_u^+)$ and $O_2(A'^3\Delta_u)$ as well as between $O_2(c^1\Sigma_u^-)$ and $O_2(a^1\Delta_g)$.

*Additionally, the following change was carried out on page 40, lines 25-27:*
This is justified because $O_2$ in these triplet excited states is decoupled from the singlet excited states according to the hypothesis of Slanger et al. (2004b) used to propose the MAC model, which were verified and validated, (see Section 3.5.
was changed to:
This is justified because the hypothesis of Slanger et al. (2004b) was adopted to propose the MAC model. Note that the MAC calculations were verified and validated, see Section 3.5 for details.

*Additionally, the following change was carried out on page 43, lines 24-25:*
... was not retrieved on the basis of VER values, but concentrations available from the previous calculation steps.
was changed to:
... was retrieved on the basis of concentrations available from the previous calculation steps, whereas VER profiles were not employed for the $[O_2(c)]$ retrievals directly.

*Additionally, discussion with C. von Savigny resulted in the update of references on page 48 line 10, page 51 line 10, page 51 line 16, page 51 line 18, and page 51 line 22.*

*Additionally, discussion with C. von Savigny resulted in numerous changes regarding English grammar and style. These relevant changes were carried out and highlighted in the manuscript.*

[Figure]

[revised manuscript text omitted]
_{\rm N2}^{\rm DP}, \rho_{\rm Oa}^{\rm DP}, \rho_{\rm Ob}^{\rm DP}, \rho_{\rm C2}^{\rm DP}}$ O($^3P$) + {N$_2$, O$_2(a)$, O$_2(b)$, CO$_2$} |
| $R_{r3.0}$ | O($^1D$) $\xrightarrow{\rho_{\rm 1D3Pe}^{\rm A}}$ O($^3P$) + $h\nu$ |

| R$_\#$ | Odd oxygen processes related to the loss of atomic oxygen |
|---|---|
| $R_{x1.1-2}$ | O($^3P$) + O($^3P$) + {N$_2$, O$_2$} $\xrightarrow{\chi_{\rm N2}^{\rm Px}, \chi_{\rm O2}^{\rm Px}}$ O$_2$ + {N$_2$, O$_2$} |

| R$_\#$ | Odd oxygen processes related to catalytic ozone destruction and photolysis |
|---|---|
| $R_{x2.1}$ | O($^3P$) + O$_3$ $\xrightarrow{\chi_{\rm O2}^{\rm 3P}}$ 2O$_2$ |
| $R_{x3.1-2}$ | O$_2$ + O($^3P$) + {N$_2$, O$_2$} $\xrightarrow{\chi_{\rm N2}^{\rm P3}, \chi_{\rm O2}^{\rm P3}}$ O$_3$ + {N$_2$, O$_2$} |
| $R_{s1.1-5}$ | O$_2$ + $h\nu$ $\xrightarrow{\sigma_{\rm PS}^{\rm UV}, \sigma_{\rm PD}^{\rm LA}, \sigma_{\rm PD}^{\rm Sc}, \sigma_{\rm PP}^{\rm Sb}, \sigma_{\rm PP}^{\rm 
[revised manuscript text omitted]

| $R_{t8.0}$ | | $\theta_{320n}^{A} = 11$ | $\text{s}^{-1}$ | r08 |
| $R_{t9.0}$ | | $\theta_{HI}^{A} = 11$ | $\text{s}^{-1}$ | r05 |
| $R_{t10.1}$ | | $\theta_{1S}^{tx} = 1 \cdot 10^{-14}$ as $\theta_{1S}^{tx} \in [1 \cdot 10^{-30}, 1 \cdot 10^{-14}]$ | $\text{molec}^{-1}\,\text{cm}^3\,\text{s}^{-1}$ | r03 |
| $R_{d1.1}$ | | $\delta_{N2}^{Pd} = dY \cdot 3 \cdot 10^{-33}(300/T)^{3.25}$ | $\text{molec}^{-2}\,\text{cm}^6\,\text{s}^{-1}$ | r01 |
| $R_{d1.2}$ | | $\delta_{O2}^{Pd} = dY \cdot 3 \cdot 10^{-33}(300/T)^{3.25}$ | $\text{molec}^{-2}\,\text{cm}^6\,\text{s}^{-1}$ | r01 |
| | | $dY = 0.18$ | 1 | r02 |
| $R_{d2.1-2}$ | | $\delta_{3P}^{dc} = cDCu \cdot \delta_{3P}^{tx}, \delta_{O2}^{dc} = cDCu \cdot \delta_{O2}^{tx}$ | $\text{molec}^{-1}\,\text{cm}^3\,\text{s}^{-1}$ | r03 |
| | | $cDCu = 1 \cdot 10^{-2}$ close to $cDCu \in [1 \cdot 10^{-30}, 1 \cdot 10^{-3}]$ | 1 | r03 |
| $R_{d3.1-2}$ | | $\delta_{3P}^{db} = cDBu \cdot \delta_{3P}^{tx}, \delta_{O2}^{db} = cDBu \cdot \delta_{O2}^{tx}$ | $\text{molec}^{-1}\,\text{cm}^3\,\text{s}^{-1}$ | r03 |
| | | $cDBu = 1 \cdot 10^{-2}$ as $cDBu \in [1 \cdot 10^{-30}, 1 \cdot 10^{-2}]$ | 1 | r03 |
| $R_{d4.1-2}$ | | $\delta_{3P}^{da} = cDAu \cdot \delta_{3P}^{dx}, \delta_{O2}^{da} = cDAu \cdot \delta_{O2}^{dx}$ | $\text{molec}^{-1}\,\text{cm}^3\,\text{s}^{-1}$ | r03 |
| | | $cDAu = 1 \cdot 10^{-2}$ as $cDAu \in [1 \cdot 10^{-30}, 1 \cdot 10^{-2}]$ | 1 | r03 |
| $R_{d5.0}$ | | $\delta_{370n}^{A} = 0.85$ | $\text{s}^{-1}$ | r08 |
| $R_{d6.0}$ | | $\delta_{Cha}^{A} = 0.85$ | $\text{s}^{-1}$ | r05 |
| $R_{d7.1}$ | | $\delta_{3P}^{dx} = 1.3 \cdot 10^{-11}$ | $\text{molec}^{-1}\,\text{cm}^3\,\text{s}^{-1}$ | r06 |
| $R_{d7.2}$ | | $\delta_{O2}^{dx} = 1.7 \cdot 10^{-11}$ | $\text{molec}^{-1}\,\text{cm}^3\,\text{s}^{-1}$ | r09 |
| $R_{d8.0}$ | | $\delta_{HIII}^{A} = 0.9$ | $\text{s}^{-1}$ | r05 |
| $R_{d9.1}$ | | $\delta_{1S}^{dx} = 1 \cdot 10^{-14}$ as $\delta_{1S}^{dx} \in [1 \cdot 10^{-30}, 1 \cdot 10^{-14}]$ | $\text{molec}^{-1}\,\text{cm}^3\,\text{s}^{-1}$ | r03 |

**Table 9. Rate values of the processes listed in Table 6.** References: r10 - Predoi-Cross et al. (2008), r11 - Slanger (1978), r12 - Kenner and Ogryzlo (1983), r13 - Burkholder et al. (2015), r14 - Minaev and Ågren (1997). Labels r01, r02, r03, r06, r08 were used in Table 8. The enthalpy change ($\Delta H$) was determined at standard temperature and pressure, see Table 11 for abbreviations.

| $R_\#$ | $\Delta H$ (eV) | Rate value | Rate unit | Ref. |
|---|---|---|---|---|
| $R_{c1.1-2}$ | | $\varsigma_{N2}^{Pc} = \varsigma_{O2}^{Pc} = cY \cdot 3 \cdot 10^{-33}(300/T)^{3.25}$ | $\text{molec}^{-2}\,\text{cm}^6\,\text{s}^{-1}$ | r01 |
| | | $cY = 0.04$ | 1 | r02 |
| $R_{c2.1}$ | | $\varsigma_{1S}^{cx} = 1.4 \cdot 10^{-8}$ | $\text{molec}^{-1}\,\text{cm}^3\,\text{s}^{-1}$ | r03 |
| $R_{c3.1}$ | | $\varsigma_{3P}^{cb} = cCBa \cdot \varsigma_{3P}^{cx}$ | $\text{molec}^{-1}\,\text{cm}^3\,\text{s}^{-1}$ | r03 |
| $R_{c3.2}$ | | $\varsigma_{O2}^{cb} = cCBm \cdot \varsigma_{O2}^{cx}$ | $\text{molec}^{-1}\,\text{cm}^3\,\text{s}^{-1}$ | r03 |
| | | $cCBa = 5.8 \cdot 10^4$ | 1 | r03 |
| | | $cCBm = 1 \cdot 10^{-1}$ as $cCBm \in [1 \cdot 10^{-30}, 1 \cdot 10^{-1}]$ | 1 | r03 |
| $R_{c4.0}$ | | $\varsigma_{cbK}^{A} = \varsigma_{RJ}^{A}/10$ | $\text{s}^{-1}$ | r03 |
| $R_{c5.1}$ | | $\varsigma_{3P}^{ca} = cCAa \cdot \varsigma_{3P}^{cx}$ | $\text{molec}^{-1}\,\text{cm}^3\,\text{s}^{-1}$ | r03 |
| $R_{c5.2}$ | | $\varsigma_{O2}^{ca} = cCAm \cdot \varsigma_{O2}^{cx}$ | $\text{molec}^{-1}\,\text{cm}^3\,\text{s}^{-1}$ | r03 |
| | | $cCAa = 1 \cdot 10^{-1}$ close to $cCAa \in [1 \cdot 10^{-30}, 1 \cdot 10^{+3}]$ | 1 | r03 |
| | | $cCAm = 1 \cdot 10^{-1}$ close to $cCAm \in [1 \cdot 10^{-30}, 1]$ | 1 | r03 |
| $R_{c6.0}$ | | $\varsigma_{RJ}^{A} = 0.073$ | $\text{s}^{-1}$ | r11 |
| $R_{c7.1}$ | | $\varsigma_{3P}^{cx} = 6 \cdot 10^{-12}$ | $\text{molec}^{-1}\,\text{cm}^3\,\text{s}^{-1}$ | r12 |
| $R_{c7.2}$ | | $\varsigma_{O2}^{cx} = 1.8 \cdot 10^{-11}$ | $\text{molec}^{-1}\,\text{cm}^3\,\text{s}^{-1}$ | r06 |
| $R_{c8.0}$ | | $\varsigma_{HII}^{A} = 0.66$ | $\text{s}^{-1}$ | r08 |
| $R_{b1.1-2}$ | $-3.49^{E}$ | $\beta_{N2}^{Pb} = \beta_{O2}^{Pb} = bY \cdot 3 \cdot 10^{-33}(300/T)^{3.25}$ | $\text{molec}^{-2}\,\text{cm}^6\,\text{s}^{-1}$ | r01 |
| | | $bY = 0.03 + pY \cdot 0.07$ | 1 | r03 |
| | | $pY = 0.5$ (for $O_2(^5\Pi)$) | 1 | r02 |
| $R_{b2.1}$ | $-0.65^{A}$ | $\beta_{O3}^{ba} = 0.15 \cdot 3.5 \cdot 10^{-11}\exp(-135/T)$ | $\text{molec}^{-1}\,\text{cm}^3\,\text{s}^{-1}$ | r13 |
| $R_{b2.2}$ | $-0.65^{E}$ | $\beta_{3P}^{ba} = cBAa \cdot \beta_{3P}^{bx}$ | $\text{molec}^{-1}\,\text{cm}^3\,\text{s}^{-1}$ | r03 |
| $R_{b2.3}$ | $-0.65^{A}$ | $\beta_{N2}^{ba} = cBAm \cdot \beta_{N2}^{bx}$ | $\text{molec}^{-1}\,\text{cm}^3\,\text{s}^{-1}$ | r03 |
| $R_{b2.4}$ | $-0.65^{A}$ | $\beta_{O2}^{ba} = cBAm \cdot \beta_{O2}^{bx}$ | $\text{molec}^{-1}\,\text{cm}^3\,\text{s}^{-1}$ | r03 |
| $R_{b2.5}$ | $-0.65^{E}$ | $\beta_{C2}^{ba} = cBAm \cdot \beta_{C2}^{bx}$ | $\text{molec}^{-1}\,\text{cm}^3\,\text{s}^{-1}$ | r03 |
| | | $cBAa = cBAm = 1 \cdot 10^{-1}$ as $cBAa, cBAm \in [1 \cdot 10^{-30}, 1 \cdot 10^{-1}]$ | 1 | r03 |
| $R_{b3.0}$ | $0.65^{E}$ | $\beta_{Nox}^{A} = 0.0014$ | $\text{s}^{-1}$ | r14 |
| $R_{b4.1}$ | $-1.63^{A}$ | $\beta_{O3}^{bx} = 0.7 \cdot 3.5 \cdot 10^{-11}\exp(-135/T)$ | $\text{molec}^{-1}\,\text{cm}^3\,\text{s}^{-1}$ | r13 |
| $R_{b4.2}$ | $-1.63^{E}$ | $\beta_{3P}^{bx} = 8 \cdot 10^{-14}$ | $\text{molec}^{-1}\,\text{cm}^3\,\text{s}^{-1}$ | r13 |
| $R_{b4.3}$ | $-1.63^{A}$ | $\beta_{N2}^{bx} = 1.8 \cdot 10^{-15}\exp(45/T)$ | $\text{molec}^{-1}\,\text{cm}^3\,\text{s}^{-1}$ | r13 |
| $R_{b4.4}$ | $-1.63^{A}$ | $\beta_{O2}^{bx} = 3.9 \cdot 10^{-17}$ | $\text{molec}^{-1}\,\text{cm}^3\,\text{s}^{-1}$ | r13 |
| $R_{b4.5}$ | $-1.63^{E}$ | $\beta_{C2}^{bx} = 4.2 \cdot 10^{-13}$ | $\text{molec}^{-1}\,\text{cm}^3\,\text{s}^{-1}$ | r13 |
| $R_{b4.6}$ | $-1.63^{A}$ | $\beta_{O3}^{bx} = 0.15 \cdot 3.5 \cdot 10^{-11}\exp(-135/T)$ | $\text{molec}^{-1}\,\text{cm}^3\,\text{s}^{-1}$ | r13 |
| $R_{b5.0}$ | $1.63^{E}$ | $\beta_{762}^{A} = 0.079$ | $\text{s}^{-1}$ | r08 |
| $R_{b6.0}$ | $1.63^{E}$ | $\beta_{Atm}^{A} = 0.083$ | $\text{s}^{-1}$ | r10 |

**Table 10. Rate values of the processes listed in Tables 6 and 7.** References: r15 - Pendleton et al. (1996), r16 - Krauss and Neumann (1975), r17 - Capetanakis et al. (1993), r18 - Gordiets et al. (1995), r19 - Atkinson and Welge (1972), r20 - Kenner and Ogryzlo (1982), r21 - Kramida et al. (2015), r22 - Pinheiro et al. (1998), r23 - Sakai et al. (2014). Labels r01, r02, r03 were used in Table 8, and labels r13, r14 were used in Table 9. The enthalpy change ($\Delta$H) was determined at standard temperature and pressure, see Table 11 for abbreviations.

[revised manuscript text omitted]

15  $P\{O(^1S)\text{-M}\} + P\{O(^1S)\text{-H}\} + P\{O(^1S)\text{-C}\}$, where $P\{O(^1S)\text{-M}\}$ **is absent**, $P\{O(^1S)\text{-H}\} = [O(^3P)]R_{c2.1}[O_2(c)]$ and $P\{O(^1S)\text{-C}\} = [O(^3P)](R_{t10.1}[O_2(A)] + R_{d9.1}[O_2(A')])$. The production term was calculated as follows: $P\{O(^1S)\} = [O(^3P)](R_{t10.1}[O_2(A)] + R_{d9.1}[O_2(A')]) + [O(^3P)]R_{c2.1}[O_2(c)]$.

The loss term was calculated as follows: $L\{O(^1S)\} = L\{O(^1S)\text{-M}\} + L\{O(^1S)\text{-H}\} + L\{O(^1S)\text{-C}\} = [O(^1S)] \times D_g$, where $L\{O(^1S)\text{-M}\}$ **is absent**, $L\{O(^1S)\text{-H}\} = [O(^1S)] \times (R_{g1.2}[O_2] + R_{g3.0} + R_{g4.0})$ and $L\{O(^1S)\text{-C}\} = [O(^1S)] \times (R_{g1.1}[O(^3P)] +$

20  $R_{g1.3}[O_3] + R_{g2.1-2}\{[N_2],[O_2(a)]\})$ **resulting in** $D_g = R_{g1.1-3}\{[O(^3P)],[O_2],[O_3]\} + R_{g2.1-2}\{[N_2],[O_2(a)]\} + R_{g3.0} + R_{g4.0}$.

$[O(^1S)]$ values were retrieved taking M-, H- and C-processes into account as follows: R-$[O(^1S)] = [O(^1S)] = [O(^1S)\text{-M}] + [O(^1S)\text{-H}] + [O(^1S)\text{-C}]$, where $[O(^1S)\text{-M}]$ **is absent**, $[O(^1S)\text{-H}] = P\{O(^1S)\text{-H}\}/(D_g D_c)$ and $[O(^1S)\text{-C}]$ **is absent**. In the case when oxygen green line emissions are not given, $[O(^1S)]$ values can be retrieved on the basis of already known $[O(^3P)]$

25  values.

$[O(^1S)]$ values were also evaluated (E-$[O(^1S)]$) on the basis of retrieved VER$\{O(^1S-{}^1D)\}$ values (R-VER$\{O(^1S-{}^1D)\}$) **using the corresponding transition probability:** E-$[O(^1S)] = $ R-VER$\{O(^1S-{}^1D)\}/R_{g3.0}$.

Finally, VER$\{O(^1S-{}^1D)\}$ values were evaluated (E-VER$\{O(^1S-{}^1D)\}$) on the basis of retrieved $[O_2(a)]$ values and the respective transition probability: E-VER$\{O(^1S-{}^1D)\} = $ R-$[O_2(a)] \times R_{g3.0}$.

30  $[O(^1S)]$ values were retrieved and then evaluated to compare and verify these calculations. VER$\{O(^1S-{}^1D)\}$ values were also evaluated to compare them with **the retrieved values in order to verify the MAC calculations**, see Section A4.2.

**A4.2 Substep 2: consistency tests in the *calculation* of $[O(^1S)]$**

The consistency tests in the *calculations* by using the MAC model is based on the comparison of the retrieved and evaluated values.

The *calculation* step 4.1 described in Section A4.1 was carried out to retrieve R-$[O(^1S)]$ and $[O(^3P)]$ values on the basis of R-VER$\{O(^1S-^1D)\}$ values and concentrations of available excited chemical species. E-$[O(^1S)]$ values were also evaluated to compare them with R-$[O(^1S)]$ values. Additionally, E-VER$\{O(^1S-^1D)\}$ values were also evaluated to compare them with R-VER$\{O(^1S-^1D)\}$ values.

**A5 The 5$^{th}$ retrieval step**

The 5$^{th}$ retrieval step was performed to calculate $[O_x]$ ($[O(^3P)]$, $[O(^1D)]$ and $[O_3]$) values on the basis of concentrations of all relevant chemical species.

**A5.1 Substep 1: *calculation* of $[O(^3P)]$ involving all relevant chemical species**

$[O(^3P)]$ values were retrieved ($[O(^3P)]$-R) on the basis of concentrations of atmospheric minor species obtained in the previous *calculation* steps according to the continuity equation for $[O(^3P)]$ considering all relevant processes of the MAC model. For instance, values of $[O_2(A)]$, $[O_2(A')]$, $[O_2(b)]$, $[O_2(c)]$, $[O_2(a)]$ and $[O(^1S)]$ were obtained **at steps 2.1, 2.2, 2.3, 3.1, 3.2 and 4.1, respectively.**

The continuity equation for $[O(^3P)]$ including terms of the $[O(^3P)]$ production ($P\{O(^3P)\}$) and loss ($L\{O(^3P)\}$) is as follows: $d[O(^3P)]/dt = P\{O(^3P)\} - L\{O(^3P)\} = 0$. The production and loss terms were calculated **considering the processes shown in Tables 5 and 6.**

The production term **consists of** terms related to the M-model discussed in Section 3.2.1 ($P\{O(^3P)\}$-M), to the H-model discussed in Section 3.2.2 ($P\{O(^3P)\}$-H) and the complementary processes relevant here ($P\{O(^3P)\}$-C):
$P\{O(^3P)\} = P\{O(^3P)\text{-M}\} + P\{O(^3P)\text{-H}\} + P\{O(^3P)\text{-C}\}$, where $P\{O(^3P)\text{-M}\} = [O(^1D)]R_{r2.1,3}\{[N_2],[O_2]\} + (R_{s1.1-2} + 2R_{s1.3-4})[O_2]$, $P\{O(^3P)\text{-H}\} = [O(^1S)](R_{g1.2}[O_2] + R_{g4.0})$ and $P\{O(^3P)\text{-C}\} = [O_2(b)]R_{b4.1}[O_3] + [O_2(a)]R_{a2.1}[O_3]$ $+[O(^1S)]R_{g2.1-2}\{[N_2],[O_2(a)]\} + [O(^1D)](R_{r1.1}[O(^3P)] + R_{r2.2,4}\{[O_2],[CO_2]\} + R_{r3.0} + 2R_{r1.2}[O_3]) + 3R_{s2.2}[O_3] + R_{s2.5-6}[O_3] +$ $[H]R_{h6.3}[HO_2]$. The production term was calculated as follows: $P\{O(^3P)\} = ([O_2(b)]R_{b4.1} + [O_2(a)]R_{a2.1})[O_3] + [O(^1S)](R_{g1.2}[O_2]$ $+R_{g2.1-2}\{[N_2],[O_2(a)]\} + R_{g4.0}) + [O(^1D)](R_{r1.1}[O(^3P)] + 2R_{r1.2}[O_3] + R_{r2.1-4}\{[N_2],[O_2],[O_2],[CO_2]\} + R_{r3.0}) + (R_{s1.1-2} +$ $2R_{s1.3-4})[O_2] + 3R_{s2.2}[O_3] + R_{s2.5-6}[O_3] + [H]R_{h6.3}[HO_2]$.

The loss term was calculated as follows: $L\{O(^3P)\} = L\{O(^3P)\text{-M}\} + L\{O(^3P)\text{-H}\} + L\{O(^3P)\text{-C}\} = [O(^3P)] \times D_o$, where $L\{O(^3P)\text{-M}\} = [O(^3P)] \times ([O(^3P)]R_{a1.1-2}\{[N_2],[O_2]\})$, $L\{O(^3P)\text{-H}\} = [O(^3P)] \times ([O(^3P)](R_{x1.1-2} + R_{c1.1-2} + R_{b1.1-2})\{[N_2],[O_2]\} + R_{c2.1}[O_2(c)])$, $L\{O(^3P)\text{-C}\} = [O(^3P)] \times (R_{t10.1}[O_2(A)] + R_{d9.1}[O_2(A')] + [O(^3P)](R_{t1.1-2} +$ $R_{d1.1-2})\{[N_2],[O_2]\} + R_{x2.1}[O_3] + [O_2]R_{x3.1-2}\{[N_2],[O_2]\} + R_{h2.1}[OH^*] + R_{h4.1}[HO_2])$ **resulting in** $D_o = R_{t10.1}[O_2(A)] +$ $R_{d9.1}[O_2(A')] + [O(^3P)](R_{x1.1-2} + R_{t1.1-2} + R_{d1.1-2} +$ $R_{c1.1-2} + R_{b1.1-2} + R_{a1.1-2})\{[N_2],[O_2]\} + R_{x2.1}[O_3] + [O_2]R_{x3.1-2}\{[N_2],[O_2]\} + R_{h2.1}[OH^*] + R_{h4.1}[HO_2] + R_{c2.1}[O_2(c)]$.

$[O(^3P)]$ values were retrieved taking M-, H- and C-processes into account as follows: $[O(^3P)\text{-R}] = [O(^3P)\text{-M}]+[O(^3P)\text{-H}]+$ $[O(^3P)\text{-C}]$, where $[O(^3P)\text{-M}] =$

$\left([O(^1D)]R_{r2.1,3}\{[N_2],[O_2]\} + (R_{s1.1-2} + 2R_{s1.3-4})[O_2]\right)/D_o$, $[O(^3P)\text{-H}] = \left([O(^1S)](R_{g1.2}[O_2] + R_{g4.0})\right)/D_o$ and $[O(^3P)\text{-C}] =$ $\left([O_2(b)]R_{b4.1}[O_3] + [O_2(a)]R_{a2.1}[O_3] + [O(^1S)]R_{g2.1-2}\{[N_2],[O_2(a)]\} + 3R_{s2.2}[O_3] + R_{s2.5-6}[O_3]\right)/D_o$

5  $+ \left([O(^1D)](R_{r1.1}[O(^3P)] + R_{r2.2,4}\{[O_2],[CO_2]\} + R_{r3.0} + 2R_{r1.2}[O_3]) + [H]R_{h6.3}[HO_2]\right)/D_o.$

The final equation for $[O(^3P)]$ is as follows: $[O(^3P)\text{-R}] = [O(^3P)] = \left([O_2(a)]R_{a2.1}[O_3] + [O_2(b)]R_{b4.1}[O_3]\right)/D_o$ $+ \left([O(^1S)](R_{g1.2}[O_2] + R_{g2.1-2}\{[N_2],[O_2(a)]\} + R_{g4.0})\right)/D_o$ $+ \left([O(^1D)](R_{r1.1}[O(^3P)] + R_{r2.1-4}\{[N_2],[O_2],[O_2],[CO_2]\} + R_{r3.0} + 2R_{r1.2}[O_3])\right)/D_o$ $+ \left((R_{s1.1-2} + 2R_{s1.3-4})[O_2] + 3R_{s2.2}[O_3] + R_{s2.5-6}[O_3] + [H]R_{h6.3}[HO_2]\right)/D_o.$

10  **A5.2  Substep 2: *calculation* of $[O(^1D)]$ involving all relevant chemical species**

$[O(^1D)]$ values were retrieved ($[O(^1D)\text{-R}]$) on the basis of concentrations of atmospheric minor species obtained in the previous *calculation* steps according to the continuity equation for $[O(^1D)]$ considering all relevant processes of the MAC model.

The continuity equation for $[O(^1D)]$ including terms of the $[O(^1D)]$ production ($P\{O(^1D)\}$) and loss ($L\{O(^1D)\}$) is as follows: $d[O(^1D)]/dt = P\{O(^1D)\} - L\{O(^1D)\} = 0.$

15  The production and loss terms were calculated considering the processes shown in Tables 5, 6 and 7.

The calculation of the production term was based on the considered M-, H- and C-processes as follows: $P\{O(^1D)\} =$ $P\{O(^1D)\text{-M}\} + P\{O(^1D)\text{-H}\} + P\{O(^1D)\text{-C}\}$, where $P\{O(^1D)\text{-M}\} = R_{s1.1-2}[O_2] + R_{s2.3}[O_3]$, $P\{O_2(^1D)\text{-H}\} = R_{g3.0}[O(^1S)]$ and $P\{O_2(^1D)\text{-C}\} = [O(^1S)]2R_{g1.1}[O(^3P)]+R_{s2.4}[O_3]$ **resulting in** $P\{O(^1D)\} = [O(^1S)](2R_{g1.1}[O(^{.}$ $R_{g3.0}) + R_{s1.1-2}[O_2] + R_{s2.3-4}[O_3].$

20  The calculation of the loss term was based on the considered M-, H- and C-processes as follows: $L\{O(^1D)\} = L\{O(^1D)\text{-M}\}+$ $L\{O(^1D)\text{-H}\} + L\{O(^1D)\text{-C}\} = [O(^1D)] \times D_r$, where $L\{O(^1D)\text{-M}\} = R_{r2.1,3}\{[N_2],[O_2]\}$, $L\{O(^1D)\text{-H}\}$ **is absent** and $L\{O(^1D)\text{-C}\} = R_{r1.1-3}\{[O(^3P)],[O_3],[O_3]\}+R_{r2.2,4}\{[O_2],[CO_2]\}+R_{r3.0}$ **resulting in** $D_r = R_{r1.1-3}\{[O(^3P)],[O_3],[O_3]\}+$

$R_{r2.1-4}\{[N_2],[O_2],[O_2],[CO_2]\} + R_{r3.0}.$

25  $[O(^1D)]$ values were retrieved taking M-, H- and C-processes into account as follows: $[O(^1D)\text{-R}] = [O(^1D)] = [O(^1D)\text{-M}]+$ $[O(^1D)\text{-H}] + [O(^1D)\text{-C}]$, where $[O(^1D)\text{-M}] = (R_{s1.1-2}[O_2] + R_{s2.3}[O_3])/D_r$, $[O(^1D)\text{-H}] = \left(R_{g3.0}[O(^1S)]\right)/D_r$ and $[O(^1D)\text{-C}] = \left([O(^1S)]2R_{g1.1}[O(^3P)]\right)/
[revised manuscript text omitted]

---

## Referee Report (RR1)

**Referee Report**

**Manuscript no**.: acp-2019-221

**Title:** Photochemical modeling of molecular and atomic oxygen based on multiple *in situ* emissions measured during the Energy Transfer in the Oxygen Nightglow rocket campaign.

**Authors:** Olexandr Lednyts'kyy and Christian von Savigny

In this paper the authors present a new airglow model (MAC, Multiple Airglow Chemistry model) that includes electronically excited states of molecular and atomic oxygen (six of $O_2$ and two of O) and their ground states. The model is based on the measurements and findings of the ETON sounding rocket campaign conducted from South Uist, Scotland in March 1982 and extends this with later efforts by several authors to model the photochemistry of the MLT (Mesosphere/Lower Thermosphere) region, and updated reaction rates. Unfortunately, the *in situ* measurements of the atmospheric neutral temperature during the ETON campaign were not successful. Instead, the temperature (and neutral density) were taken from the NRLMSISE-00 model in the current study. A sensitivity study was conducted by the authors to investigate the influence of changes in temperature and neutral density in the retrieval of the different excited and ground states of molecular and atomic oxygen.

**General comments**

The paper presents an extensive model to explain the excitation mechanisms responsible for the observed airglow emissions from the MLT region of the Earth's atmosphere. It is a nice review of the current knowledge of airglow photochemistry, and it constrain the precursors responsible for the Atmospheric band, Infrared Atmospheric band and the Oxygen Green Line emissions. It is an important contribution to the scientific community.

The authors have done a good job in revising the manuscript in accordance with the Referee Comments in the interactive discussion. The structure and language have been substantially improved and the manuscript is much easier to read

**Specific comments**

A few very minor (technical) suggestions:

On page 27 line 3 it is written "…are not known *a-priori*. Instead…". Write "*a priori*" here to be consistent.

On page 37 line 10: "…the Stern-Volmer method Lakowicz (2006)." Put Lakowicz inside the parenthesis, i.e. "…the Stern-Volmer method (Lakowicz, 2006)."

On page 43 line 15: "…complementary processes proposing the MAC model." Change "proposing" to "proposed".

---

## Author Response (AR2)

*The letter to the Handling Co-Editor William Ward including the response regarding comments of the Co-Editor and Anonymous Referees #1 and #2 on acp-2019-221:*

**Photochemical modeling of molecular and atomic oxygen based on multiple *in-situ* emissions measured during the Energy Transfer in the Oxygen Nightglow rocket campaign**

*Comments of the Co-Editor*
*Feedback:*

Your paper has been revised and clarified to the satisfaction of the reviewers and is close to being ready for publication in ACP. There are a few minor changes recommended by the reviewers in the body of the paper which need to be addressed before final submission.

In addition, there are a couple of more substantial changes which do not affect the main content of the paper but do affect its accessibility and value to the reader.

*List of corrections regarding specific comments of the Co-Editor*

The authors of the manuscript are grateful to the Co-Editor for the specific comments. The manuscript was revised and has been hopefully improved to make it easier to follow. Changes provided in this response are indicated by numbers of lines and pages by black (red) font to find these changesin the previous (current) version of the manuscript.

*Specific comment:*

The first is to carefully go through the paper and change "retrieve" to "calculate" for all steps that are based on direct calculation of new quantities from known ones. Use the term retrieve for situations where the statistics and sampling of observations need to be taken into account when deriving an output profile. As suggested by the Anonymous Referee #2 this is the conventional way to use these terms and not to do so causes confusion.

*Response:*

The authors agree with the comments of the Co-Editor and the Anonymous Referee #2 that these terms should not cause confusion. The Anonymous Referee #2 preferred to avoid the term "retrieve" for the two cases when (1) the division by an Einstein A coefficient and (2) a population scaling factor is employed. Note that in the first case we use the term "evaluate". We assume that, in the second case, the Anonymous Referee #2 meant the increasing of the association rates of $O_2(b, a, X)$ in the three-body recombination reactions. In order to avoid confusion when this assumption regarding the second case is not true, we consider the comment of the Co-Editor: "Use the term retrieve for situations where the statistics and sampling of observations need to be taken into account when deriving an output profile." In this regard we would like to emphasize that our method is applied increasing of some association rate values in the three-body recombination reactions to determine oxygen concentration profiles on the basis of VER profiles, and VER profiles were retrieved using

statistical procedures on the basis of sampled integrated emission profiles.

In the Appendix A2, A3.2, A3.3 and A4.1, where VER profiles were employed to retrieve oxygen concentration profiles, we are going to use the term "retrieve". We are going to use the term "calculate" in the Appendix A1.1, A1.2, A1.3, A3.1 and A5, where VER profiles were not employed. To avoid confusion comparing three kinds of the MAC products, we are going to use the term "retrieve" to mark concentration profiles by a character R and summarizing calculations and evaluations in the Appendix or elsewhere in the article. Our decision is backed up by using the search engine GoogleScholar for the two following cases: (1) retrieve oxygen density rocket VER and (2) calculate oxygen density rocket VER. Both terms are used approximately equally in the literature.

*The following relevant change was carried out in lines 6-7 (30-31) on page 41 (41), line 7 (7) on page 42 (42):*
. . .retrieve. . .
was changed to:
. . .calculate. . .

*Additionally, the following relevant change was carried out in line 22 (18) on page 44 (45), line 4 (29) on page 45 (45), lines 10-12 (4-6) on page 45 (46), lines 22, 24 (16, 18) on page 45 (46), lines 8, 11 (1, 4) on page 46 (47), lines 6, 9, 20 (1, 4, 15) on page 48 (49), line 4 (28) on page 49 (49), lines 17, 19 (11, 13) on page 49 (50), lines 5, 7, 26 (2, 4, 23) on page 50 (51), line 18 (15) on page 51 (52), line 1 (31) on page 52 (52):*
. . .retrieved. . .
was changed to:
. . .calculated. . .

*Additionally, the following relevant change was carried out in line 24 (18) on page 45 (46), lines 23, 26, 29 (16, 19, 22) on page 46 (47), lines 16, 22 (12, 18) on page 47 (48), lines 19, 21 and 25 (14, 16, 19) on page 48 (49), line 29 (24) on page 49 (50), line 6 (3) on page 50 (51):*
. . .calculation step. . .
was changed to:
. . .step. . .

*Additionally, the following relevant change was carried out in line 3 (32) on page 43 (42), line 7 (3) on page 43 (44), line 21 (16) on page 48 (49):*
. . .retrievals. . .
was changed to:
. . .calculations carried out. . .

*Additionally, the following relevant change was carried out in lines 5, 26, 31 (2, 23, 28) on page 50 (51):*

... $[O(^3P)\text{-R}]$ ...

was changed to:

... $R\text{-}[O(^3P)]$ ...

*Additionally, the following relevant change was carried out in lines 4, 18, 21 (2, 15, 18) on page 51 (52):*

... $[O(^1D)\text{-R}]$ ...

was changed to:

... $R\text{-}[O(^1D)]$ ...

*Additionally, the following relevant change was carried out in lines 1, 3 (31, 33) on page 52 (52):*

... $[O_3\text{-R}]$ ...

was changed to:

... $R\text{-}[O_3]$ ...

***Specific comment:***

The Anonymous Referee #2 also suggested that Appendix A is not written clearly enough for the reader to duplicate the model. This is a necessary requirement for scientific papers and I agree with their opinion. To rectify this, I suggest adding a couple of paragraphs at the beginning of this Appendix describing how a user should approach using the MAC model and what they might use it for. As currently written, the first couple of paragraphs address details which a typical reader would initially not be familiar with. This is useful information, but only once the reader has set up the model.

Thank you for your work on revising and clarifying the paper to this point.

***The following paragraphs were added after line 3 (3) on page 41 (41):***
The MAC model was implemented to study the photochemistry of excited oxygen species in the MLT. $[O(^3P)]$ retrievals are carried out sequentially and start with higher excited $O_2$ species, concentrations of which are applied at the next retrieval steps to obtain concentrations of lower excited $O_2$ and O species, see Table 12. During the first $[O(^3P)]$ retrieval steps, available VER profiles of strong emissions are employed to retrieve concentrations of the corresponding excited oxygen species and $[O(^3P)]$, see Sections A2.1, A2.2, A2.3, A3.1, A3.2 and A4.1. Retrieving $[O(^3P)]$ profiles on the basis of VER profiles is widely accepted in the scientific community dealing with processing of remote and *in situ* measurements. The last retrieval step is applied retrieve concentrations of odd oxygen species on the basis of concentrations of all relevant chemical species, see Sections A5.1, A5.2 and A5.3 for details regarding calculations of $[O(^3P)]$, $[O(^1D)]$ and $[O_3]$, respectively. The last retrieval step was conceptualized keeping in mind that the obtained system of reactions should in the end be incorporated in a General Circulation Model (GCM), where $[O(^3P)]$ and concentrations of excited oxygen species are simulated. Calculations carried out by using a GCM

are usually initialized on the basis of *a priori* values of concentrations of excited $O_2$ and O species, and these concentrations were retrieved by using the MAC model at the first retrieval steps accurately.

During the first retrieval steps, the MAC calculations are carried out on the basis of multiple VER profiles of strong nightglow emissions discussed using Table 1. The obtained verification and validation results, see Section 3.5, enabled assessing the most effective group of emissions for the measurement, e.g., of $[O(^3P)]$. This group is represented by emissions in the Atmospheric band, the Infrared Atmospheric band and the oxygen green line emission. Additionally, the results obtained studying the influence of perturbations in parameters of the MAC model on the retrieved $[O(^3P)]$ profiles, see Fig. 6 in Section 3.6 for details, enabled assessing the most effective emission line for the $[O(^3P)]$ retrievals. This emission line measured at $761.9\,\mathrm{nm}$ is represented by transitions $O_2(b-X)\{0-0\}$ in the Atmospheric band. Figure 6 enables concluding that only profiles of temperature, atmospheric density and $\mathrm{VER}\{O_2(b-X)\}$ are required for the $[O(^3P)]$ retrievals, see Section A2.3 for details. Another essential characteristic of the MAC model is that calculations discussed in Section A2.3 are carried out by using simple steady state chemical balance equations (referred to as continuity equations) represented by the polynomial equations of the second or third degree with respect to $[O(^3P)]$. Solutions of such equations are easy to interpret. These findings might be of great help to the scientific community dealing with processing of remote and *in situ* measurements to design future $[O(^3P)]$ experiments.

***Additionally, the following relevant change was carried out in lines 6-7 (31) on page 41 (41):***
Then simple steady state chemical balance equations (referred to as continuity equations)...
was changed to:
Then continuity equations...

***Additionally, the following relevant change was carried out in line 25 (16-21) on page 41 (42):***
There are three kinds of the MAC products:
was changed to:
onsidering M-, H- and C-processes involving $O_2(A)$, the continuity equation of the second degree with respect to $[O(^3P)]$ is established and solved for $[O(^3P)]$ values on the basis of the corresponding VER profile values denoted "retrieved" values and marked with a character R, i.e. $\mathrm{R}\text{-}\mathrm{VER}\{O_2(A-X)\}$. Then $[O_2(A)]$ values are computed using the continuity equation and denoted "calculated" values at step 2.1. Summarizing retrievals and evaluations in the following sections or elsewhere in the article, $[O_2(A)]$ values are also denoted "retrieved" values and marked with a character R, i.e. $\mathrm{R}\text{-}[O_2(A)]$, in order to emphasize that $[O_2(A)]$ values are computed on the basis of retrieved $[O(^3P)]$ values. This notation is employed in order to avoid confusion comparing three kinds of the MAC products:

*Additionally, the following relevant change was carried out in line 27 (24) on page 41 (42):*
R-[O$_2$($b$)]
was changed to:
R-[O$_2$($A$)]

*Additionally, the following relevant change was carried out in line 30 (27) on page 41 (42):*
E-[O$_2$($b$)]
was changed to:
E-[O$_2$($A$)]

*Additionally, the following relevant change was carried out in line 32 (29) on page 41 (42):*
E-VER$\{$O$_2$($b - X$)$\}$
was changed to:
E-VER$\{$O$_2$($A - X$)$\}$

*Comments of the Anonymous Referee #1*
*Feedback:*
In this paper the authors present a new airglow model (MAC, Multiple Airglow Chemistry model) that includes electronically excited states of molecular and atomic oxygen (six of O2 and two of O) and their ground states. The model is based on the measurements and findings of the ETON sounding rocket campaign conducted from South Uist, Scotland in March 1982 and extends this with later efforts by several authors to model the photochemistry of the MLT (Mesosphere/Lower Thermosphere) region, and updated reaction rates. Unfortunately, the in situ measurements of the atmospheric neutral temperature during the ETON campaign were not successful. Instead, the temperature (and neutral density) were taken from the NRLMSISE-00 model in the current study. A sensitivity study was conducted by the authors to investigate the influence of changes in temperature and neutral density in the retrieval of the different excited and ground states of molecular and atomic oxygen.

*General comments:*
The paper presents an extensive model to explain the excitation mechanisms responsible for the observed airglow emissions from the MLT region of the Earths atmosphere. It is a nice review of the current knowledge of airglow photochemistry, and it constrain the precursors responsible for the Atmospheric band, Infrared Atmospheric band and the Oxygen Green Line emissions. It is an important contribution to the scientific community.

The authors have done a good job in revising the manuscript in accordance

with the Referee Comments in the interactive discussion. The structure and language have been substantially improved and the manuscript is much easier to read.

**List of corrections regarding specific comments of the Anonymous Referee #1**

The authors of the manuscript are grateful to the Anonymous Referee #1 for the general and specific comments. The manuscript was revised and has been hopefully improved to make it easier to follow. Changes provided in this response are indicated by numbers of lines and pages by black (red) font to find these changesin the previous (current) version of the manuscript.

**Specific comment:**

A few very minor (technical) suggestions. On page 27 line 3 it is written "...are not known *a-priori*. Instead...". Write "*a priori*" here to be consistent.

**Page 27 (27), line 3 (3):**
...are not known *a-priori*.
was changed to:
...are not known *a priori*.

**Specific comment:**

On page 37 line 10: "...the Stern-Volmer method Lakowicz (2006)." Put Lakowicz inside the parenthesis, i.e. "...the Stern-Volmer method (Lakowicz, 2006)."

**Page 37 (37), line 5 (5):**
...the Stern-Volmer method Lakowicz (2006).
was changed to:
... the Stern-Volmer method (Lakowicz, 2006).

**Specific comment:**

On page 43 line 15: "...complementary processes proposing the MAC model." Change "proposing" to "proposed".

**Page 43 (44), line 8 (4):**
...complementary processes proposing the MAC model.
was changed to:
...complementary processes in the MAC model.

**Comments of the Anonymous Referee #2**
**Feedback:**

Firstly I would like to thank the authors for the efforts they have put into revising this paper. The revised version is much more readable particularly in the first sections. I would also apologise for taking such a long time to complete this review.

The paper provides a comprehensive review of the many papers and model in the area of nightglow emissions and attempts to produce an all inclusive model that relates the concentration of atomic oxygen to the volume emission rates. I believe that this paper makes a significant contribution to the field and should be published.

**List of corrections regarding specific comments of the Anonymous Referee #2**

The authors of the manuscript are grateful to the Anonymous Referee #2 for the specific comments. The manuscript was revised and has been hopefully improved to make it easier to follow. Changes provided in this response are indicated by numbers of lines and pages by black (red) font to find these changesin the previous (current) version of the manuscript.

**Specific comment to page 32, line 22:**

page 34 line 22: It is here correctly stated that the accuracy of the ETON profiles is in the order of 10-20%. The accuracy of the integrated emission intensities should however be better than this probably better that 5%. Have the authors considered this in their model comparisons?

**Response:**

VER profiles published in McDade et al. (1986), Greer et al. (1986) and Greer et al. (1987) were retrieved on the basis of Integrated Emission Rate (IER) profiles measured during the ETON campaign. Unfortunately, in these works IER profiles were not published to retrieve VER profiles again. If new VER profiles with higher accuracy values were obtained, then this could be considered in the model comparisons explicitely. We would like to emphasize that this had been implicitely considered using our appendix and Fig. 6 already. Specifically, readers of our article can see the dependence of the retrieved $[O(^3P)]$ values on VER values of the considered emission using calculations provided in our appendix. This enables the readers to scale $[O(^3P)]$ profiles shown in our Fig. 6 to consider this in the model comparisons.

**Specific comment:**

Appendix A: I still find this section to be very heavy reading and difficult to follow. I wonder whether a reader could recreate the MAC model and all of these steps.

**Response:**

The response to this specific comment was worked out in the text provided above according to insightful comments of the Co-Editor.

**Specific comment:**

I also do not really like the use of the word retrieve in the first part of the first sentences of sections A2.x with respect to the VER. I did mention that I though that calculate would be a better word in my first review. Retrieve to me tends to imply more than the division by an Einstein A coefficient (or a population scaling factor) as appropriate.

***Response:***
The response to this specific comment was worked out in the text provided above according to insightful comments of the Co-Editor.

[revised manuscript text omitted]
)\text{-C}\})$: $P\{O(^1S)\} =$ 5   $P\{O(^1S)\text{-M}\} + P\{O(^1S)\text{-H}\} + P\{O(^1S)\text{-C}\}$, where $P\{O(^1S)\text{-M}\}$ is absent, $P\{O(^1S)\text{-H}\} = [O(^3P)]R_{c2.1}[O_2(c)]$ and $P\{O(^1S)\text{-C}\} = [O(^3P)](R_{t10.1}[O_2(A)] + R_{d9.1}[O_2(A')])$. The production term was calculated as follows: $P\{O(^1S)\} = [O(^3P)](R_{t10.1}[O_2(A)] + R_{d9.1}[O_2(A')]) + [O(^3P)]R_{c2.1}[O_2(c)]$.

The loss term was calculated as follows: $L\{O(^1S)\} = L\{O(^1S)\text{-M}\} + L\{O(^1S)\text{-H}\} + L\{O(^1S)\text{-C}\} = [O(^1S)] \times D_g$, where $L\{O(^1S)\text{-M}\}$ is absent, $L\{O(^1S)\text{-H}\} = [O(^1S)] \times (R_{g1.2}[O_2] + R_{g3.0} + R_{g4.0})$ and $L\{O(^1S)\text{-C}\} = [O(^1S)] \times (R_{g1.1}[O(^3P)] +$ 10   $R_{g1.3}[O_3] + R_{g2.1-2}\{[N_2], [O_2(a)]\})$ resulting in $D_g = R_{g1.1-3}\{[O(^3P)], [O_2], [O_3]\} + R_{g2.1-2}\{[N_2], [O_2(a)]\} + R_{g3.0} + R_{g4.0}$.

$[O(^1S)]$ values were calculated taking M-, H- and C-processes into account as follows: R-$[O(^1S)] = [O(^1S)] = [O(^1S)\text{-M}] + [O(^1S)\text{-H}] + [O(^1S)\text{-C}]$, where $[O(^1S)\text{-M}]$ is absent, $[O(^1S)\text{-H}] = P\{
[revised manuscript text omitted]